# Stochastic Bias-Reduced Gradient Methods

Hilal Asi[*]   Yair Carmon[†]   Arun Jambulapati[*]   Yujia Jin[*]   Aaron Sidford[*]

## Abstract

We develop a new primitive for stochastic optimization: a low-bias, low-cost estimator of the minimizer $x_\star$ of any Lipschitz strongly-convex function. In particular, we use a multilevel Monte-Carlo approach due to Blanchet and Glynn [8] to turn any optimal stochastic gradient method into an estimator of $x_\star$ with bias $\delta$, variance $O(\log(1/\delta))$, and an expected sampling cost of $O(\log(1/\delta))$ stochastic gradient evaluations. As an immediate consequence, we obtain cheap and nearly unbiased gradient estimators for the Moreau-Yoshida envelope of any Lipschitz convex function, allowing us to perform dimension-free randomized smoothing. We demonstrate the potential of our estimator through four applications. First, we develop a method for minimizing the maximum of $N$ functions, improving on recent results and matching a lower bound up to logarithmic factors. Second and third, we recover state-of-the-art rates for projection-efficient and gradient-efficient optimization using simple algorithms with a transparent analysis. Finally, we show that an improved version of our estimator would yield a nearly linear-time, optimal-utility, differentially-private non-smooth stochastic optimization method.

## 1   Introduction

Consider the fundamental problem of minimizing a $\mu$-strongly convex function $F : \mathcal{X} \to \mathbb{R}$ given access to a stochastic (sub-)gradient estimator $\hat{\nabla} F$ satisfying $\mathbb{E}\, \hat{\nabla} F(x) \in \partial F(x)$ and $\mathbb{E}\|\hat{\nabla} F(x)\|^2 \leq G^2$ for every $x \in \mathcal{X}$. Is it possible to transform the unbiased estimator $\hat{\nabla} F$ into a (nearly) unbiased estimator of the minimizer $x_\star := \operatorname{argmin}_{x \in \mathcal{X}} F(x)$? In particular, can we improve upon the $O(G/(\mu\sqrt{T}))$ bias achieved by $T$ iterations of stochastic gradient descent (SGD)?

In this paper, we answer this question in the affirmative, proposing an *optimum estimator* $\hat{x}_\star$, which (for any fixed $\delta > 0$) has

$$\text{bias } \|\mathbb{E}\hat{x}_\star - x_\star\| = O(\delta) \text{ and variance } \mathbb{E}\|\hat{x}_\star - \mathbb{E}\hat{x}_\star\|^2 = O\left(\frac{G^2}{\mu^2}\log\left(\frac{G}{\mu\delta}\right)\right),$$

and, *in expectation*, costs $O(\log(\frac{G}{\mu\delta}))$ evaluations of $\hat{\nabla} F$.[3] Setting $\delta = G/(\mu\sqrt{T})$, we obtain the same bias bound as $T$ iterations of SGD, but with expected cost of only $O(\log T)$ stochastic gradient evaluations (the worst-case cost is $T$). Further, the bias can be made arbitrarily small with only logarithmic increase in the variance and the stochastic gradient evaluations of our estimator, and therefore—paralleling the term "nearly linear-time" [27]—we call $\hat{x}_\star$ *nearly unbiased*.

Our estimator is an instance of the multilevel Monte Carlo technique for de-biasing estimator sequences [25] and more specifically the method of Blanchet and Glynn [8]. Our key observation is that this method is readily applicable to strongly-convex variants of SGD, or indeed any stochastic optimization method with the same (optimal) rate of convergence.

---

[*]Stanford University, {asi,jmblpati,yujiajin,sidford}@stanford.edu

[†]Tel Aviv University, ycarmon@tauex.tau.ac.il

[3]When $\mathcal{X} = \mathbb{B}_R(x_0) \subset \mathbb{R}^d$, $F(x) = \frac{1}{n}\sum_{i\in[n]} \hat{F}(x;i)$, and $\hat{\nabla} F$ is the subgradient of a uniformly random $\hat{F}(x;i)$ we can also get an estimator with bias 0 and expected cost $O(\log(nd))$. See Appendix A.1 for details.

35th Conference on Neural Information Processing Systems (NeurIPS 2021).

| Objective | Expensive operation $\mathcal{O}$ | $\mathcal{N}_{\mathcal{O}}$ | $\mathbb{E}\mathcal{N}_{\hat{\nabla} f}$ |
|---|---|---|---|
| $\max_{i \in [N]} f_{(i)}(x)$ (Sec. 4) | $f_{(1)}(x), \ldots, f_{(N)}(x)$ | $\widetilde{O}(\epsilon^{-2/3})$ | $\widetilde{O}(\epsilon^{-2})$ |
| $f(x)$ in domain $\mathcal{X}$ (Sec. 3) | $\mathsf{Proj}_{\mathcal{X}}(x)$ | $O(\epsilon^{-1})$ | " |
| $\Lambda(x) + f(x)$ for $L$-smooth $\Lambda$ (Sec. 5) | $\nabla \Lambda(x)$ | $O(\sqrt{L/\epsilon})$ | " |

**Table 1.** Summary of our applications of accelerated bias-reduced stochastic gradient methods. We use $\mathcal{N}_{\mathcal{O}}$ and $\mathcal{N}_{\hat{\nabla} f}$ to denote the number of expensive operations and subgradient estimations, respectively. The $\widetilde{O}$ notation hides polylogarithmic factors. See Section 1.2 for additional description.

## 1.1 Estimating proximal points and Moreau-Yoshida envelope gradients

Given a convex function $f$ and regularization level $\lambda$, the proximal point of $y$ is $\mathsf{P}_{f,\lambda}(y) :=$ $\mathrm{argmin}_{x \in \mathbb{R}^d} \{ f(x) + \frac{\lambda}{2} \|x - y\|^2 \}$. Since computing $\mathsf{P}_{f,\lambda}$ amounts to solving a $\lambda$-strongly-convex problem, our technique provides low-bias and cheap proximal point estimators. Proximal points are ubiquitous in optimization [43, 19, 52, 38] and estimating them efficiently with low bias opens up new algorithmic possibilities. One of these possibilities is estimating the gradient of the Moreau-Yoshida envelope $f_\lambda(y) := \min_{x \in \mathbb{R}^d} \{ f(x) + \frac{\lambda}{2} \|x - y\|^2 \}$, which is a $\lambda$-smooth, $G^2/(2\lambda)$-accurate approximation of any $G$-Lipschitz $f$ (see, e.g., [43, 29] and Appendix B.3). Since $\nabla f_\lambda(y) = \lambda(y - \mathsf{P}_{f,\lambda}(y))$, our optimum estimator provides a low-bias estimator for $\nabla f_\lambda(y)$ with second moment and expected cost greater than those of $\hat{\nabla} f$ by only a logarithmic factor. Thus, for any non-smooth $f$ we can turn $\hat{\nabla} f$ into a gradient estimator for the smooth surrogate $f_\lambda$, whose smoothness is independent of the problem dimension, allowing us to perform *dimension-free* randomized smoothing [20].

## 1.2 Applications via accelerated bias-reduced methods

Our optimum estimator is a new primitive in stochastic convex optimization and we expect it to find multiple applications. We now describe three such applications: the first improves on previously known complexity bounds while the latter two recover existing bounds straightforwardly. For simplicity of presentation we assume (in the introduction only) $\mathbb{E}\|\hat{\nabla} f\|^2 \leq 1$ and unit domain size.

In each application, we wish to minimize an objective function given access to a cheap subgradient estimator $\hat{\nabla} f$ as well as an expensive application-specific operation $\mathcal{O}$ (e.g., a projection to a complicated set). Direct use of the standard stochastic gradient method finds an $\epsilon$-accurate solution using $O(\epsilon^{-2})$ computations of both $\hat{\nabla} f$ and $\mathcal{O}$, and our goal is to improve the $\mathcal{O}$ complexity without hurting the $\hat{\nabla} f$ complexity.

To that end, we design stochastic accelerated methods consisting of $T$ iterations, each one involving only a constant number of $\mathcal{O}$ and proximal point computations, which we approximate by averaging copies of our optimum estimator.[4] Its low bias allows us to bound $T \ll \epsilon^{-2}$ as though our proximal points were exact, while maintaining an $\widetilde{O}(\epsilon^{-2})$ bound on the total expected number of $\hat{\nabla} f$ calls.[5] Thus, we save expensive operations without substantially increasing the gradient estimation cost. Table 1 summarizes each application, and we briefly describe them below.

**Minimizing the maximal loss (Section 4).** Given $N$ convex, 1-Lipschitz functions $f_{(1)}, \ldots, f_{(N)}$ we would like to find an $\epsilon$-approximate minimizer of their maximum $f_{\max}(x) = \max_{i \in [N]} f_{(i)}(x)$. This problem naturally arises when optimizing worst-case behavior, as in maximum margin classification and robust optimization [53, 15, 45, 6]. We measure complexity by the number of individual function and subgradient evaluations, so that the expensive operation of evaluating $f_{(1)}, \ldots, f_{(N)}$ at a single point has complexity $O(N)$ and the subgradient method solves this problem with complexity $O(N\epsilon^{-2})$. Carmon et al. [13] develop an algorithm for minimizing $f_{\max}$ with complexity $\widetilde{O}(N\epsilon^{-2/3} + \epsilon^{-8/3})$, improving on the subgradient method for sufficiently large $N$. Using our bias-reduced Moreau gradient envelope estimator in a Monteiro-Svaiter-type accelerated proximal point method [12, 11, 38], we obtain improved complexity $\widetilde{O}(N\epsilon^{-2/3} + \epsilon^{-2})$. This matches (up to logarithmic factors) a lower bound shown in [13], settling the complexity of minimizing the maximum

---

[4]While averaging is parallelizable, our optimum estimator itself is sequential. Consequently, our approach does not yield improve parallelism; see Appendix A.2 for further discussion

[5]It is easy to turn expected complexity bounds into deterministic ones; see Appendix A.3.

of $N$ non-smooth functions. Our result reveals a surprising fact: for $N \ll (GR/\epsilon)^{-4/3}$, minimizing the maximum of $N$ functions is no harder than minimizing their average.

**Projection-efficient optimization via dimension-free randomized smoothing (Section 3).** Consider the problem of minimizing a convex function $f$ using an unbiased gradient estimator $\hat{\nabla} f$ over convex set $\mathcal{X}$ for which Euclidean projections are expensive to compute (for example, the cone of PSD matrices). When $f$ is $L$-smooth, a stochastic version of Nesterov's accelerated gradient descent (AGD) [16] performs only $O(\sqrt{L/\epsilon})$ projections. For non-smooth $f$ we instead apply AGD to the Moreau envelope smoothing of $f$ (with appropriate $\lambda = O(\epsilon^{-1})$) using our nearly-unbiased stochastic estimator for $\nabla f_\lambda$. This yields a solution in $O(\epsilon^{-1})$ projections and $\widetilde{O}(\epsilon^{-2})$ evaluations of $\hat{\nabla} f$. Our algorithm provides a simple alternative to the recent work of Thekumparampil et al. [51] whose performance guarantees are identical up to a logarithmic factor.

**Gradient-efficient composite optimization (Section 5).** We would like to minimize $\Psi(x) = \Lambda(x) + f(x)$, where $\Lambda$ is convex and $L$-smooth but we can access it only via computing (expensive) exact gradients, while $f$ is a non-smooth convex functions for which we have a (cheap) unbiased subgradient estimator $\hat{\nabla} f$. Problems of this type include inverse problems with sparsity constraints and regularized loss minimization in machine learning [34]. To save $\nabla \Lambda$ computations, it is possible to use composite AGD [41] which solves $O(\sqrt{L/\epsilon})$ subproblems of the form $\text{minimize}_x \{ \langle \nabla \Lambda(y), x \rangle + f(x) + \frac{\beta}{2} \|x - x'\|^2 \}$. Lan [34] designed a specialized method, gradient sliding, for which the total subproblem solution cost is $O(\epsilon^{-2})$ evaluations of $\hat{\nabla} f$. We show that a simple alternative—estimating the subproblem solutions via our low-bias optimum estimator—recovers its guarantees up to logarithmic factors.

### 1.3 Non-smooth differentially private stochastic convex optimization

We now discuss a potential application of our technique that is conditional on the existence of an improved optimum estimator. In it, we minimize the population objective function $f(x) = \mathbb{E}_{S \sim P} \hat{f}(x; S)$ under the well-known constraint of differential privacy [22]. Given $n$ i.i.d. samples $S_i \sim P$ and assuming that each $\hat{f}$ is 1-Lipschitz, convex and sufficiently smooth, Feldman et al. [23] develop algorithms that obtain the optimal error and compute $O(n)$ subgradients of $\hat{f}$. The non-smooth case is more challenging and the best existing bound is $O(n^{11/8})$ for the high-dimensional setting $d = n$ [32, 3]. In Section 6 we show that our optimum estimator, combined with recent localization techniques [23], reduces the problem to private mean estimation. Unfortunately, our estimator is heavy-tailed, leading to insufficient utility. Nevertheless, assuming a version of our estimator that has bounded outputs, we give an algorithm that queries $\widetilde{O}(n)$ subgradients for non-smooth functions, solving a longstanding open problem in private optimization [14, 4]. This motivates the study of improved versions of our estimators that have constant sensitivity.

### 1.4 Related work

Multilevel Monte-Carlo (MLMC) techniques originate from the literature on parametric integration for solving integral and differential equations [25]. Our approach is based on an MLMC variant put forth by Blanchet and Glynn [8] for estimating functionals of expectations. Among several applications, they propose [8, Section 5.2] an estimator for $\text{argmin}_x \mathbb{E}_{S \sim P} \hat{f}(x; S)$ where $\hat{f}(\cdot; s)$ is convex for all $s$ and assuming access to minimizers of empirical objectives of the form $\sum_{i \in [N]} \hat{f}(x; s_i)$. The authors provide a preliminary analysis of the estimator's variance (later elaborated in [9]) using an asymptotic Taylor expansion around the population minimizer. In comparison, we study the more general setting of stochastic gradient estimators and provide a complete algorithm based on SGD, along with a non-asymptotic analysis and concrete settings where our estimator is beneficial.

A number of works have used the Blanchet-Glynn estimator in the context of optimization and machine learning. These applications include estimating the ratio of expectations for semisupervised learning [7], estimating gradients of distributionally robust optimization objectives [35], and estimating gradients in deep latent variable models [47]. Our estimator is similar to that of Levy et al. [35] in that we also have to pick a "critical" doubling probability for the (random) computational budget, which makes the expected cost and variance of our estimators depend logarithmically on the bias.

## 1.5 Limitations

Our paper demonstrates that our proposed optimum estimator is a useful *proof device*: it allows us to easily prove upper bounds on the complexity of structured optimization problems, and at least in one case (minimizing the maximum loss) improve over previously known bounds. However, our work does not investigate the *practicality* of our optimum estimator, as implementation and experiments are outside its scope.

Nevertheless, let us briefly discuss the practical prospects of the algorithms we propose. On the one hand, our optimum estimator itself is fairly easy to implement, adding only a few parameters on top of a basic gradient method. On the other hand, in the settings of Sections 3 and 5, gradient-sliding based methods [34, 51] are roughly as simple to implement and enjoy slightly stronger convergence bounds (better by logarithmic factors) than our optimum estimator. Consequently, in these settings we have no reason to assume that our algorithms are better in practice. In the setting of Section 4 (minimizing the maximum loss) our algorithm does enjoy a stronger guarantee than the previous best bound [13]. However, both our algorithm and [13] are based on an accelerated proximal point method that, in its current form, is not practical [13, Sec. 6.2]. Thus, evaluating the benefit of stochastic bias reduction in the context of minimizing the maximum loss would require us to first develop a practical accelerated proximal point algorithm, which is an open question under active research [see, e.g., 50].

Another limitation of our optimum estimator is that, while it has a bounded second moment, its higher moments are unbounded. While this does not matter for most of our results, the lack of higher moment bounds prevents us from setting the complexity of non-smooth private stochastic convex optimization in Section 6. Finding an optimum estimator that is bounded with high probability—or proving that one does not exist—remains an open question for future work.

Finally, our analyses are limited to convex objective functions. However, while outside the scope of the paper, we believe our results are possibly relevant for non-convex settings as well. In particular, for smooth non-convex functions (and weakly-convex functions [17] more broadly) the problem of computing proximal points with sufficiently high regularization is strongly convex and our estimator applies. Such non-convex proximal points play an important role in non-convex optimization [17] with applications in deep learning [see, e.g., 49]. Applying the optimum-estimator technique in non-convex optimization is therefore a viable direction for future work.

## 1.6 Notation

We let $\mathbb{B}_R(x) = \{y \in \mathbb{R}^d : \|y - x\| \le R\}$ denote the ball of radius $R$ around $x$, where $\|\cdot\|$ is the Euclidean norm throughout. We write $\mathsf{Proj}_{\mathcal{S}}$ for the Euclidean projection to $\mathcal{S}$. We write $\mathbb{1}_{\{A\}}$ for the indicator of event $A$, i.e., $\mathbb{1}_{\{A\}} = 1$ when $A$ holds and 0 otherwise. Throughout the paper, $\hat{\nabla} f$ denotes a (stochastic) subgradient estimator for the function $f$, and $\mathcal{X} \subset \mathbb{R}^d$ denotes the optimization domain, which we always assume is closed and convex. We use $\mathsf{P}_{f,\lambda}$ to denote the proximal operator (2) and $f_\lambda$ to denote the Moreau envelope (3) associated with function $f$ and regularization parameter $\lambda$. Finally, we use $\mathcal{N}_f$ and $\mathcal{N}_{\hat{\nabla} f}$ to denote function and subgradient estimator evaluation complexity, respectively.

## 2 A multilevel Monte-Carlo optimum estimator

In this section, we construct a low-bias estimator for the minimizer of any strongly convex function $F : \mathcal{X} \to \mathbb{R}$. This estimator is the key component of our algorithms in the subsequent sections, which use it to approximate proximal points and Moreau envelope gradients. We assume that $F$ is of the form $F = f + \psi$, where the function $\psi$ is "simple" and that $f$ satisfies the following.

**Assumption 1.** *The function* $f : \mathcal{X} \to \mathbb{R}$ *is convex (with closed and convex domain $\mathcal{X}$) and is accessible via an unbiased subgradient estimator $\hat{\nabla} f$ which satisfies $\mathbb{E}\|\hat{\nabla} f(x)\|^2 \le G^2$ for all $x$.*

Our applications only use $\psi$ of the form $\psi(x) = \frac{\lambda}{2}\|x - x'\|^2$ but our estimator applies more broadly to cases where $\operatorname{argmin}_x \left\{ \langle v, x \rangle + \psi(x) + \frac{1}{2\eta}\|x - y\|^2 \right\}$ is easy to compute for all $v$ and $y$.

### 2.1 ODC algorithms

Our estimator can use, in a black-box fashion, any method for minimizing $F$ with sufficiently fast convergence to $x_\star = \operatorname{argmin}_{x \in \mathcal{X}} F(x)$. We highlight the required convergence property as follows.

| **Algorithm 1:** OPTEST$(\hat{\nabla} f, \psi, \mu, \delta, \sigma^2, \mathcal{X})$ | **Algorithm 2:** MORGRADEST$(\hat{\nabla} f, y, \lambda, \delta, \sigma^2, \mathcal{X})$ |
|---|---|
| $\triangleright \delta, \sigma^2$ are required bias and square error | $\triangleright \delta, \sigma^2$ are required bias and square error |
| $\triangleright c$ is the ODC algorithm constant | $\triangleright \lambda$ is the regularization level |
| $T_{\max} = \left\lceil \frac{4cG^2}{\mu^2 \min\{\delta^2, \frac{1}{2}\sigma^2\}} \right\rceil$ | $\triangleright y$ is the point at which to estimate $\nabla f_\lambda(y)$ |
| $N = \left\lceil \frac{32cG^2 \log(T_{\max})}{\mu^2 \sigma^2} \right\rceil$ | $\psi_\lambda(x) = \frac{\lambda}{2}\|x - y\|^2$ |
| **for** $i = 1, \ldots, N$ **do** | $\hat{x}_\star = \text{OPTEST}\left(\hat{\nabla} f, \psi_\lambda, \lambda, \frac{\delta}{\lambda}, \frac{\sigma^2}{\lambda^2}, \mathcal{X}\right)$ |
| $\quad \hat{x}_\star^{(i)} = $ a draw of the estimator (1) | **return** $\lambda(y - \hat{x}_\star)$ |
| **return** $\frac{1}{N}\sum_{i\in[N]} \hat{x}_\star^{(i)}$ | |

**Definition 1.** *An* optimal-distance-convergence *algorithm* ODC *takes as input $\hat{\nabla} f$ satisfying Assumption 1, a simple function $\psi$ and a budget $T \geq 1$. If $F = f + \psi$ is $\mu$-strongly convex with minimizer $x_\star$, the algorithm's output $x = \text{ODC}(\hat{\nabla} f, \psi, T)$ requires at most $T$ evaluations of $\hat{\nabla} f$ to compute and satisfies $\mathbb{E}\|x - x_\star\|^2 \leq c\frac{G^2}{\mu^2 T}$ for some constant $c > 0$.*

Standard lower bound constructions imply that the $O(\frac{G^2}{\mu^2 T})$ squared distance convergence rate is indeed optimal; see Appendix A.4 for additional discussion. Conversely, ODC algorithms are readily available in the literature [44, 28] since any point $x$ satisfying $\mathbb{E}F(x) - F(x_\star) = O(\frac{G^2}{\mu T})$ (the optimal rate of convergence in strongly convex, Lipschitz optimization) also satisfies $\mathbb{E}\|x - x_\star\|^2 \leq O(\frac{G^2}{\mu^2 T})$ by due to the strong convexity of $F$. We provide a concrete ODC algorithm consisting of a generalization of epoch SGD [28], which allows us to optimize over the composite objective $F = f + \psi$ instead of only $f$ as in the prior study of epoch SGD.

**Lemma 1.** EPOCHSGD *(Algorithm 8 in Appendix B.1) is an ODC algorithm with constant $c = 32$.*

## 2.2 Constructing an optimum estimator

To turn any ODC algorithm into a low-bias, low-cost and near-constant variance optimum estimator, we use the multilevel Monte Carlo (MLMC) technique of Blanchet and Glynn [8]. Given a problem instance $\hat{\nabla} f, \psi$, an algorithm ODC and a cutoff parameter $T_{\max} \in \mathbb{N}$, our estimator $\hat{x}_\star$ is:

Draw $J \sim \text{Geom}\left(\frac{1}{2}\right) \in \mathbb{N}$ and, writing $x_j := \text{ODC}(\hat{\nabla} f, \psi, 2^j)$, compute

$$\hat{x}_\star = x_0 + \begin{cases} 2^J(x_J - x_{J-1}) & 2^J \leq T_{\max} \\ 0 & \text{otherwise.} \end{cases} \tag{1}$$

We note that for certain ODC algorithms it is possible to extract $x_0, x_{J-1}$ from the intermediate steps of computing $x_J$, so that we only need to invoke ODC once. This is particularly simple to do for EPOCHSGD, as we explain in Appendix B.1. The key properties of our estimator are as follows.

**Proposition 1.** *Let $f$ and $\hat{\nabla} f$ satisfy Assumption 1, $F = f + \psi$ be $\mu$-strongly convex with minimizer $x_\star$ and $T_{\max} \in \mathbb{N}$. For any ODC algorithm with constant $c$, the estimator (1) has bias $\|\mathbb{E}\hat{x}_\star - x_\star\| \leq \sqrt{2c}\frac{G}{\mu\sqrt{T_{\max}}}$ and variance $\mathbb{E}\|\hat{x}_\star - \mathbb{E}\hat{x}_\star\|^2 \leq 16c\frac{G^2}{\mu^2} \log_2(T_{\max})$. Moreover, the expected number of $\hat{\nabla} f$ evaluations required to compute $\hat{x}_\star$ is $O(\log T_{\max})$.*

*Proof.* Let $j_{\max} = \max\{j \in \mathbb{N} \mid 2^j \leq T_{\max}\} = \lfloor \log_2 T_{\max} \rfloor$. The expectation of $\hat{x}_\star$ is

$$\mathbb{E}\hat{x}_\star = \mathbb{E}x_0 + \sum_{j=1}^{j_{\max}} \mathbb{P}(J = j)2^j(\mathbb{E}x_j - \mathbb{E}x_{j-1}) = \mathbb{E}x_{j_{\max}},$$

where the second equality follows from $\mathbb{P}(J = j) = 2^{-j}$ and the sum telescoping. Noting that $x_{j_{\max}} = \text{ODC}(\hat{\nabla} f, \psi, T)$ for $T = 2^{j_{\max}} \geq T_{\max}/2$, we have that

$$\|\mathbb{E}x_{j_{\max}} - x_\star\| \leq \sqrt{\mathbb{E}\|x_{j_{\max}} - x_\star\|^2} \leq \sqrt{c}\frac{G}{\mu\sqrt{T_{\max}/2}}$$

by Definition 1. To bound the variance we use $\|a+b\|^2 \le 2\|a\|^2 + 2\|b\|^2$ and note that

$$\mathbb{E}\|\hat{x}_\star - \mathbb{E}\hat{x}_\star\|^2 \le \mathbb{E}\|\hat{x}_\star - x_\star\|^2 \le 2\mathbb{E}\|\hat{x}_\star - x_0\|^2 + 2\mathbb{E}\|x_0 - x_\star\|^2.$$

The ODC property implies that $\mathbb{E}\|x_0 - x_\star\|^2 \le cG^2/\mu^2$. For the term $\mathbb{E}\|\hat{x}_\star - x_0\|^2$ we have

$$\mathbb{E}\|\hat{x}_\star - x_0\|^2 = \sum_{j=1}^{j_{\max}} \mathbb{P}(J=j)2^{2j}\mathbb{E}\|x_j - x_{j-1}\|^2 = \sum_{j=1}^{j_{\max}} 2^j \mathbb{E}\|x_j - x_{j-1}\|^2, \text{ and}$$

$$\mathbb{E}\|x_j - x_{j-1}\|^2 \le 2\mathbb{E}\|x_j - x_\star\|^2 + 2\mathbb{E}\|x_{j-1} - x_\star\|^2 \le 6c\frac{G^2}{\mu^2}2^{-j}.$$

Substituting, we get $\mathbb{E}\|\hat{x}_\star - x_0\|^2 \le 6c\frac{G^2}{\mu^2}j_{\max}$ and $\mathbb{E}\|\hat{x}_\star - \mathbb{E}\hat{x}_\star\|^2 \le 16c\frac{G^2}{\mu^2}\log_2(T_{\max})$. Finally, the expected number of $\hat{\nabla}f$ evaluations is $1 + \sum_{j=1}^{j_{\max}} \mathbb{P}(J=j)(2^j + 2^{j-1}) = O(j_{\max})$. $\qquad\square$

The function OPTEST in Algorithm 1 computes an estimate of $x_\star$ with and desired bias $\delta$ and square error $\sigma^2$ by averaging independent draws of the MLMC estimator (1). The following guarantees are immediate from Proposition 1; see Appendix B.2 for a short proof.

**Theorem 1.** *Let $f$ and $\hat{\nabla}f$ satisfy Assumption 1, $F = f + \psi$ be $\mu$-strongly convex with minimizer $x_\star \in \mathcal{X}$, and $\delta, \sigma > 0$. The function OPTEST$(\hat{\nabla}f, \psi, \mu, \delta, \sigma^2, \mathcal{X})$ outputs $\hat{x}_\star$ satisfying*

$$\|\mathbb{E}\hat{x}_\star - x_\star\| \le \delta \quad and \quad \mathbb{E}\|\hat{x}_\star - x_\star\|^2 \le \sigma^2$$

*using $\mathcal{N}_{\hat{\nabla}f}$ stochastic gradient computations, where*

$$\mathbb{E}\mathcal{N}_{\hat{\nabla}f} = O\left(\frac{G^2}{\mu^2\sigma^2}\log^2\left(\frac{G}{\mu\min\{\delta,\sigma\}}\right) + \log\left(\frac{G}{\mu\min\{\delta,\sigma\}}\right)\right).$$

### 2.3 Estimating proximal points and Moreau envelope gradients

The proximal point of function $f : \mathcal{X} \to \mathbb{R}$ with regularization level $\lambda$ at point $y$ is

$$\mathsf{P}_{f,\lambda}(y) := \operatorname*{argmin}_{x \in \mathcal{X}}\left\{f(x) + \tfrac{\lambda}{2}\|x-y\|^2\right\}. \tag{2}$$

When $f$ satisfies Assumption 1, we may use OPTEST (with $\psi(x) = \frac{\lambda}{2}\|x-y\|^2$ and $\mu = \lambda$) to obtain a reduced-bias proximal point estimator. The proximal point $\mathsf{P}_{f,\lambda}(y)$ is closely related to the Moreau envelope

$$f_\lambda(y) := \min_{x \in \mathcal{X}}\left\{f(x) + \tfrac{\lambda}{2}\|x-y\|^2\right\} \tag{3}$$

via the relationship $\nabla f_\lambda(y) = \lambda(y - \mathsf{P}_{f,\lambda}(y))$ (see Appendix B.3). Therefore, we can use our optimum estimator to turn $\widetilde{O}(1)$ calls to $\hat{\nabla}f$ into a nearly unbiased estimator for $\nabla f_\lambda$. We formulate this as:

**Corollary 2.** *Let $f$ and $\hat{\nabla}f$ satisfy Assumption 1, let $y \in \mathcal{X}$ and let $\lambda, \sigma, \delta > 0$. The function MORGRADEST$(\hat{\nabla}f, \lambda, y, \delta, \sigma^2, \mathcal{X})$ outputs $\hat{\nabla}f_\lambda(y)$ satisfying $\|\mathbb{E}\hat{\nabla}f_\lambda(y) - \nabla f_\lambda(y)\| \le \delta$ and $\mathbb{E}\|\hat{\nabla}f_\lambda(y) - \nabla f_\lambda(y)\|^2 \le \sigma^2$ and has complexity $\mathbb{E}\mathcal{N}_{\hat{\nabla}f} = O\left(\frac{G^2}{\sigma^2}\log^2\left(\frac{G}{\min\{\delta,\sigma\}}\right) + \log\left(\frac{G}{\min\{\delta,\sigma\}}\right)\right)$.*

## 3 Projection-efficient convex optimization

In this section, we combine the bias-reduced Moreau envelope gradient estimator with a standard accelerated gradient method to recover the result of Thekumparampil et al. [51]. We consider the problem of minimizing a function $f$ satisfying Assumption 1 over the domain $\mathbb{B}_R(0)$ subject to the constraint $x \in \mathcal{X}$, where $\mathcal{X} \subset \mathbb{B}_R(0)$ is a complicated convex set that we can only access via (expensive) projections of the form $\mathsf{Proj}_{\mathcal{X}}(x) := \operatorname*{argmin}_{y \in \mathcal{X}}\|y - x\|$. We further assume that an initial point $x_0 \in \mathcal{X}$ satisfies $\|x_0 - x_\star\| \le D$.

Algorithm 3 applies a variant of Nesterov's accelerated gradient descent method (related to [2, 1]) on the ($\lambda$-smooth) Moreau envelope $f_\lambda$ defined in eq. (3). Since computing the Moreau envelope

---

**Algorithm 3:** Stochastic accelerated gradient descent on the Moreau envelope

---

**Input:** A gradient estimator $\hat{\nabla} f$ satisfying Assumption 1 in $\mathbb{B}_R(0)$, projection oracle $\mathsf{Proj}_{\mathcal{X}}$, and initial point $x_0 = v_0$ with $\|x_0 - x_\star\| \le D$.

**Parameters:** Iteration budget $T$ , Moreau regularization $\lambda$, approximation parameters $\delta_k, \sigma_k^2$

1 **for** $k = 1, \cdots, T$ **do**

2     $y_{k-1} = \frac{k-1}{k+1} x_k + \frac{2}{k+1} v_{k-1}$

3     $g_k = \text{MORGRADEST}(\hat{\nabla} f, y_{k-1}, \lambda, \delta_k, \sigma_k^2, \mathbb{B}_R(0))$

4     $x_k = \mathsf{Proj}_{\mathcal{X}} \left( y_{k-1} - \frac{1}{3\lambda} g_k \right)$

5     $v_k = \mathsf{Proj}_{\mathbb{B}_R(0)} \left( v_{k-1} - \frac{k}{6\lambda} g_k \right)$

6 **return** $x_T$

---

does not involve projection to $\mathcal{X}$, for sufficiently accurate approximation of $\nabla f_\lambda$ we require only $T = O(\sqrt{\lambda D^2/\epsilon})$ projections to $\mathcal{X}$ for finding an $O(\epsilon)$-suboptimal point of $f_\lambda$ constrained to $\mathcal{X}$. For that point to be also $\epsilon$-suboptimal for $f$ itself, we must choose $\lambda$ of the order of $G^2/\epsilon$, so that the number of projections is $O(GD/\epsilon)$.

As noted in [51] computing $\nabla f_\lambda$ to accuracy $O(\epsilon/R)$ is sufficient for the above guarantee to hold, but doing so using a stochastic gradient method requires $O((GD/\epsilon)^2)$ evaluations of $\hat{\nabla} f$ per iteration, and $O((GD/\epsilon)^3)$ evaluations in total. To improve this, we employ Algorithm 2 to compute nearly-unbiased estimates for $\nabla f_\lambda$ and bound the error incurred by their variance. Our result matches the gradient sliding-based technique of Thekumparampil et al. [51] up to polylogarithmic factors while retaining the conceptual simplicity of directly applying AGD on the Moreau envelope. We formally state the guarantees of our method below, and provide a self-contained proof in Appendix C.

**Theorem 3.** *Let $f : \mathbb{B}_R(0) \to \mathbb{R}$ and $\hat{\nabla} f$ satisfy Assumption 1. Let $\mathcal{X} \subseteq \mathbb{B}_R(0)$ be a convex set admitting a projection oracle $\mathsf{Proj}_{\mathcal{X}}$. Let $x_0 \in \mathcal{X}$ be an initial point with $\|x - x_\star\| \le D$ for some $x_\star \in \mathcal{X}$. With $\lambda = \frac{2G^2}{\epsilon}$, $\delta_k = \frac{\epsilon}{8R}$, $\sigma_k^2 = \frac{2\epsilon\lambda}{k+1}$, and $T = \frac{7GD}{\epsilon}$ Algorithm 3 computes $x \in \mathcal{X}$ with $\mathbb{E}[f(x)] \le f(x_\star) + \epsilon$ with complexity $\mathbb{E} \mathcal{N}_{\hat{\nabla} f} = O\left( \frac{G^2 D^2}{\epsilon^2} \log^2 \left( \frac{GR}{\epsilon} \right) \right)$ and $O\left( \frac{GD}{\epsilon} \right)$ calls to $\mathsf{Proj}_{\mathcal{X}}$.*

## 4 Accelerated proximal methods and minimizing the maximal loss

In this section we apply our estimator in an accelerated proximal point method and use it to obtain an optimal rate for minimizing the maximum of $N$ convex functions (up to logarithmic factors).

### 4.1 Accelerated proximal point method via Moreau gradient estimation

Algorithm 4 is an Monteiro-Svaiter-type [38, 12] accelerated proximal point method [36, 24] that leverages our reduced-bias Moreau envelope gradient estimator. To explain the method, we contrast it with stochastic AGD on the Moreau envelope (Algorithm 3). First and foremost, Algorithm 3 provides a suboptimality bound on the Moreau envelope $f_\lambda$ (which for small $\lambda$ is far from $f$) while Algorithm 4 minimizes $f$ itself.

Second, while Algorithm 3 uses a fixed regularization parameter $\lambda$, Algorithm 4 handles an arbitrary sequence $\{\lambda_k\}$ given by a black-box function NEXTLAMBDA. To facilitate our application of the method to minimizing the maximal loss—where gradient estimation is only tractable in small Euclidean balls around a reference point—we include an optional parameter $r$ such that the proximal point movement bound $\|\mathsf{P}_{f, \lambda_{k+1}}(y_k) - y_k\| \le r$ holds for all $k$. However, most of our analysis of Algorithm 4 does not require this parameter (i.e., holding for $r = \infty$), making it potentially applicable to other settings that use accelerated proximal point methods [11, 38, 50].

The third and final notable difference between Algorithms 3 and 4 is the method of updating the $x_k$ iteration sequence. While a projected stochastic gradient descent step suffices for Algorithm 3, here we require a more direct approximation of function value decrease attained by the exact proximal mapping $\mathsf{P}_{f, \lambda}$ (see eq. (2)). For a given accuracy $\varphi$, we define the $\varphi$-approximate proximal mapping

$$\widetilde{\mathsf{P}}_{f, \lambda}^{\varphi}(y) := \text{ any } x \in \mathcal{X} \text{ such that } \mathbb{E} F(x) \le F(\mathsf{P}_{f, \lambda}(y)) + \varphi \text{ for } F(z) := f(z) + \frac{\lambda}{2} \|z - y\|^2. \quad (4)$$

**Algorithm 4:** Stochastic accelerated proximal point method

---

**Input:** Gradient estimator $\hat{\nabla} f$, function NEXTLAMBDA, initialization $x_0 = v_0$ and $A_0 \geq 0$.
**Parameters:** Approximation parameters $\{\varphi_k, \delta_k, \sigma_k\}$, stopping parameters $A_{\max}$ and $K_{\max}$,
  optional movement bound $r > 0$.

1 **for** $k = 0, 1, \ldots$ **do**
2    $\lambda_{k+1} = \text{NEXTLAMBDA}(x_k, v_k, A_k)$ ▷ guaranteeing that $\|\mathsf{P}_{f,\lambda_{k+1}}(y_k) - y_k\| \leq r$
3    $a_{k+1} = \frac{1}{2\lambda_{k+1}}\sqrt{1 + 4\lambda_{k+1}A_k}$ and $A_{k+1} = A_k + a_{k+1}$ and $\mathcal{X}_k = \mathcal{X} \cap \mathbb{B}_r(y_k)$
4    $y_k = \frac{A_k}{A_{k+1}}x_k + \frac{a_{k+1}}{A_{k+1}}v_k$ and $x_{k+1} = \widetilde{\mathsf{P}}^{\varphi_{k+1}}_{f,\lambda_{k+1}}(y_k)$ ▷ defined in eq. (4)
5    $g_{k+1} = \text{MORGRADEST}(\hat{\nabla} f, y_k, \lambda_{k+1}, \delta_k, \sigma_k^2, \mathcal{X}_k)$ and $v_{k+1} = \text{Proj}_{\mathcal{X}}\left(v_k - \frac{1}{2}a_{k+1}g_{k+1}\right)$
6    **if** $A_{k+1} \geq A_{\max}$ **or** $k + 1 = K_{\max}$ **then return** $x_{k+1}$

---

Note that $\widetilde{\mathsf{P}}^0_{f,\lambda} = \mathsf{P}_{f,\lambda}$ and that for $\varphi > 0$ we can compute $\widetilde{\mathsf{P}}^{\varphi}_{f,\lambda}$ with an appropriate SGD variant (such as EPOCHSGD) using $O(G^2/(\lambda\varphi))$ evaluations of $\hat{\nabla} f$.

With the differences between the algorithms explained, we emphasize their key similarity: both algorithms update the $v_k$ sequence using our bias reduction method MORGRADEST (Algorithm 2), which holds the key to their efficiency. The following proposition shows that Algorithm 4 has the same bound on $K_{\max}$ as an exact accelerated proximal point method [12], while requiring at most $\widetilde{O}(G^2R^2\epsilon^{-2})$ stochastic gradient evaluations; see proof in Appendix D.1.

**Proposition 2.** *Let $f : \mathcal{X} \to \mathbb{R}$ and $\hat{\nabla} f$ satisfy Assumption 1, and let $\mathcal{X} \subseteq \mathbb{B}_R(x_0)$. For a target accuracy $\epsilon \leq GR$ let $\varphi_{k+1} = \frac{\epsilon}{60\lambda_{k+1}a_{k+1}}$, $\delta_{k+1} = \frac{\epsilon}{120R}$, $\sigma^2_{k+1} = \frac{\epsilon}{60a_{k+1}}$, $A_0 = \frac{R}{G}$, and $A_{\max} = \frac{9R^2}{\epsilon}$. If $\lambda_k \geq \lambda_{\min} \geq \frac{1}{A_{\max}} = \Omega(\frac{\epsilon}{R^2})$ for all $k \leq K_{\max}$, then lines 4 and 5 of Algorithm 4 have total complexity $\mathbb{E}\mathcal{N}_{\hat{\nabla} f} = O\left(K_{\max}\log\frac{GR}{\epsilon} + \frac{G^2R^2}{\epsilon^2}\log^2\frac{GR}{\epsilon}\right)$. If in addition $\|\mathsf{P}_{f,\lambda_k}(y_{k-1}) - y_{k-1}\| \geq 3r/4$ whenever $\lambda_k \geq 2\lambda_{\min}$ then for $K_{\max} = O\left(\left(\frac{R}{r}\right)^{2/3}\log\left(\frac{GR}{\epsilon}\right) + \sqrt{\frac{\lambda_{\min}R^2}{\epsilon}}\right)$, the algorithm's output $x_K$ satisfies $f(x_K) - f(x_\star) \leq \epsilon$ with probability at least $\frac{2}{3}$.*

### 4.2 Minimizing the maximal loss

We now consider objectives of the form $f_{\max}(x) := \max_{i \in [N]} f_{(i)}(x)$ where each function $f_{(i)} : \mathcal{X} \to \mathbb{R}$ is convex and $G$-Lipschitz. Our approach to minimizing $f_{\max}$ largely follows Carmon et al. [13]; the main difference is that we approximate proximal steps via Algorithm 4 and our reduced-bias bias estimator. The first step of the approach is to replace $f_{\max}$ with the "softmax" function, defined for a given target accuracy $\epsilon$ as

$$f_{\text{smax}}(x) := \epsilon' \log\left(\sum_{i \leq N} \exp\left(f_{(i)}(x)/\epsilon'\right)\right), \text{ where } \epsilon' := \frac{\epsilon}{2\log N}.$$

Since $f_{\text{smax}}(x) - f_{\max}(x) \in [0, \frac{\epsilon}{2}]$, any $\frac{\epsilon}{2}$-accurate solution of $f_{\text{smax}}$ is $\epsilon$-accurate for $f_{\max}$.

The second step is to develop an efficient gradient estimator for $f_{\text{smax}}$; this is non-trivial because $f_{\text{smax}}$ is not a finite sum or expectation. In [13] this is addressed via an "exponentiated softmax" trick; we develop an alternative, rejection sampling-based approach that fits Algorithm 4 more directly (see Algorithm 9). To produce an unbiased estimate for $\nabla f_{\text{smax}}(x)$ for $x$ in a ball of radius $r_\epsilon = \epsilon'/G$ we require a single $\nabla f_{(i)}(x)$ evaluation (for some $i$), $O(1)$ evaluations of $f_{(i)}(x)$ in expectation, and evaluation of the $N$ functions $f_{(1)}(y), \ldots, f_{(N)}(y)$ for pre-processing. Plugging this estimator into Algorithm 4 with $r = r_\epsilon$, the total pre-processing overhead of lines 4 and 5 is $O(K_{\max}N)$.

The final step is to find a function NEXTLAMBDA such that $\|\mathsf{P}_{f_{\text{smax}},\lambda_{t+1}}(y_k) - y_k\| \leq r_\epsilon$ for all $k$ (enabling gradient estimation), and $\|\mathsf{P}_{f_{\text{smax}},\lambda_{t+1}}(y_k) - y_k\| \geq \frac{3}{4}r_\epsilon$ when $\lambda_{k+1} > 2\lambda_{\min}$ (allowing us to bound $K_{\max}$ with Proposition 2). Here we use the bisection subroutine from [13] as is (see Algorithm 10). By judiciously choosing $\lambda_{\min}$—an improvement over the analysis in [13]—we obtain the following complexity guarantee on $\mathcal{N}_{f_{(i)}}$ and $\mathcal{N}_{\partial f_{(i)}}$, the total numbers of individual function and subgradient evaluations, respectively. (See proof Appendix D.2).

**Theorem 4.** *Let $f_{(1)}, \ldots, f_{(N)} : \mathcal{X} \to \mathbb{R}$ be convex and $G$-Lipschitz and let $\mathcal{X} \subseteq \mathbb{B}_R(x_0)$. For any $\epsilon < \frac{1}{2} GR/\log N$, Algorithm 4 (with $\widetilde{\mathsf{P}}^{\varphi}_{f_{\mathrm{smax}}, \lambda}$ implemented in Algorithm 8, $\hat{\nabla} f_{\mathrm{smax}}$ given by Algorithm 9 and NEXTLAMBDA given by Algorithm 10 with $\lambda_{\min} = \widetilde{O}(\epsilon/(r_\epsilon^{4/3} R^{2/3}))$) outputs $x \in \mathcal{X}$ that with probability at least $\frac{1}{2}$ is $\epsilon$-suboptimal for $f_{\max}(x) = \max_{i \in [N]} f_{(i)}(x)$ and has complexity*

$$\mathbb{E}\mathcal{N}_{f_{(i)}} = O\left( \left[ N \left( \frac{GR \log N}{\epsilon} \right)^{2/3} + \left( \frac{GR}{\epsilon} \right)^2 \right] \log^2 \frac{GR}{\epsilon} \right) \text{ and } E\mathcal{N}_{\partial f_{(i)}} = O\left( \left( \frac{GR}{\epsilon} \right)^2 \log^2 \frac{GR}{\epsilon} \right).$$

The rate given by Theorem 4 matches (up to logarithmic factors) the lower bound $\Omega(N(GR/\epsilon)^2 + (GR/\epsilon)^2)$ shown in [13] and is therefore near-optimal.

## 5 Gradient-efficient composite optimization

Consider the problem of finding a minimizer of the following convex composite optimization problem

$$\underset{x \in \mathcal{X}}{\mathrm{minimize}} \; \Psi(x) := \Lambda(x) + f(x) \text{ where } \Lambda \text{ is } L\text{-smooth and } f \text{ satisfies Assumption 1}, \quad (5)$$

given $x_0$ such that $\|x_0 - x_\star\| \leq R$ for some $x_\star \in \operatorname{argmin}_{x \in \mathcal{X}} \Psi(x)$. Lan [34] developed a method called "gradient sliding" that finds an $\epsilon$-accurate solution to (5) with complexity $\mathcal{N}_{\nabla \Lambda} = O(\sqrt{LR^2/\epsilon})$ evaluations of $\nabla \Lambda(x)$ and $\mathcal{N}_{\hat{\nabla} f} = O((GR/\epsilon)^2)$ evaluations of $\hat{\nabla} f(x)$, which are optimal even for each component separately.[6]

In this section, we provide an alternative algorithm that matches the complexity of gradient up to logarithmic factors and is conceptually simple. Our approach, Algorithm 5, is essentially composite AGD [41], where at the $k$th iteration we compute a proximal point (2) with respect to a partial linearization of $\Psi$ around $y_k$. In particular, letting $\bar{\Lambda}_k(v) := \Lambda(y_k) + \langle \nabla \Lambda(y_k), v - y_k \rangle$ and $\beta_k = \frac{2L}{k}$, we approximate $\mathsf{P}_{\bar{\Lambda}+f, \beta_k}(v_{k-1})$. Similar to Algorithm 4, Algorithm 5 computes two types of approximations: one is an $\epsilon_k$-approximate proximal point $\widetilde{\mathsf{P}}^{\epsilon_k}_{\bar{\Lambda}+f, \beta_k}(v_{k-1})$ as per its definition (4), while the other is our bias-reduced optimum estimator from Algorithm 1. We note, however, that unlike Algorithm 4 which approximates the $x_k$ update, here we approximate $v_k$, the "mirror descent" update.

Below we state the formal guarantees for Algorithm 5; we defer its proof to Appendix E.

---

**Algorithm 5:** Stochastic composite accelerated gradient descent

**Input:** A problem of the form (5) with $\Lambda, f, \nabla \Lambda, \hat{\nabla} f$.
**Parameters:** Step size parameters $\beta_k = \frac{2L}{k}$ and $\gamma_k = \frac{2}{k+1}$, iteration number $N$, approximation parameters $\{\epsilon_k, \delta_k, \sigma_k^2\}$ and $x_0 = v_0$ satisfying $\|x_0 - x_\star\| \leq R$.

1 **for** $k = 1, 2, \cdots, N$ **do**
2     $y_k = (1 - \gamma_k)x_{k-1} + \gamma_k \mathsf{Proj}_{\mathcal{X}}(v_{k-1})$
3     $\bar{v}_k = \widetilde{\mathsf{P}}^{\epsilon_k}_{\bar{\Lambda}+f, \beta_k}(v_{k-1})$ for $\bar{\Lambda}_k(v) := \Lambda(y_k) + \langle \nabla \Lambda(y_k), v - y_k \rangle$
4     $v_k = \mathrm{OPTEST}(\hat{\nabla} f, \psi_k, \beta_k, \delta_k, \sigma_k^2, \mathbb{B}_R(v_0) \cap \mathcal{X})$ for $\psi_k(z) = \frac{\beta_k}{2}\|z - v_{k-1}\|^2 + \bar{\Lambda}_k(z)$
5     $x_k = (1 - \gamma_k)x_{k-1} + \gamma_k \bar{v}_k$
6 **return** $x_N$

---

**Theorem 5.** *Given problem (5) with solution $x_\star$, a point $x_0$ such that $\|x_0 - x_\star\| \leq R$ and target accuracy $\epsilon > 0$, Algorithm 5 with $\epsilon_k = LR/2kN$, $\delta_k = R/16N$, $\sigma_k^2 = R^2/4N$, and $N = \Theta(\sqrt{LR^2/\epsilon})$ finds an approximate solution $x$ satisfying $\mathbb{E}\Psi(x) \leq \Psi(x_\star) + \epsilon$ and has complexity*

$$\mathcal{N}_{\nabla \Lambda} = O\left( \sqrt{\frac{LR^2}{\epsilon}} \right) \text{ and } \mathbb{E}\mathcal{N}_{\hat{\nabla} f} = O\left( \left( \frac{GR}{\epsilon} \right)^2 \log^2 \frac{GR}{\epsilon} + \sqrt{\frac{LR^2}{\epsilon}} \log \left( \frac{GR}{\epsilon} \right) \right).$$

## 6 Efficient non-smooth private convex optimization

We conclude the paper with a potential application of our optimum estimator for differentially private stochastic convex optimization (DP-SCO). In this problem we are given $n$ i.i.d. sample

---

[6]The gradient sliding result holds under a relaxed Lipschitz assumption [see 34, eq. (1.2)]. It is straightforward to extend EPOCHSGD, and hence all of our results, to that assumption as well.

$s_i \sim P$ taking values in a set $\mathbb{S}$, and out objective is to privately minimize the population average $f(x) = \mathbb{E}_{S \sim P}[\hat{f}(x; S)]$, where $\hat{f} : \mathcal{X} \times \mathbb{S} \to \mathbb{R}$, is convex in the first argument and $\mathcal{X} \subset \mathbb{R}^d$ is a convex set, and $\mathbb{S}$ is a population of data points. That is, We wish to find $\hat{x} \in \mathcal{X}$ with small excess loss $f(\hat{x}) - \min_{x \in \mathcal{X}} f(x)$ while preserving differential privacy.

**Definition 2** ([22]). *A randomized algorithm $\mathcal{A}$ is $(\alpha, \beta)$-differentially private ($(\alpha, \beta)$-DP) if, for all datasets $\mathcal{S}, \mathcal{S}' \in \mathbb{S}^n$ that differ in a single data element and for every event $\mathcal{O}$ in the output space of $\mathcal{A}$, we have $P[\mathcal{A}(\mathcal{S}) \in \mathcal{O}] \le e^\alpha P[\mathcal{A}(\mathcal{S}') \in \mathcal{O}] + \beta$.*

DP-SCO has received increased attention over the past few years. Bassily et al. [5] developed (inefficient) algorithms that attain the optimal excess loss $1/\sqrt{n} + \sqrt{d \log(1/\beta)}/n\alpha$. When each function is $O(\sqrt{n})$ smooth, Feldman et al. [23] gave algorithms with optimal excess loss and $O(n)$ gradient query complexity. In the non-smooth setting, however, their algorithms require $O(n^2)$ subgradients. Subsequently, Asi et al. [3] and Kulkarni et al. [32] developed more efficient algorithms for non-smooth functions which need $O(\min(n^2/\sqrt{d}, n^{5/4}d^{1/8}, n^{3/2}/d^{1/8}))$ subgradients which is $O(n^{11/8})$ for the high-dimensional setting $d = n$. Whether a linear gradient complexity is achievable for DP-SCO in the non-smooth setting is still open.

In this section, we develop an efficient algorithm for non-smooth DP-SCO that queries $\widetilde{O}(n)$ subgradients conditional on the existence of an optimum estimator with the following properties.

**Definition 3.** *Let $F = f + \psi$ be $\mu$-strongly convex with minimizer $x_\star$ and $f$ is $G$-Lipschitz. For $\delta > 0$, we say that $\mathcal{O}_\delta$ is efficient bounded low-bias estimator if it returns $\hat{x}_\star = \mathcal{O}_\delta(F)$ such that $\|\mathbb{E}[\hat{x}_\star - x_\star]\|^2 \le \delta^2$, $\|\hat{x}_\star - x_\star\|^2 \le C_1 G^2 \log(G/\mu\delta)/\mu^2$, and the expected number of gradient queries is $C_2 \log(G/\mu\delta)$.*

Comparing to our MLMC estimator (1) and Proposition 1, we note that the only place our current estimator falls short of satisfying Definition 3 is the probability 1 bound on $\|\hat{x}_\star - x_\star\|^2$, which for (1) holds only in expectation. Indeed, for our estimator, $\|\hat{x}_\star - x_\star\|$ can be as large as $O(G/(\mu\delta))$, meaning that it is heavy-tailed.

It is not clear whether an EBBOE as defined above exists. Nevertheless, assuming access to such estimator, Algorithm 6 solves the DP-SCO problem with a near-linear amount of gradient computations. The algorithm builds on the recent localization-based optimization methods in [23] which iteratively solve regularized minimization problems.

---

**Algorithm 6:** Differentially-private stochastic convex optimization via optimum estimation

---

**Input:** $(s_1, \ldots, s_n) \in \mathbb{S}^n$, domain $\mathcal{X} \subset \mathbb{B}_R(x_0)$, EBBOE $\mathcal{O}$ (satisfying Definition 3).

1 Set $k = \lceil \log n \rceil$, $B = 20(\log(\frac{1}{\beta}) + C_2 \log^2 n)$, $\bar{n} = \frac{n}{k}$, $\eta = \frac{R}{G} \min\left\{ \frac{1}{\sqrt{n}}, \frac{\alpha}{B \log(n)\sqrt{d \log(\frac{1}{\beta})}} \right\}$

2 **for** $i = 1, 2, \cdots, k$ **do**

3     Let $\eta_i = 2^{-4i}\eta$, $f_i(x) = \frac{1}{\bar{n}} \sum_{j=1+(k-1)\bar{n}}^{k\bar{n}} \hat{f}(x; s_j)$, $\psi_i(x) = \|x - x_{i-1}\|^2/(\eta_i \bar{n})$

4     Let $\tilde{x}_i = \frac{1}{\bar{n}} \sum_{j=1}^{\bar{n}} \mathcal{O}_{\delta_i}(F_i)$ with $F_i = f_i + \psi_i$, $\delta_i^2 = G^2 \eta_i^2 \bar{n}$

5     Set $x_i = \tilde{x}_i + \zeta_i$ where $\zeta_i \sim \mathsf{N}(0, \sigma_i^2 I_d)$ with $\sigma_i = 8B(\sqrt{C_1 \log n} + 2)\eta_i \sqrt{\log(2/\beta)}/\alpha_i$

6 **return** $x_k$

---

We average multiple draws of the (hypothetical) bounded optimum estimator to solve the regularized problems, and apply private mean estimation procedures to preserve privacy. We defer the proof of the following results Appendix F.

**Theorem 6** (conditional). *Given an efficient bounded low-bias estimator $\mathcal{O}_\delta$ satisfying Definition 3 for any $\delta > 0$, then for $\alpha \le \log(1/\beta)$, $\mathcal{X} \in \mathbb{B}_R(x_0)$, convex and $G$-Lipschitz $\hat{f}(x; s)$, Algorithm 6 is $(\alpha, \beta)$-DP, queries $\widetilde{O}(n)$ subgradients and has (hiding logarithmic factors in n) $\mathbb{E}[f(x_k) - \min_{x \in \mathcal{X}} f(x)] \le GR \cdot \widetilde{O}\left( \frac{1}{\sqrt{n}} + \frac{\sqrt{d \log^3(1/\beta)}}{n\alpha} \right)$.*

Theorem 6 provides a strong motivation for constructing bounded optimum estimators that satisfy Definition 3 . In Appendix F.3, we discuss the challenges in making our MLMC estimator bounded, as well as some directions to overcome them.

## Acknowledgments

HA was supported by ONR YIP N00014-19-2288 and the Stanford DAWN Consortium. YC was supported in part by Len Blavatnik and the Blavatnik Family foundation, and the Yandex Machine Learning Initiative for Machine Learning. YJ was supported by a Stanford Graduate Fellowship. AS was supported in part by a Microsoft Research Faculty Fellowship, NSF CAREER Award CCF-1844855, NSF Grant CCF-1955039, a PayPal research award, and a Sloan Research Fellowship.

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
