# Appendix

## A  Additional results and discussion

Here we provide additional discussion of three topics pertinent to our results: a zero-bias optimum estimator, the parallel depth of our estimator, turning expected complexity bounds into deterministic ones, and a justification for the adjective "optimal" in Definition 1 of the "optimal distance convergence" property. We recommend reading Sections 1 to 3 before the subsections below.

### A.1  Zero-bias optimum estimation given exact gradients

The main tool developed in this paper is an optimum estimator with bias $\delta$ whose expected query complexity is $O(\log(G/(\mu\delta)))$. In this section, we show how to obtain a completely unbiased optimum estimator when, in addition to a stochastic subgradient oracle, we assume access to a first-order oracle, i.e., one which outputs the functions exact value and subgradient at the query point.

To be concrete, assume that the domain is a ball of radius $R$ in $\mathbb{R}^d$, and that the objective $F : \mathbb{B}_R(x_0) \to \mathbb{R}$ is $\mu$-strongly-convex and of the form $F(x) = \frac{1}{n} \sum_{i \in [n]} \hat{F}(x; i)$, where each $\hat{F}(\cdot; i)$ is $G$-Lipschitz and given by a first-order oracle. In this case, we can compute an unbiased subgradient estimator with single oracle query by sampling $i \sim \mathsf{Unif}([n])$ and taking $\hat{\nabla} F \in \partial \hat{F}(x; i)$. Further, value and subgradient evaluations of $F$ can be implemented at $n$-times the cost by querying each each $F_i$. In this setting, we design an unbiased estimator of $x_\star = \operatorname{argmin}_{x \in \mathbb{B}_R(0)} F(x)$ with variance $O((G^2/\mu^2) \log(nd))$, expected query complexity $O(\log(nd))$ and expected runtime $O(d \log(nd))$.

To obtain this result, we leverage that first-order methods can compute the minimizer of a convex function with a number of queries and runtime that depends polynomially on dimension and logarithmically on regularity parameters and the desired accuracy. In fact, any polynomial bound suffices for our purposes and effect only constants factors in our expected complexity bounds. For concreteness, we use the classic ellipsoid method [55, 48, 31] whose complexity we describe in the following lemma. (We remark, however, that improved query complexities and runtimes are achievable; see [30] for the state-of-the-art).

**Lemma 2** (Ellipsoid method). *There is an algorithm,* $\mathrm{ELLIPSOID}(x_0, f, T)$*, which given* $x_0 \in \mathbb{R}^d$*, a first order oracle for $G$-Lipschitz, $\mu$-strongly-convex $f : \mathbb{B}_R(x_0) \to \mathbb{R}$, and query budget $T \geq 0$, runs in $O(d^2 T)$ time, makes at most $T$ queries, and outputs $\hat{x}_\star \in \mathbb{B}_R(x_0)$ with $\|\hat{x}_\star - x_\star\|_2^2 \leq (8G^2/\mu^2) \exp(-T/(2d^2))$ for $x_\star := \operatorname{argmin}_{x \in \mathbb{B}_R(x_0)} f(x)$.*

*Proof.* Since $f$ is $G$-Lipschitz for all $x \in \mathbb{B}_R(x_0)$ we have $|f(x) - f(x_0)| \leq GR$. Consequently, the ellipsoid method applied to $f(x) - f(x_0)$ can compute $\hat{x}_\star \in \mathbb{B}_R(x_0)$ with $f(\hat{x}_\star) - f(x_\star) \leq 2GR \exp(-T/(2d^2))$ with $O(T)$ queries and $O(d^2 T)$ time [see, e.g., 10, Theorem 2.4]. Since by strong convexity $\|\hat{x}_\star - x_\star\|_2^2 \leq \frac{2}{\mu}[f(\hat{x}_\star) - f(x_\star)]$ this implies that $\|\hat{x}_\star - x_\star\|_2^2 \leq (4GR/\mu) \exp(-T/(2d^2))$. Further, since $f$ is $G$-Lipschitz and $\mu$-strongly-convex we know that for all $y \in \mathbb{B}_R(x_0)$ we have $G\|y - x_\star\| \geq f(y) - f(x_\star) \geq \frac{\mu}{2}\|y - x_\star\|^2$ and since $\mathbb{B}_R(x_0)$ contain a point $y$ with $\|y - x_\star\| \geq R$ this implies $R \leq 2G/\mu$. Combining yields the result. $\square$

Combining the ellipsoid method with an ODC algorithm (see Definition 1) we obtain our unbiased optimum estimator, which we formally describe in Algorithm 7. The procedure is similar to the

---

**Algorithm 7:** Unbiased optimum estimator

**Input:** Initialization $x_0 \in \mathbb{R}^d$, first-order oracles for $\hat{F}(x; i)$ for all $i \in [n]$ and ODC algorithm ODC.

**1** Let $J_0 := \lceil 4 \log_2(14(nd^2 + d^4)) \rceil$

**2** For all $j > 1$ let $x_j := \begin{cases} \mathrm{ODC}(\nabla \hat{F}(\cdot; i), 0, 2^j) & \text{if } j \leq J_0 \\ \mathrm{ELLIPSOID}(x_0, F, \lceil 2^{j/2} \rceil) & \text{if } j > J_0 \end{cases}$

**3** Draw $J \sim \mathsf{Geom}\left(\frac{1}{2}\right)$

**4 return** $x_0 + 2^J(x_{J-1} - x_J)$ ▷ Only $x_0$, $x_{J-1}$, and $x_J$ are computed explicitly by the algorithm.

---

MLMC estimator (1), with the key difference that when (1) would output $x_0$, we instead apply the ellipsoid method. The following theorem establishes the performance of our algorithm.

**Theorem 7** (Unbiased optimum estimator). *Let $F : \mathcal{X} \to \mathbb{R}$ be $\mu$-strongly convex with, $\mathcal{X} = \mathbb{B}_R(x_0)$, $F(x) = \frac{1}{n} \sum_{i \in [n]} \hat{F}(x; i)$ for all $x \in \mathcal{X}$ and $\mathbb{E}\|\nabla F(x; i)\|^2 \leq G^2$ for all $x \in \mathcal{X}$ and $i \sim \mathsf{Unif}([n])$. Algorithm 7 outputs $\hat{x}_\star$ with $\mathbb{E}\hat{x}_\star = x_\star = \arg\min_{x \in \mathbb{B}_R(x_0)} F(x)$ and $\mathbb{E}\|x - x_\star\|_2^2 = O(\frac{G^2}{\mu^2} \log(nd))$ with expected $O(\log(nd))$ queries to $(\hat{F}(x; i), \nabla \hat{F}(x; i))$ and expected $O(d \log(nd))$ time.*

*Proof.* Note that

$$
\mathbb{E}\|\hat{x}_\star - x_\star\|^2 = \sum_{j=1}^{\infty} \frac{1}{2^j} \cdot \mathbb{E}\|x_0 - x_\star + 2^j(x_{j-1} - x_j)\|^2
$$

$$
\leq \sum_{j=1}^{\infty} \frac{2}{2^j} \mathbb{E}\left[ \|x_0 - x_\star\|^2 + 2^{2j} \|x_{j-1} - x_j\|^2 \right]
$$

$$
\leq 2\mathbb{E}\|x_0 - x_\star\|^2 + 4 \sum_{j=1}^{\infty} 2^j \mathbb{E}\left[ \|x_{j-1} - x_\star\|^2 + \|x_j - x_\star\|^2 \right]
$$

$$
= 10\|x_0 - x_\star\|^2 + 4 \sum_{j=1}^{\infty} (2^j + 2^{j+1})\mathbb{E}\|x_j - x_\star\|^2 \leq 12 \sum_{j=0}^{\infty} 2^j \mathbb{E}\|x_j - x_\star\|^2.
$$

Further, by definition of ODC we have that $\mathbb{E}\|x_j - x_\star\|_2^2 \leq (cG^2/\mu^2)2^{-j/2}$ for all $j \leq J_0$ where $c$ is the constant in Definition 1. Also, by Lemma 2 we have $\|x_j - x_\star\|_2^2 \leq (8G^2/\mu^2)\exp(-\lceil 2^{j/2}\rceil/(2d^2))$ for all $j > J_0$ (since by assumption and Jensen's inequality for $i \sim \mathsf{Unif}([n])$ we have $\|\nabla F(x)\|^2 = \|\mathbb{E}\nabla F(x; i)\|^2 \leq \mathbb{E}\|\nabla F(x, i)\|^2 \leq G^2$ for alll $x \in \mathcal{X}$ and therefore $F$ is $G$-Lipschitz). Note that $j \leq 5 \cdot 2^{j/4}$ for all $j \geq 1$ and $5 \cdot 2^{j/4} \ln 2 \leq 2^{j/2}/(4d^2)$ for all $j \geq 4 \log_2(14d^2)$. Consequently, $\frac{\lceil 2^{j/2}\rceil}{2d^2} \geq 2j \ln 2$ and $\|x_j - x_\star\|_2^2 \leq (8G^2/\mu^2)2^{-2j}$ for all $j > J_0$. Therefore,

$$
\mathbb{E}\|\hat{x}_\star - x_\star\|^2 \leq 12 \sum_{j=0}^{J_0} \frac{cG^2}{\mu^2} + 12 \sum_{j=J_0+1}^{\infty} \frac{8G^2}{\mu^2} 2^{-j} \leq 12(cJ_0 + 8)\frac{G^2}{\mu^2} = O\left( \frac{G^2}{\mu^2} \log(nd) \right).
$$

Further,

$$
\mathbb{E}x_\star = \sum_{j=1}^{\infty} 2^{-j}[x_0 + 2^j(x_j - x_{j-1})] = x_0 + \sum_{j=1}^{\infty} (x_j - x_{j-1}) = \lim_{j \to \infty} x_j = x_\star.
$$

Now, note that when $J \leq J_0$ the algorithm makes $2^J$ subgradient queries and runs in time $O(d2^J)$. Further, when $J > J_0$ by Lemma 2 the algorithm makes $\lceil 2^{J/2} \rceil \leq 2^{1+(J/2)}$ first-order oracle queries, costing $O(n2^{(J/2)})$ sub-gradient in total, and runs in time $O((nd^2 + d^4)2^{J/2})$. Consequently, the expected number of subgradient queries is upper bounded by

$$
\sum_{j \in [J_0]} \frac{1}{2^j} \cdot 2^j + \sum_{j > J_0} \frac{1}{2^j} \cdot n2^{1+(j/2)} = J_0 + \frac{2n}{2^{J_0/2}} \sum_{j=1}^{\infty} \frac{1}{2^{j/2}} = O(J_0) = O(\log(nd))
$$

where in the last step we used that $J_0 = \Omega(\log(n))$ and $\sum_{j=1}^{\infty} \frac{1}{2^{j/2}} = O(1)$. Similarly, since $J_0 \geq \log_2(nd^2 + d^4)$ the expected runtime is at most

$$
\sum_{j \in [J_0]} \frac{1}{2^j} \cdot O(d2^j) + \sum_{j > J_0} \frac{1}{2^j} \cdot O\left( (nd^2 + d^4)2^{j/2} \right)
$$

$$
= O(J_0 \cdot d) + O\left( 2^{-J_0}(nd^2 + d^4) \right) \cdot \sum_{j=1}^{\infty} \frac{1}{2^{j/2}} = O(J_0 \cdot d) = O(d \log(nd)).
$$

$\square$

## A.2 The sequential depth of our optimum estimator

Let us discuss the implications of our development—or more precisely, the lack thereof—on the parallel complexity of non-smooth optimization. Following the standard setting for this problem, consider the task of minimizing a $G$-Lipschitz convex function $f$ in a domain of diameter $R$ in $\mathbb{R}^d$ given the ability to query a subgradient oracle for $f$ in batches of $B$ parallel queries. That is, at round $t$ we query points $x_t^{(1)}, \ldots, x_t^{(B)}$ and observe subgradients $g_t^{(i)} \in \partial f(x_t^{(i)})$ for $i \in [B]$. In sufficiently high dimension, the ability to query $B$ points in parallel does not improve worst-case complexity: for required accuracy $\epsilon$ and algorithm with batch size $B = \mathrm{poly}(1/\epsilon)$, there exists a problem instance in dimension $d = O((\frac{GR}{\epsilon})^4 \log \frac{GR}{\epsilon})$ for which the algorithm must make $T = \Omega((\frac{GR}{\epsilon})^2)$ queries in sequence in order to find an $\epsilon$-accurate solution [11].

At first glance, our algorithms—and Algorithm 3 in particular—seem to contradict the lower bound described above. Indeed, the algorithm performs $O(\frac{GR}{\epsilon})$ iterations, where each iterations consists of averaging $\widetilde{O}(\frac{GR}{\epsilon})$ copies of the optimum estimator (1). Since we can compute copies of the estimator in parallel, the sequential depth of the algorithm appears to be only $O(\frac{GR}{\epsilon})$. To resolve the apparent contradiction, recall that each evaluation of (1) itself involves a sequential computation. In particular, while an evaluation of (1) has depth $\widetilde{O}(1)$ on average, it also has depth $\Omega(\frac{GR}{\epsilon})$ with probability $\Omega(\frac{\epsilon}{GR})$. Therefore, for a batch of $O(\frac{GR}{\epsilon})$ copies of the estimator, one of them would have depth $\Omega(\frac{GR}{\epsilon})$ with constant probability, implying an overall bound of $\Omega((\frac{GR}{\epsilon})^2)$ on the sequential depth of Algorithm 3.

Viewed another way, the parallelism lower bound implies a limitation on the sequential depth *distribution* of any lower bias optimum estimator. More specifically, let $\hat{T}$ be a random variable representing the sequential depth of a single copy of a low-bias optimum estimator and let $\hat{T}_1, \ldots, \hat{T}_{BK}$ be i.i.d. copies of that random variable, with $B$ and $K$ denoting batch size and AGD depth respectively. Then, when setting $B = K = O(\frac{GR}{\epsilon})$ we must have

$$\sum_{k \in [K]} \max_{b \in [B]} \left\{ \hat{T}_{b+(k-1)B} \right\} = \Omega\left( \left( \frac{GR}{\epsilon} \right)^2 \right)$$

with high probability. In particular, it is impossible to create a low-bias optimum estimator whose depth is $\widetilde{O}(1)$ with high probability. This fact might serve as a useful sanity check when designing new optimum estimators.

## A.3 Obtaining deterministic complexity bounds

This paper measures complexity via $\mathcal{N}_{\hat{\nabla} f}$, the number of gradient estimator evaluations by the algorithm. The performance guarantees of our algorithms bound the *expected complexity* while guaranteeing correctness with constant probability. In particular our guarantees in Sections 3 to 5 have the following general form: the algorithm outputs $x$ such that $f(x) - \min_{z \in \mathcal{X}} f(z) \leq \epsilon$ with probability at least $p$, and $\mathbb{E}\mathcal{N}_{\hat{\nabla} f} \leq \mathsf{C}(\epsilon)$. To guarantee a probability 1 bound on $\mathcal{N}_{\hat{\nabla} f}$, we may terminate the algorithm and output an arbitrary point whenever $\mathcal{N}_{\hat{\nabla} f}$ exceeds $\frac{2}{p}\mathsf{C}(\epsilon)$. By Markov's inequality such termination occurs with probability at most $p/2$ and therefore by the union bound we will output a correct $x$ (satisfying $f(x) - \min_{z \in \mathcal{X}} f(z) \leq \epsilon$) with probability at least $p/2$.

In Section 6 we describe a differentially-private algorithm with bounded expected error and expected gradient estimation complexity. Here too, we may terminate the algorithm if the number of gradient estimations exceeds the bound on the expectation by more than a constant, and maintain a constant probability bound on the error. Since the random amount of gradient estimations in this algorithm is independent of the input (and in fact can be computed ahead of the algorithm's execution), the termination strategy described above does not affect the algorithm's privacy guarantee.

## A.4 The optimal distance convergence rate

Definition 1 of an optimal-distance-convergence (ODC) algorithm implies a claim on the optimal rate of convergence (in Euclidean norm) to the minimizer of strongly-convex and Lipschitz functions. Lemma 1 shows that this rate is achievable, and here we sketch a matching lower bound, showing that this rate is not improvable and therefore optimal. More precisely, we exhibit a function $F$ that is $G$-Lipschitz, $\mu$ strongly-convex, has minimizer $x_\star$ and satisfies the following: for every algorithm

that queries points in the span of previously observed subgradients and outputs $x_T$ after $T$ queries, we have $\|x_T - x_\star\| \geq \Omega(G/(\mu\sqrt{T}))$. The restriction of queries to the span of previous gradients is a standard simplifying assumptions [42], and we can extend the claim to any randomized algorithm by choosing a random coordinate system [54, 11].

Let us describe our hard instance construction for algorithms that execute $T$ steps, which we denote by $F$. The function $F : \mathbb{R}^{2T} \to \mathbb{R}$ is a strongly-convex variant of Nemirovski's function [40, 39, 18, 11], defined as follows

$$F(x) := \frac{G}{2} \max_{i \in [2T]} \{x_{[i]}\} + \frac{\mu}{2} \|x\|^2.$$

Note that the function is $\mu$-strongly-convex, and—when constrained to a ball of radius $G/(2\mu)$ around the origin—is $G$-Lipschitz as required. It is also easy to verify that the minimizer of the function is

$$x_\star = -\frac{G}{4\mu T}\mathbf{1},$$

where $\mathbf{1}$ denotes to the all-ones vector in $\mathbb{R}^{2T}$, since a calculation shows that $0 \in \partial F(x_\star)$.

To establish our claimed lower bound, consider a subgradient oracle for $\max_{i \in [2T]}\{x_{[i]}\}$ which only outputs 1-sparse subgradients of $F$ (it is also possible to design differentiable hard instances via Moreau-Yoshida smoothing, see, e.g., [18]). Then, the query $x_T$ at iteration $T$ is in the span of $T$ 1-sparse vectors, which means that at least $T$ of its coordinates are zero. Recalling the expression of $x_\star$, this implies the claim that

$$\|x_T - x_\star\| \geq \frac{G}{4\mu T}\sqrt{T} = \Omega\left(\frac{G}{\mu\sqrt{T}}\right).$$

## B  Proofs and additional results from Section 2

### B.1  Analysis of EPOCHSGD

Algorithm 8 is a composite variant of the "epoch SGD" algorithm of Hazan and Kale [28]. We note that when $\psi(x) = \frac{\mu}{2}\|x - z\|^2$ (as it is in all of our applications), the gradient step in line 5 of the algorithm is simply

$$x_k^{t+1} = \mathsf{Proj}_{\mathcal{X}}\left(\frac{1}{1 + \mu\eta_k}\left[x_k^t + \mu\eta_k z - \eta_k\hat{\nabla}f(x_k^t)\right]\right),$$

where $\mathsf{Proj}_{\mathcal{X}}$ is the Euclidean projection to $\mathcal{X}$. To analyze Algorithm 8, we first prove the following standard single-epoch optimization guarantee. Below, we let $V_x(x') := \frac{1}{2}\|x' - x\|_2^2$ denote the Bregman divergence induced by $\frac{1}{2}\|\cdot\|_2^2$.

---

**Algorithm 8:** EPOCHSGD($\hat{\nabla}f, \psi, \mu, \mathcal{X}, T$)

**Input:** A $\mu$-strongly-convex function $F = f + \psi : \mathcal{X} \to \mathbb{R}$ with $f$ satisfying Assumption 1, iteration budget $T$.

**Parameters:** Initial step size $\eta_1 = 1/(4\mu)$ and epoch length $T_1 = 16$.

1  Initialize $x_1^0 \in \arg\min_{x \in \mathcal{X}} \psi(x)$, and set $k = 1$
2  **while** $\sum_{i \in [k]} T_i \leq T$ **do**
3  $\quad x_k^1 = \arg\min_{x \in \mathcal{X}}\left(\eta_k\psi(x) + \frac{1}{2}\|x - x_k^t\|^2\right)$
4  $\quad$ **for** $t = 1, 2, \cdots T_k - 1$ **do**
5  $\quad\quad x_k^{t+1} = \arg\min_{x \in \mathcal{X}}\left(\eta_k\left(\langle\hat{\nabla}f(x_k^t), x\rangle + \psi(x)\right) + \frac{1}{2}\|x - x_k^t\|^2\right)$
6  $\quad$ Set $x_{k+1}^0 = \frac{1}{T_k}\sum_{t \in [T_k]} x_k^t$, update $T_{k+1} = 2T_k$, $\eta_{k+1} = \eta_k/2$ and $k \leftarrow k + 1$
7  **return** $x = x_k^0$

---

**Lemma 3.** *Let $f : \mathcal{X} \to \mathbb{R}$ and $\hat{\nabla}f$ satisfy Assumption 1. For any $k \geq 1$, $T \geq 1$ and $u \in \mathcal{X}$, the iterates of Algorithm 8 satisfy*

$$\mathbb{E}\left[F\left(\frac{1}{T}\sum_{t \in [T]} x_k^t\right)\right] - F(u) \leq \frac{V_{x_k^0}(u)}{\eta T} + \frac{\eta}{2}G^2.$$

*Proof.* $x_k^t \equiv x^t$, $\eta_k \equiv \eta$, and $T_k \equiv T$. We furthermore let $x^{T+1} \equiv u$, and let $g^t := \hat{\nabla} f(x^t)$ for $t \geq 1$ and $g^0 := 0$. By the optimality conditions of the minimization in line 3 and line 5, we have

$$\langle \eta \left( g^{t-1} + \nabla \psi(x^t) \right) + x^t - x^{t-1}, x^t - u \rangle \leq 0 \text{ for all } t \in [T],$$

and consequently

$$\langle g^{t-1} + \nabla \psi(x^t), x^t - u \rangle \leq \frac{1}{\eta} \left( V_{x^{t-1}}(u) - V_{x^t}(u) - V_{x^{t-1}}(x^t) \right) \text{ for all } t \in [T].$$

Using the convexity of $\psi$ and the bound above, we obtain

$$\sum_{t \in [T]} \langle g^{t-1}, x^t - u \rangle + \sum_{t \in [T]} \left( \psi(x^t) - \psi(u) \right)$$

$$\leq \sum_{t \in [T]} \langle g^{t-1} + \nabla \psi(x^t), x^t - u \rangle$$

$$\leq \frac{1}{\eta} \sum_{t \in [T]} \left( V_{x^{t-1}}(u) - V_{x^t}(u) - V_{x^{t-1}}(x^t) \right)$$

$$\leq \frac{1}{\eta} V_{x^0}(u) - \frac{1}{\eta} \sum_{t=0}^{T} V_{x_t}(x^{t+1}).$$

Adding $\sum_{t \in [T]} \langle g^t, x^t - x^{t+1} \rangle$ to both sides, recalling that $x^{T+1} \equiv u$ and $g^t = \hat{\nabla} f(x^t) \mathbb{1}_{\{t>0\}}$, and rearranging terms, we have

$$\sum_{t \in [T]} \langle \hat{\nabla} f(x^t), x^t - u \rangle + \sum_{t \in [T]} \left( \psi(x^t) - \psi(u) \right)$$

$$\leq \frac{1}{\eta} V_{x^0}(u) - \frac{1}{\eta} \sum_{t=0}^{T} V_{x_t}(x^{t+1}) + \sum_{t \in [T]} \langle \hat{\nabla} f(x^t), x^t - x^{t+1} \rangle$$

$$\leq \frac{1}{\eta} V_{x^0}(u) + \sum_{t \in [T]} \frac{\eta}{2} \| \hat{\nabla} f(x^t) \|^2,$$

where in the last transition we used $\langle g, x - y \rangle \leq \frac{1}{\eta} V_y(x) + \frac{\eta}{2} \|g\|^2$. Taking expectation, applying Assumption 1 and using convexity of $f$, we have

$$\mathbb{E} \sum_{t \in [T]} \left( F(x^t) - F(u) \right) \leq \frac{1}{\eta} V_{x^0}(u) + \frac{T}{2} \eta G^2.$$

Dividing by $T$ and applying Jensen's inequality to bound $F\left( \frac{1}{T} \sum_{t \in [T]} x^t \right) \leq \frac{1}{T} \sum_{t \in [T]} F(x^t)$ yields the claimed bound. $\qquad \square$

We now are ready to prove the main guarantee of Algorithm 8 (see also Lemma 8, Theorem 5 in Hazan and Kale [28]), which implies Lemma 1.

**Proposition 3.** *Let $F : \mathcal{X} \to \mathbb{R}$ by a $\mu$-strongly-convex function of the form $F = f + \psi$, such that $f$ satisfies Assumption 1 and $x_\star = \operatorname{argmin}_{x \in \mathcal{X}} F(x)$. Then, for any $T \geq 1$, we have that $x = \text{EPOCHSGD}(\hat{\nabla} f, \psi, \mu, \mathcal{X}, T)$ satisfies*

$$\mathbb{E} F(x) - F(x_\star) \leq \frac{16G^2}{\mu T} \quad \text{and} \quad \mathbb{E} \|x - x_\star\|^2 \leq \frac{32G^2}{\mu^2 T}.$$

*Consequently,* EPOCHSGD *is an ODC algorithm with constant $c = 32$.*

*Proof.* First we claim that $F(x_1^0) - F(x_\star) \leq \frac{G^2}{2\mu}$. To see this we have by $\mu$-strong-convexity of $F$ that

$$F(x_\star) \geq F(x_1^0) + \langle \nabla f(x_1^0), x_\star - x_1^0 \rangle + \langle \nabla \psi(x_1^0), x_\star - x_1^0 \rangle + \frac{\mu}{2} \|x_1^0 - x_\star\|^2$$

$$\geq F(x_1^0) + \langle \nabla f(x_1^0), x_\star - x_1^0 \rangle + \frac{\mu}{2} \|x_1^0 - x_\star\|^2,$$

where we use the definition that $x_1^0 \in \arg\min_{x \in \mathcal{X}} \psi(x)$ and its first-order optimality condition for the second inequality. Rearranging terms gives

$$F(x_1^0) - F(x_\star) \leq -\langle \nabla f(x_1^0), x_\star - x_1^0 \rangle - \frac{\mu}{2}\|x_1^0 - x_\star\|^2$$

$$\leq \max_x \left( -\langle \nabla f(x_1^0), x - x_1^0 \rangle - \frac{\mu}{2}\|x_1^0 - x\|^2 \right) = \frac{\|\nabla f(x_1^0)\|^2}{2\mu} \leq \frac{G^2}{2\mu}.$$

For $x_\star = \arg\min_{x \in \mathcal{X}} F(x)$, we so define the potential $\Delta_k = F(x_k^0) - F(x_\star)$ and use induction to prove that $\mathbb{E}\Delta_k \leq \frac{G^2}{2^k \mu}$ for all $k$, with the base case $k = 1$ established above. Suppose that $\mathbb{E}\Delta_k \leq \frac{G^2}{2^k \mu}$ for a fixed $k$. Then for $k + 1$ Lemma 3 yields

$$\mathbb{E}\Delta_{k+1} \leq \frac{\mathbb{E}V_{x_k^0}(x_\star)}{\eta_k T_k} + \frac{\eta_k}{2}G^2 \overset{(i)}{\leq} \frac{\mathbb{E}\Delta_k}{\mu\eta_k T_k} + \frac{\eta_k}{2}G^2 \overset{(ii)}{=} \frac{\mathbb{E}\Delta_k}{4} + \frac{G^2}{2^{k+2}\mu} \overset{(iii)}{\leq} \frac{G^2}{2^{k+1}\mu},$$

with the transitions above following from $(i)$ strong convexity of $F$, which implies that $V_{x_k^0}(x_\star) = \frac{1}{2}\|x_k^0 - x_\star\|^2 \leq \frac{1}{\mu}\Delta_k$; $(ii)$ the choice of parameters ensures $\eta_k T_k = \frac{4}{\mu}$ and $\eta_k = \frac{1}{2^{k+1}\mu}$; and $(iii)$ the inductive hypothesis $\mathbb{E}\Delta_k \leq \frac{G^2}{2^k \mu}$. This completes the induction.

Let $K$ be such that the algorithm outputs $x = x_K^0$, and note that $T \leq 16 \cdot (2^K - 1) - 1$. Therefore, we have

$$\mathbb{E}F(x) - F(x_\star) = \mathbb{E}\Delta_K \leq \frac{G^2}{2^K \mu} \leq \frac{16G^2}{\mu T},$$

and

$$\mathbb{E}\|x - x_\star\|^2 \leq \frac{2}{\mu}(\mathbb{E}F(x) - F(x_\star)) \leq \frac{32G^2}{\mu^2 T}.$$

Recalling Definition 1, we conclude that EPOCHSGD is an ODC algorithm with constant $c = 32$. $\quad\square$

**Remark 1** (Using EPOCHSGD for optimum estimation)**.** *When using* EPOCHSGD *as the ODC algorithm in our MLMC optimum estimator* (1)*, we need only call once with $T = 2^J$ and take $x_0, x_{J-1}$ and $x_J$ to be the iterates $x_1^0, x_{K-1}^0$ and $x_K^0$ of* EPOCHSGD*, for $K$ the last value of $k$ that* EPOCHSGD *reaches.*

## B.2 Proof of Theorem 1

**Theorem 1.** *Let $f$ and $\hat{\nabla}f$ satisfy Assumption 1, $F = f + \psi$ be $\mu$-strongly convex with minimizer $x_\star \in \mathcal{X}$, and $\delta, \sigma > 0$. The function* OPTEST$(\hat{\nabla}f, \psi, \mu, \delta, \sigma^2, \mathcal{X})$ *outputs $\hat{x}_\star$ satisfying*

$$\|\mathbb{E}\hat{x}_\star - x_\star\| \leq \delta \quad and \quad \mathbb{E}\|\hat{x}_\star - x_\star\|^2 \leq \sigma^2$$

*using $\mathcal{N}_{\hat{\nabla}f}$ stochastic gradient computations, where*

$$\mathbb{E}\mathcal{N}_{\hat{\nabla}f} = O\left( \frac{G^2}{\mu^2\sigma^2} \log^2\left( \frac{G}{\mu\min\{\delta,\sigma\}} \right) + \log\left( \frac{G}{\mu\min\{\delta,\sigma\}} \right) \right).$$

*Proof.* Write the algorithm's output as $\hat{x}_\star = \frac{1}{N}\sum_{i=1}^N \hat{x}_\star^{(i)}$ where $\hat{x}_\star^{(1)}, \ldots, \hat{x}_\star^{(N)}$ are independent draws of the estimator (1), with

$$T_{\max} = \left\lceil \frac{(2c)^2 G^2}{\mu^2 \min\{\delta^2, \frac{1}{2}\sigma^2\}} \right\rceil \quad \text{and} \quad N = \left\lceil \frac{2(4c)^2 G^2}{\mu^2\sigma^2} \log(T_{\max}) \right\rceil$$

as in Algorithm 1. Then, Proposition 1 implies that

$$\|\mathbb{E}\hat{x}_\star^{(1)} - x_\star\| \leq \min\left\{ \delta, \frac{1}{\sqrt{2}}\sigma \right\} \quad \text{and} \quad \mathbb{E}\|\hat{x}_\star^{(1)} - \mathbb{E}\hat{x}_\star^{(1)}\|^2 \leq \frac{N}{2}\sigma^2.$$

Noting that $\mathbb{E}\hat{x}_\star = \mathbb{E}\hat{x}_\star^{(1)}$ and

$$\mathbb{E}\|\hat{x}_\star - x_\star\|^2 = \frac{1}{N}\mathbb{E}\|\hat{x}_\star^{(1)} - \mathbb{E}\hat{x}_\star^{(1)}\|^2 + \|\mathbb{E}\hat{x}_\star^{(1)} - x_\star\|^2,$$

we obtain the claimed bias and error bounds. Finally, Proposition 1 guarantees that $\mathbb{E}\mathcal{N}_{\hat{\nabla}f} = O(N \cdot \log(T_{\max}))$, giving the claimed bound on the number of evaluations. $\quad\square$

### B.3 Properties of the proximal operator and Moreau envelope

For a convex function $f : \mathcal{X} \to \mathbb{R}$ we recall the definitions of

$$\text{the proximal operator } \mathsf{P}_{f,\lambda}(x) := \underset{y \in \mathcal{X}}{\operatorname{argmin}}\left\{f(y) + \tfrac{\lambda}{2}\|y - x\|^2\right\}$$

$$\text{and the Moreau envelope } f_\lambda(x) := \underset{y \in \mathcal{X}}{\min}\left\{f(y) + \tfrac{\lambda}{2}\|y - x\|^2\right\}.$$

Below, we collect several well-known properties that we use throughout the paper.

**Fact 1.** *Given a convex function $f : \mathcal{X} \to \mathbb{R}$, and $\lambda > 0$ defined on a closed convex set $\mathcal{X}$, the following properties of the Moreau envelope $f_\lambda : \mathbb{R}^d \to \mathbb{R}$ and the proximal operator $\mathsf{P}_{f,\lambda} : \mathcal{X} \to \mathcal{X}$ hold for all $x \in \mathcal{X}$*

1. *Convexity: $f_\lambda$ is convex.*

2. *Differentiablility: $f_\lambda$ is $\lambda$-smooth and $\nabla f_\lambda(x) = \lambda(x - \mathsf{P}_{f,\lambda}(x))$.*

3. *Approximation: If $f$ is $G$-Lipschitz then $f(x) - \frac{G^2}{2\lambda} \leq f_\lambda(x) \leq f(x)$.*

4. *Subgradient: $\nabla f_\lambda(x) \in \partial f(\mathsf{P}_{f,\lambda}(x))$,*

5. *Three point inequality: for all $u \in \mathcal{X}$:*

$$\langle \nabla f_\lambda(x), \mathsf{P}_{f,\lambda}(x) - u \rangle \leq \frac{\lambda}{2}\|u - x\|^2 - \frac{\lambda}{2}\|u - \mathsf{P}_{f,\lambda}(x)\|^2 - \frac{\lambda}{2}\|x - \mathsf{P}_{f,\lambda}(x)\|^2.$$

See [29, Section 4.1] as well as [13, Lemma 1] and [51, Lemma 1] for proofs and additional background and properties.

## C   Proofs from Section 3

In this section, we give a proof of Theorem 3. Before we give the technical details, we briefly comment on our algorithm and its analysis. Algorithm 3 is at its core an instantiation of Nesterov's accelerated gradient method applied to the Moreau envelope $f_\lambda(x) = \min_{y \in \mathbb{B}_R(0)}\left\{f(y) + \frac{\lambda}{2}\|y - x\|^2\right\}$. We compute stochastic gradient estimates of $f_\lambda$ via Algorithm 2, and apply techniques from [1, 2] to bound the accumulated error.

Based on the iterates $\{x_k, v_k\}$ of Algorithm 3, we define

$$E_k = f_\lambda(x_k) - f_\lambda(u), R_k = \frac{1}{2}\|v_k - u\|^2, \text{ and } P_k = k(k+1)E_k + 12\lambda R_k$$

for any fixed $u \in \mathbb{B}_R(0)$. We first prove that (conditioned on the iterates $x_{k-1}, v_{k-1}$) the potential $P_k$ cannot increase significantly in expectation.

**Lemma 4.** *Consider an execution of Algorithm 3 with parameters given by Theorem 3. Fix any $u \in \mathbb{B}_R(0)$. For any $k \geq 1$ we have $y_{k-1} \in \mathbb{B}_R(0)$ and*

$$\mathbb{E}\left[P_k | x_{k-1}, v_{k-1}\right] \leq P_{k-1} + \epsilon k.$$

*Proof.* We first remark that $x_{k-1} \in \mathcal{X} \subseteq \mathbb{B}_R(0)$ and $v_{k-1} \in \mathbb{B}_R(0)$ by construction. As a result, $y_{k-1} \in \mathbb{B}_R(0)$ as well. Following [1, 2], we define the function

$$\mathsf{Prog}(y; g) := \underset{x \in \mathcal{X}}{\min}\left\{\frac{3\lambda}{2}\|x - y\|^2 + \langle g, x - y \rangle\right\}.$$

We observe

$$\mathsf{Prog}(y_{k-1}; g_k) = \min_{x \in \mathcal{X}} \left\{ \frac{3\lambda}{2} \|x - y_{k-1}\|^2 + \langle g_k, x - y_{k-1} \rangle \right\} \tag{6}$$

$$\overset{(i)}{=} \frac{3\lambda}{2} \|x_k - y_{k-1}\|^2 + \langle g_k, x_k - y_{k-1} \rangle$$

$$= \left( \frac{\lambda}{2} \|x_k - y_{k-1}\|^2 + \langle \nabla f_\lambda(y_{k-1}), x_k - y_{k-1} \rangle \right)$$

$$\quad + \lambda \|x_k - y_{k-1}\|^2 + \langle g_k - \nabla f_\lambda(y_{k-1}), x_k - y_{k-1} \rangle$$

$$\overset{(ii)}{\geq} f_\lambda(x_k) - f_\lambda(y_{k-1}) + \lambda \|x_k - y_{k-1}\|^2 + \langle g_k - \nabla f_\lambda(y_{k-1}), x_k - y_{k-1} \rangle$$

$$\overset{(iii)}{\geq} f_\lambda(x_k) - f_\lambda(y_{k-1}) - \frac{1}{4\lambda} \|g_k - \nabla f_\lambda(y_{k-1})\|^2.$$

Here, we use $(i)$ the definition of $x_k$, $(ii)$ smoothness of $f_\lambda$ (Item 2 of Fact 1), and $(iii)$ Young's inequality $\langle a, b \rangle + \frac{1}{2}\|b\|^2 \geq -\frac{1}{2}\|a\|^2$ with $a = \frac{1}{2\lambda}(g_k - \nabla f_\lambda(y_{k-1}))$ and $b = 2\lambda(x_k - y_{k-1})$. Define the point

$$\widetilde{y}_{k-1} = \frac{k-1}{k+1} x_{k-1} + \frac{2}{k+1} v_k.$$

We observe that

$$y_{k-1} - \widetilde{y}_{k-1} = \left( \frac{k-1}{k+1} x_{k-1} + \frac{2}{k+1} v_{k-1} \right) - \left( \frac{k-1}{k+1} x_{k-1} + \frac{2}{k+1} v_k \right) = \frac{2}{k+1} (v_{k-1} - v_k).$$

Consequently, we have

$$\frac{k}{6\lambda} \langle g_k, v_{k-1} - u \rangle = \frac{k}{6\lambda} \langle g_k, v_{k-1} - v_k \rangle + \frac{k}{6\lambda} \langle g_k, v_k - u \rangle$$

$$\overset{(i)}{\leq} \frac{k}{6\lambda} \langle g_k, v_{k-1} - v_k \rangle + \frac{1}{2} \left( \|v_{k-1} - u\|^2 - \|v_k - u\|^2 - \|v_{k-1} - v_k\|^2 \right)$$

$$\overset{(ii)}{=} \frac{k(k+1)}{12\lambda} \langle g_k, y_k - \widetilde{y}_{k-1} \rangle - \frac{(k+1)^2}{8} \|y_k - \widetilde{y}_{k-1}\|^2 + R_{k-1} - R_k$$

$$\overset{(iii)}{\leq} \frac{k(k+1)}{12\lambda} \left( \langle g_k, y_{k-1} - \widetilde{y}_{k-1} \rangle - \frac{3\lambda}{2} \|y_k - \widetilde{y}_{k-1}\|^2 \right) + R_{k-1} - R_k$$

$$\overset{(iv)}{\leq} -\frac{k(k+1)}{12\lambda} \mathsf{Prog}(y_{k-1}; g_k) + R_{k-1} - R_k$$

$$\overset{(v)}{\leq} \frac{k(k+1)}{12\lambda} \left( f_\lambda(y_{k-1}) - f_\lambda(x_k) + \frac{1}{4\lambda} \|g_k - \nabla f_\lambda(y_{k-1})\|^2 \right) + R_{k-1} - R_k. \tag{7}$$

Here we use $(i)$ the proximal three-point inequality (Item 5 of Fact 1), $(ii)$ the definition of $\widetilde{y}_{k-1}$, $(iii)$ $\frac{(k+1)^2}{8} \geq \frac{3\lambda}{2} \cdot \frac{k(k+1)}{12\lambda}$ and $\|y_{k-1} - \widetilde{y}_{k-1}\|^2 \geq 0$, $(iv)$ the definition of $\mathsf{Prog}$, and $(v)$ Equation (6). Thus,

$$\frac{k}{6\lambda} \left( f_\lambda(y_{k-1}) - f_\lambda(u) \right) \leq \frac{k}{6\lambda} \langle \nabla f_\lambda(y_{k-1}), y_{k-1} - u \rangle$$

$$\leq \frac{k}{6\lambda} \langle \nabla f_\lambda(y_{k-1}), y_{k-1} - v_{k-1} \rangle + \frac{k}{6\lambda} \langle \nabla f_\lambda(y_{k-1}), v_{k-1} - u \rangle$$

$$\overset{(i)}{=} \frac{k(k-1)}{12\lambda} \langle \nabla f_\lambda(y_{k-1}), x_{k-1} - y_{k-1} \rangle + \frac{k}{6\lambda} \langle \nabla f_\lambda(y_{k-1}), v_{k-1} - u \rangle$$

$$\leq \frac{k(k-1)}{12\lambda} \left( f_\lambda(x_{k-1}) - f_\lambda(y_{k-1}) \right) + \frac{k}{6\lambda} \langle \nabla f_\lambda(y_{k-1}), v_{k-1} - u \rangle$$

$$\overset{(ii)}{\leq} \frac{k(k-1)}{12\lambda} \left( f_\lambda(x_{k-1}) - f_\lambda(y_{k-1}) \right) + R_{k-1} - R_k$$

$$\quad + \frac{k(k+1)}{12\lambda} \left( f_\lambda(y_{k-1}) - f_\lambda(x_k) + \frac{1}{4\lambda} \|g_k - \nabla f_\lambda(y_{k-1})\|^2 \right)$$

$$\quad + \frac{k}{6\lambda} \langle \nabla f_\lambda(y_{k-1}) - g_k, v_{k-1} - u \rangle,$$

where we use $(i)$ $y_{k-1} - v_{k-1} = \frac{k-1}{2}(x_{k-1} - y_{k-1})$ and $(ii)$ Equation (7). Rearranging, we obtain

$$\frac{1}{12\lambda}(P_k - P_{k-1}) = \frac{k(k+1)}{12\lambda}E_k + R_k - \frac{k(k-1)}{12\lambda}E_{k-1} - R_{k-1}$$
$$\leq \frac{k(k+1)}{48\lambda^2}\|g_k - \nabla f_\lambda(y_{k-1})\|^2 + \frac{k}{6\lambda}\langle \nabla f_\lambda(y_{k-1}) - g_k, v_{k-1} - u\rangle \quad (8)$$

Applying Corollary 2, we observe

$$\mathbb{E}\left[\|g_k - \nabla f_\lambda(y_k)\|^2 | x_{k-1}, v_{k-1}\right] \leq \sigma_k^2 = \frac{2\epsilon\lambda}{k+1}$$

and

$$\mathbb{E}\left[\langle \nabla f_\lambda(y_{k-1}) - g_k, v_{k-1} - u\rangle | x_{k-1}, v_{k-1}\right] \leq \|\mathbb{E}[g_k] - \nabla f_\lambda(y_{k-1})\|\|v_{k-1} - u\| \leq 2R\delta_k = \frac{\epsilon}{4}$$

by the Cauchy-Schwarz inequality, the constraint that $u, v_k \in \mathbb{B}_R(0)$, and the choice of parameters $\sigma_k, \delta_k$. Taking expectations and applying these to Equation (8), we obtain

$$\frac{1}{12\lambda}(\mathbb{E}[P_k|x_{k-1}, v_{k-1}] - P_{k-1}) \leq \frac{\epsilon k}{24\lambda} + \frac{\epsilon k}{24\lambda} = \frac{\epsilon k}{12\lambda}.$$

Multiplying both sides by $12\lambda$ yields the claim. $\square$

With Lemma 4 in hand, we complete the proof of Theorem 3.

**Theorem 3.** *Let $f : \mathbb{B}_R(0) \to \mathbb{R}$ and $\hat{\nabla} f$ satisfy Assumption 1. Let $\mathcal{X} \subseteq \mathbb{B}_R(0)$ be a convex set admitting a projection oracle $\mathsf{Proj}_{\mathcal{X}}$. Let $x_0 \in \mathcal{X}$ be an initial point with $\|x - x_\star\| \leq D$ for some $x_\star \in \mathcal{X}$. With $\lambda = \frac{2G^2}{\epsilon}$, $\delta_k = \frac{\epsilon}{8R}$, $\sigma_k^2 = \frac{2\epsilon\lambda}{k+1}$, and $T = \frac{7GD}{\epsilon}$ Algorithm 3 computes $x \in \mathcal{X}$ with $\mathbb{E}[f(x)] \leq f(x_\star) + \epsilon$ with complexity $\mathbb{E}\mathcal{N}_{\hat{\nabla} f} = O\left(\frac{G^2 D^2}{\epsilon^2}\log^2\left(\frac{GR}{\epsilon}\right)\right)$ and $O\left(\frac{GD}{\epsilon}\right)$ calls to $\mathsf{Proj}_{\mathcal{X}}$.*

*Proof of Theorem 3.* Applying the law of total probability and inductively applying Lemma 4, we obtain

$$\mathbb{E}[P_T] \leq P_0 + \epsilon\sum_{k=1}^{T} k = P_0 + \frac{\epsilon}{2}T(T+1).$$

We choose $u = x_\star$ and observe $P_T = T(T+1)E_T + 12\lambda R_T \geq T(T+1)(f_\lambda(x_T) - f_\lambda(x_\star))$ and $P_0 = 12\lambda R_0 \leq 6\lambda D^2$. Plugging these in, we have

$$\mathbb{E}[f_\lambda(x_T)] - f_\lambda(x_\star) \leq \frac{6\lambda D^2}{T(T+1)} + \frac{\epsilon}{2}.$$

As $f$ is $G$-Lipschitz, we apply Item 1 of Fact 1 and our choices of $\lambda$ and $T$: this gives

$$\mathbb{E}[f(x_T)] - f(x_\star) \leq \frac{G^2}{2\lambda} + \frac{6\lambda D^2}{T(T+1)} + \frac{\epsilon}{2} \leq \frac{\epsilon}{4} + \frac{12G^2 D^2}{\epsilon T^2} + \frac{\epsilon}{2} \leq \frac{\epsilon}{4} + \frac{12\epsilon}{49} + \frac{\epsilon}{2} < \epsilon$$

as desired.

To finish, we bound the number of oracle queries. The bound on the number of projection oracle calls is immediate since we only call it once per iteration of the algorithm. To bound the number of stochastic gradients needed, we apply Corollary 2 together with the fact that $y_k \in \mathbb{B}_R(0)$ at all times. Thus, we need

$$O\left(\sum_{k=0}^{T-1} \frac{G^2}{\sigma_k^2}\log^2\left(\frac{G}{\delta_k}\right)\right) = O\left(\sum_{k=0}^{T-1} \frac{G^2(k+1)}{\epsilon\lambda}\log^2\left(\frac{GR}{\epsilon}\right)\right) = O\left(\frac{G^2 D^2}{\epsilon^2}\log^2\left(\frac{GR}{\epsilon}\right)\right)$$

subgradient computations as desired. $\square$

# D  Proofs from Section 4

## D.1  Analysis of the stochastic accelerated proximal method

In this section we provide a complete analysis of the stochastic accelerated proximal method. We first prove Lemma 5, which shows potential decrease in (conditional) expectation for the iterates of Algorithm 4. Then we give Lemma 6 which provides an in-expectation bound on the potential when the algorithm terminates. In Lemma 7 we give a deterministic error bound resulting from the growth of the $A_k$ sequence. We then combine these ingredients to prove Proposition 2.

**Notation.** Define the filtration

$$\mathcal{F}_k = \sigma(x_1, v_1, A_1, \zeta_1 \ldots, x_k, v_k, A_k, \zeta_k)$$

where $\zeta_i$ is the internal randomness in $\textsc{NextLambda}(x_i, v_i, A_i)$. Throughout, we let

$$\hat{x}_k = \mathsf{P}_{f, \lambda_k}(y_{k-1})$$

denote the exact proximal mapping which iteration $x_k$ of the algorithm approximate. We note that $A_{k+1}, y_k, \hat{x}_{k+1}, \in \mathcal{F}_k$, i.e., they are deterministic when conditioned on $x_k, v_k, A_k, \zeta_k$.

For each iteration of Algorithm 4, we obtain the following bound on potential decrease.

**Lemma 5.** *Let $f : \mathcal{X} \to \mathbb{R}$ satisfy Assumption 1. If $\mathcal{X} \subseteq \mathbb{B}_R(x_0)$, we have*

$$\mathbb{E}\big[A_{k+1}(f(x_{k+1}) - f(x_\star)) + \|v_{k+1} - x_\star\|^2 \mid \mathcal{F}_k\big]$$

$$\leq A_k(f(x_k) - f(x_\star)) + \|v_k - x_\star\|^2 - \frac{1}{6}\lambda_{k+1}A_{k+1}\|\hat{x}_{k+1} - y_k\|^2$$

$$+ \lambda_{k+1}a_{k+1}^2\varphi_{k+1} + a_{k+1}^2\sigma_{k+1}^2 + 2Ra_{k+1}\delta_{k+1}.$$

*Proof.* We let

$$\hat{g}_k = \nabla f_{\lambda_k}(y_{k-1}) = \lambda_k(y_{k-1} - \hat{x}_k)$$

and bound from both sides the quantity $a_{k+1}\langle \hat{g}_{k+1}, v_k - x_\star \rangle$. First, note that

$$v_k - x_\star = \hat{x}_{k+1} - x_\star + \frac{A_k}{a_{k+1}}(\hat{x}_{k+1} - x_k) - \frac{A_{k+1}}{a_{k+1}}(\hat{x}_{k+1} - y_k).$$

Since $\hat{g}_{k+1} \in \partial f(\hat{x}_{k+1})$ (see Item 4 in Fact 1), $f$ is convex and $\langle \hat{g}_{k+1}, \hat{x}_{k+1} - y_k \rangle = -\lambda_{k+1}\|\hat{x}_{k+1} - y_k\|^2$, we have that

$$\langle \hat{g}_{k+1}, v_k - x_\star \rangle = \langle \hat{g}_{k+1}, \hat{x}_{k+1} - x_\star \rangle + \frac{A_k}{a_{k+1}}\langle \hat{g}_{k+1}, \hat{x}_{k+1} - x_k \rangle - \frac{A_{k+1}}{a_{k+1}}\langle \hat{g}_{k+1}, \hat{x}_{k+1} - y_k \rangle$$

$$\geq f(\hat{x}_{k+1}) - f(x_\star) + \frac{A_k}{a_{k+1}}(f(\hat{x}_{k+1}) - f(x_k)) - \frac{A_{k+1}}{a_{k+1}}\langle \hat{g}_{k+1}, \hat{x}_{k+1} - y_k \rangle$$

$$= \frac{A_{k+1}}{a_{k+1}}(f(\hat{x}_{k+1}) - f(x_\star)) - \frac{A_k}{a_{k+1}}(f(x_k) - f(x_\star)) + \frac{\lambda_{k+1}A_{k+1}}{a_{k+1}}\|\hat{x}_{k+1} - y_k\|^2.$$

Moreover, by definition of $x_{k+1}$ we have that

$$\mathbb{E}[f(x_{k+1}) \mid \mathcal{F}_k] \leq \mathbb{E}\left[f(x_{k+1}) + \frac{\lambda_{k+1}}{2}\|x_{k+1} - y_k\|^2 \ \middle|\ \mathcal{F}_k\right] \leq f(\hat{x}_{k+1}) + \frac{\lambda_{k+1}}{2}\|\hat{x}_{k+1} - y_k\|^2 + \varphi_{k+1}.$$

Substituting back, we have

$$a_{k+1}\langle \hat{g}_{k+1}, v_k - x_\star \rangle \geq A_{k+1}(\mathbb{E}[f(x_{k+1}) \mid \mathcal{F}_k] - f(x_\star)) - A_k(f(x_k) - f(x_\star))$$

$$+ \frac{\lambda_{k+1}A_{k+1}}{2}\|\hat{x}_{k+1} - y_k\|^2 - A_{k+1}\varphi_{k+1}. \tag{9}$$

To upper bound $a_{k+1}\langle \hat{g}_{k+1}, v_k - x_\star \rangle$, note that, since $x_\star \in \mathcal{X}$,

$$\|v_{k+1} - x_\star\|^2 \leq \left\|v_k - \frac{1}{2}a_{k+1}g_{k+1} - x_\star\right\|^2 = \|v_k - x_\star\|^2 - a_{k+1}\langle g_{k+1}, v_k - x_\star \rangle + \frac{a_{k+1}^2}{4}\|g_{k+1}\|^2.$$

Our Moreau Envelope gradient estimtor (see Corollary 2) guarantees that

$$\mathbb{E}[\langle g_{k+1}, v_k - x_\star \rangle \mid \mathcal{F}_k] \geq \langle \hat{g}_{k+1}, v_k - x_\star \rangle - \|\mathbb{E}[g_{k+1} \mid \mathcal{F}_k] - \hat{g}_{k+1}\| \|v_k - x_\star\|$$

$$\geq \langle \hat{g}_{k+1}, v_k - x_\star \rangle - 2R\delta_{k+1},$$

and moreover

$$\mathbb{E}\left[\|g_{k+1}\|^2 \ \middle|\ \mathcal{F}_k\right] = \left(1 + \frac{1}{3}\right)\mathbb{E}\|\hat{g}_{k+1}\|^2 + (1 + 3)\mathbb{E}\left[\|g_{k+1} - \hat{g}_{k+1}\|^2 \ \middle|\ \mathcal{F}_k\right]$$

$$\leq \frac{4}{3}\|\hat{g}_{k+1}\|^2 + 4\sigma_{k+1}^2.$$

Combining the last three displays and rearranging, we obtain

$$a_{k+1} \langle \hat{g}_{k+1}, v_k - x_\star \rangle \le \|v_k - x_\star\|^2 - \mathbb{E}\Big[\|v_{k+1} - x_\star\|^2 \;\Big|\; \mathcal{F}_k\Big] + \frac{\lambda_{k+1}^2 a_{k+1}^2}{3}\|\hat{x}_{k+1} - y_k\|^2$$
$$+ a_{k+1}^2 \sigma_{k+1}^2 + 2Ra_{k+1}\delta_{k+1} \tag{10}$$

Combining (9) and (10) and simplifying using $A_{k+1} = \lambda_{k+1}a_{k+1}^2$, we obtain the claimed bound. $\quad\square$

Combining Lemma 5 with the optional stopping theorem, one obtains the following bound on the potential at the final iteration $K$ of the algorithm.

**Lemma 6.** *Let $K \le K_{\max}$ be the iteration in which Algorithm 4 returns and let*

$$\bar{\varepsilon} \ge \max_{k \le K_{\max}} \left\{ \lambda_k a_k \varphi_k + a_k \sigma_k^2 + 2R\delta_k \right\}$$

*with probability 1. Then, under the assumptions of Lemma 5, we have*

$$\mathbb{E}\left[ A_K(f(x_K) - f(x_\star) - \bar{\varepsilon}) + \frac{1}{6}\sum_{i \le K} \lambda_i A_i \|\hat{x}_i - y_{i-1}\|^2 \right] \le A_0(f(x_0) - f(x_\star)) + R^2.$$

*Proof.* Define $M_k = A_k(f(x_k) - f(x_\star) - \bar{\varepsilon}) + \frac{1}{6}\sum_{i \le k} \lambda_i A_i \|\hat{x}_i - y_{i-1}\|^2 + \|v_k - x_\star\|^2$ for all $k \in [K]$. We argue that it is a supermartingale adapted to filtration $\mathcal{F}_k$. Clearly, $\mathbb{E}[|M_k|] < \infty$ for each $k$ due to boundedness of $f, K, \lambda_i$ and $A_i$. It therefore suffices to show that $\mathbb{E}[M_{k+1}|\mathcal{F}_k] \le M_k$ for all $k + 1 \in [K]$. By Lemma 5 we have

$$\mathbb{E}\left[M_{k+1}|\mathcal{F}_k\right] \le A_k(f(x_k) - f(x_\star)) + \|v_k - x_\star\|^2 - \frac{1}{6}\lambda_{k+1}A_{k+1}\|\hat{x}_{k+1} - y_k\|^2$$

$$+ \lambda_{k+1}a_{k+1}^2\varphi_{k+1} + a_{k+1}^2\sigma_{k+1}^2 + 2Ra_{k+1}\delta_{k+1} - A_{k+1}\bar{\varepsilon} + \frac{1}{6}\sum_{i \le k+1} \lambda_i A_i\|\hat{x}_{i+1} - y_i\|^2$$

$$\le A_k(f(x_k) - f(x_\star) - \bar{\varepsilon}) + \|v_k - x_\star\|^2 + \frac{1}{6}\sum_{i \le k} \lambda_i A_i\|\hat{x}_{i+1} - y_i\|^2 = M_k,$$

where the second inequality used the definition of $\bar{\varepsilon}$ and $A_{k+1} = A_k + a_{k+1}$ for the second inequality. This completes the proof that $M_k$ being a supermartingale adapted to filtration $\mathcal{F}_k$.

Now note $K$ is a stopping time adapted to $\mathcal{F}_k$ as it only depends on $A_{k+1}$. Also, $K$ as a random variable is finitely bounded by $K_{\max}$ with probability 1. Thus, by optional stopping theorem for supermartingale [26], we have

$$\mathbb{E}M_K \le M_0 = A_0(f(x_0) - f(x_\star) - \bar{\varepsilon}) + \|v_0 - x_\star\|^2 \le A_0(f(x_0) - f(x_\star)) + R^2.$$

$\square$

Further, following a similar argument to Carmon et al. [12, 13], we obtain a deterministic growth bound on the coefficients $A_k$.

**Lemma 7.** *Fix $k > 0$ and let*

$$T_\lambda = \sum_{i \le k} \mathbb{1}_{\{\lambda_i < 2\lambda_{\min}\}} \quad and \quad T_r = \sum_{i \le k} \mathbb{1}_{\{\|\hat{x}_i - y_{i-1}\| \ge 3r/4\}}$$

*count the number of times $\lambda_i < 2\lambda_{\min}$ and $\|\hat{x}_i - y_{i-1}\| \ge 3r/4$, respectively. Then, the following holds with probability 1,*

$$\frac{1}{A_k}\left(9R^2 - \frac{1}{6}\sum_{i \le k} \lambda_i A_i\|\hat{x}_i - y_{i-1}\|^2\right) \le O\left(\min\left\{\frac{\lambda_{\min}R^2}{T_\lambda^2}, \frac{R^2}{A_0}\exp\left(-\Omega(1)\frac{r^{2/3}}{R^{2/3}}T_r\right)\right\}\right).$$

*Proof.* When $9R^2 - \frac{1}{6}\sum_{i \le k} \lambda_i A_i\|\hat{x}_i - y_{i-1}\|^2 \le 0$, the inequality holds true trivially. Thus, we only consider the case when $\frac{1}{6}\sum_{i \le k} \lambda_i A_i\|\hat{x}_i - y_{i-1}\|^2 \le 9R^2$. Consider the following iterate index subsets

$$\mathcal{I}_\lambda := \{i \le k : \lambda_i < 2\lambda_{\min}\}$$

and, for $t \leq k$
$$\mathcal{I}_{r,t} := \{i \leq t : \|\hat{x}_i - y_{i-1}\| \geq 3r/4\}.$$

We first show that
$$\frac{1}{A_k}\left(9R^2 - \frac{1}{6}\sum_{i \leq k}\lambda_i A_i\|\hat{x}_i - y_{i-1}\|^2\right) \leq O\left(\frac{R^2}{A_0}\exp\left(-\Omega(1)\frac{r^{2/3}}{R^{2/3}}T_r\right)\right). \tag{11}$$

To see this, observe that for any $t \leq k$ by definition of $\mathcal{I}_{r,t}$,
$$\frac{1}{6}\sum_{i \in \mathcal{I}_{r,t}}\lambda_i A_i \cdot \left(\frac{9}{16}r^2\right) \leq \frac{1}{6}\sum_{i \leq t}\lambda_i A_i\|\hat{x}_i - y_{i-1}\|^2 \leq 9R^2,$$

which by rearranging terms implies
$$\sum_{i \in \mathcal{I}_{r,t}}\lambda_i A_i \leq \frac{96R^2}{r^2}. \tag{12}$$

Note the reverse Hölder's inequality with $p = 2/3$ states that for any $u, v \in \mathbb{R}_{>0}^d$,
$$\langle u, v \rangle \geq \left(\sum_{i \in [d]}u_i^{2/3}\right)^{3/2} \cdot \left(\sum_{i \in [d]}v_i^{-2}\right)^{-1/2}.$$

We have
$$\sqrt{A_t} \overset{(i)}{\geq} \frac{1}{2}\sum_{i \in \mathcal{I}_{r,t}}\frac{1}{\sqrt{\lambda_i}} \overset{(ii)}{\geq} \frac{1}{2}\left(\sum_{i \in \mathcal{I}_{r,t}}\left(\sqrt{A_i}\right)^{2/3}\right)^{3/2} \cdot \left(\sum_{i \in \mathcal{I}_{r,t}}\left(\frac{1}{\sqrt{A_i\lambda_i}}\right)^{-2}\right)^{-1/2}$$
$$\overset{(iii)}{\geq} \frac{r}{8\sqrt{6}R} \cdot \left(\sum_{i \in \mathcal{I}_{r,t}}\left(\sqrt{A_i}\right)^{2/3}\right)^{3/2},$$

where we used $(i)$ Lemma 23 of [12] and $\mathcal{I}_{r,t} \subseteq [t]$, $(ii)$ the reverse Hölder's inequality with $u_i = \sqrt{A_i}$, and $v_i = 1/\sqrt{A_i\lambda_i}$, and $(iii)$ the bound (12). Rearranging, we have
$$A_t^{1/3} \geq \frac{r^{2/3}}{4\sqrt[3]{6}R^{2/3}}\left(\sum_{i \in \mathcal{I}_{r,t}}A_i^{1/3}\right), \quad \text{for all } t \leq k, \tag{13}$$

which by applying Lemma 32 of [12] and noting that $T_r = |\mathcal{I}_{r,k}|$ gives
$$A_k^{1/3} \geq \exp\left(\frac{r^{2/3}}{4\sqrt[3]{6}R^{2/3}}T_r\right)A_0^{1/3},$$

and thus
$$\frac{1}{A_k}\left(9R^2 - \frac{1}{6}\sum_{i \leq k}\lambda_i A_i\|\hat{x}_i - y_{i-1}\|^2\right) \leq \frac{9R^2}{A_k} \leq O\left(\frac{R^2}{A_0}\exp\left(-\Omega(1)\frac{r^{2/3}}{R^{2/3}}T_r\right)\right).$$

Next, we show that
$$\frac{1}{A_k}\left(9R^2 - \frac{1}{6}\sum_{i \leq k}\lambda_i A_i\|\hat{x}_i - y_{i-1}\|^2\right) \leq O\left(\frac{\lambda_{\min}R^2}{T_\lambda^2}\right). \tag{14}$$

Using Lemma 23 of [12] again, along with $\mathcal{I}_\lambda \subseteq [k]$ and $|\mathcal{I}_\lambda| = T_\lambda$, we have
$$\sqrt{A_k} \geq \frac{1}{2}\sum_{i \in \mathcal{I}_\lambda}\frac{1}{\sqrt{\lambda_i}} \geq \frac{T_\lambda}{2\sqrt{2\lambda_{\min}}}.$$

Rearranging the terms, we see that $1/A_k \leq O(\lambda_{\min}/T_\lambda^2)$ as desired.

Combining Equations (11) and (14) we obtain the claimed bound. $\qquad \square$

Putting these pieces together gives Proposition 2, which we prove below.

**Proposition 2.** *Let $f : \mathcal{X} \to \mathbb{R}$ and $\hat{\nabla} f$ satisfy Assumption 1, and let $\mathcal{X} \subseteq \mathbb{B}_R(x_0)$. For a target accuracy $\epsilon \leq GR$ let $\varphi_{k+1} = \frac{\epsilon}{60\lambda_{k+1}a_{k+1}}$, $\delta_{k+1} = \frac{\epsilon}{120R}$, $\sigma_{k+1}^2 = \frac{\epsilon}{60a_{k+1}}$, $A_0 = \frac{R}{G}$, and $A_{\max} = \frac{9R^2}{\epsilon}$. If $\lambda_k \geq \lambda_{\min} \geq \frac{1}{A_{\max}} = \Omega(\frac{\epsilon}{R^2})$ for all $k \leq K_{\max}$, then lines 4 and 5 of Algorithm 4 have total complexity $\mathbb{E}\mathcal{N}_{\hat{\nabla} f} = O\left(K_{\max} \log \frac{GR}{\epsilon} + \frac{G^2R^2}{\epsilon^2} \log^2 \frac{GR}{\epsilon}\right)$. If in addition $\|\mathsf{P}_{f,\lambda_k}(y_{k-1}) - y_{k-1}\| \geq 3r/4$ whenever $\lambda_k \geq 2\lambda_{\min}$ then for $K_{\max} = O\left(\left(\frac{R}{r}\right)^{2/3} \log\left(\frac{GR}{\epsilon}\right) + \sqrt{\frac{\lambda_{\min}R^2}{\epsilon}}\right)$, the algorithm's output $x_K$ satisfies $f(x_K) - f(x_\star) \leq \epsilon$ with probability at least $\frac{2}{3}$.*

*Proof.* First, let us prove correctness of the algorithm. The settings of $\varphi_k$, $\delta_k$ and $\sigma_k$ in the proposition guarantee that

$$\max_{k \leq K_{\max}} \left\{\lambda_k a_k \varphi_k + a_k \sigma_k^2 + 2R\delta_k\right\} = \frac{\epsilon}{20}.$$

Therefore, Lemma 6 with $\bar{\varepsilon} = \epsilon/20 \leq R^2/(2.2A_{\max})$ yields

$$\mathbb{E}\left[A_K(f(x_K) - f(x_\star)) + \frac{1}{6}\sum_{i \leq K} \lambda_i A_i \|\hat{x}_i - y_{i-1}\|^2\right] \leq R^2 + \bar{\varepsilon} \cdot \mathbb{E}A_K + A_0(f(x_0) - f(x_\star)).$$

Note that $A_{K-1} \leq A_{\max}$ by definition. Therefore, $\lambda_{\min} \geq \frac{1}{A_{\max}}$ implies that

$$a_K = \frac{1}{2}\sqrt{\frac{1}{\lambda_K^2} + \frac{4A_{K-1}}{\lambda_K}} \leq \frac{\sqrt{5}}{2} A_{\max},$$

and therefore $A_K \leq 2.2A_{\max} \leq R^2/\bar{\varepsilon}$ with probability 1. Moreover the choice of $A_0 = R/G$ and the fact that $f$ is $G$ Lipschitz imply that $A_0(f(x_0) - f(x_\star)) \leq R^2$. Therefore, $\mathbb{E}\left[A_K(f(x_K) - f(x_\star)) + \frac{1}{6}\sum_{i \leq K} \lambda_i A_i \|\hat{x}_i - y_{i-1}\|^2\right]$ is at most $3R^2$. Since the term in the expectation is non-negative, we conclude that with probability at least $2/3$ it is bounded by $9R^2$, which implies

$$f(x_K) - f(x_\star) \leq \frac{1}{A_K}\left(9R^2 - \frac{1}{6}\sum_{i \leq K} \lambda_i A_i \|\hat{x}_i - y_{i-1}\|^2\right).$$

If $A_K \geq A_{\max} = 9R^2/\epsilon$ we are done. Otherwise, $K = K_{\max}$ and by the assumption on NEXTLAMBDA we have $T_\lambda + T_r \geq K_{\max}$ for $T_\lambda$ and $T_r$ defined in Lemma 7. Therefore, either $T_r \geq K_{\max}/2$ or $T_\lambda \geq K_{\max}/2$, and in either case taking $K_{\max} = O\left(\left(\frac{R}{r}\right)^{2/3} \log\left(\frac{GR}{\epsilon}\right) + \sqrt{\frac{\lambda_{\min}R^2}{\epsilon}}\right)$ and applying Lemma 7 yields $f(x_K) - f(x_\star) \leq \epsilon$ and establishing correctness.

Next, let us prove the stated complexity bound. We note each step of computing $x_k$ in Line 4 requires $O(G^2/\lambda_k\varphi_k)$ queries via Proposition 3 and the definition (4) of the approximate proximal mapping. Moreover, by Corollary 2 computing $g_k$ in Line 5 requires

$$O\left(\log\left(\frac{G}{\min\{\delta_k, \sigma_k\}}\right) + \frac{G^2}{\sigma_k^2} \log^2\left(\frac{G}{\min\{\delta_k, \sigma_k\}}\right)\right)$$

queries in expectation. Summing over $k \in [K]$ and substituting $\varphi_k, \delta_k, \sigma_k$, we obtain

$$\mathbb{E}\mathcal{N}_{\hat{\nabla} f} = \sum_{k \in [K]} O\left(\frac{G^2}{\lambda_k\varphi_k}\right) + \sum_{k \in [K]} O\left(\log\left(\frac{G}{\min\{\delta_k, \sigma_k\}}\right) + \frac{G^2}{\sigma_k^2} \log^2\left(\frac{G}{\min\{\delta_k, \sigma_k\}}\right)\right)$$

$$= \sum_{k \in [K]} O\left(\log\left(\frac{GR}{\epsilon}\right) + \frac{a_k G^2}{\epsilon} \log^2\left(\frac{GR}{\epsilon}\right)\right) = O\left(\log\left(\frac{GR}{\epsilon}\right) \cdot K + \frac{A_K G^2}{\epsilon} \log^2\left(\frac{GR}{\epsilon}\right)\right)$$

$$= O\left(K_{\max} \log\left(\frac{GR}{\epsilon}\right) + \frac{G^2R^2}{\epsilon^2} \log^2\left(\frac{GR}{\epsilon}\right)\right),$$

where we have used $A_K = O(A_{\max}) = O(R^2/\epsilon)$ once more. $\qquad\square$

## D.2 Minimizing the maximum of $N$ functions

In this section, we first revisit the problem setup of minimizing the maximum of $N$ functions and reintroduce key notation. Then we provide the procedure of estimating the gradient of the softmax using rejection sampling in Algorithm 9 and prove its guarantees in Lemma 8. Next, we bound the query complexity of lines 4 and 5 of Algorithm 4 in Lemmas 9 and 10 respectively. Citing [13], we provide a bisection procedure in Algorithm 10 and state its guarantee in Lemma 4. For this procedure we use the Ball Regularization Optimization Oracle (BROO) implementation of [13]; see Definition 4 and Lemma 11. Combining these components with the developments of the previous subsection, we prove Theorem 4.

**Notation.** Consider the problem of approximately minimizing the maximum of $N$ convex functions: given $f_{(i)}$ such that for every $i \in [N]$ the function $f_{(i)} : \mathbb{R}^d \to \mathbb{R}$ is convex, $G$-Lipschitz, with a subgradient oracle $\nabla f_{(i)}$ and a target accuracy $\epsilon$ we wish to

$$\text{find a point } x \text{ such that } f_{\max}(x) - \inf_{x_\star \in \mathbb{R}^d} f_{\max}(x_\star) \leq \epsilon \text{ where } f_{\max}(x) := \max_{i \in [N]} f_{(i)}(x) . \quad (15)$$

A common approach to solving this problem is to consider the following "softmax" approximation of $f_{\max}$,

$$f_{\text{smax}}(x) := \epsilon' \log \left( \sum_{i \in [N]} e^{f_{(i)}(x)/\epsilon'} \right), \quad \text{where } \epsilon' = \frac{\epsilon}{2 \log N}. \quad (16)$$

It is straightforward to show that $0 \leq f_{\text{smax}}(x) - f_{\max}(x) \leq \frac{\epsilon}{2}$ for all $x \in \mathbb{R}^d$, and that the subgradients of $f_{\text{smax}}$ are of the form

$$\nabla f_{\text{smax}}(x) = \sum_{i \in [N]} p_i(x) \nabla f_{(i)}(x) \quad \text{where } p_i(x) = \frac{e^{f_{(i)}(x)/\epsilon'}}{\sum_{j \in [N]} e^{f_{(j)}(x)/\epsilon'}} \quad (17)$$

for $\nabla f_{(i)}(x) \in \partial f_{(i)}(x)$ for all $i \in [N]$. The small radius

$$r_\epsilon := \frac{\epsilon'}{G} = \frac{\epsilon}{2G \log N}$$

plays a key role in our analysis, since—as we now discuss in detail—this is a domain size where we can efficiently minimize $f_{\text{smax}}$ using stochastic gradient methods.

### D.2.1 Gradient estimation via rejection sampling

We first construct the gradient estimator of $f_{\text{smax}}(x)$ using rejection sampling. The high-level idea of the technique is as follows. Given a ball $\mathbb{B}_{r_\epsilon}(\bar{x})$ where $r_\epsilon = \epsilon'/G$, Lipschitz continuity of $f_{(i)}$ implies

$$\frac{\left| f_{(i)}(x) - f_{(i)}(\bar{x}) \right|}{\epsilon'} \leq \frac{G r_\epsilon}{\epsilon'} = 1. \quad (18)$$

As a result, we can perform a full data pass *once* to compute $p(\bar{x})$, and use it to sample from $p(x)$ at nearby points $x \in \mathbb{B}_{r_\epsilon}(\bar{x})$ via rejection sampling. In particular, we draw $i$ from $p(\bar{x})$ and accept it with probability $q_{\text{accept}} = \exp(f_{(i)}(x)/\epsilon' - f_{(i)}(\bar{x})/\epsilon' - 1)$, and otherwise repeat the process. The the bound (18) guarantees that $q_{\text{accept}} < 1$ (so it is indeed a probably), and therefore the output $i$ has distribution $p$. The bound (18) also guarantees that $q_{\text{accept}} = \Omega(1)$ and consequently that the query complexity of the procedure is $O(1)$. We sate the procedure formally in Algorithm 9 and give its guarantees in Lemma 8.

**Lemma 8** (Rejection sampling). *Given $G$-Lipschitz functions $f_{(i)}$ and $\bar{p} = p(\bar{x})$, $\forall i \in [N]$, the procedure SOFTMAXGRADEST with input $x \in \mathbb{B}_{r_\epsilon}(\bar{x})$ returns a vector $\hat{\nabla} f_{\text{smax}}(x)$ such that $\mathbb{E} \hat{\nabla} f_{\text{smax}}(x) \in \partial f_{\text{smax}}(x)$ and $\|\hat{\nabla} f_{\text{smax}}(x)\| \leq G$. The procedure has complexity $\mathbb{E} \mathcal{N}_{f_{(i)}} = O(1)$ and $\mathcal{N}_{\partial f_{(i)}} = 1$.*

*Proof.* We first prove correctness. Note that $G$-Lipschitz continuity of the $f_{(i)}$'s along with $\|x - \bar{x}\| \leq r_\epsilon = \epsilon'/G$ guarantees that, $f_{(i)}(x)\epsilon' - f_{(i)}(\bar{x})/\epsilon' \leq 1$ and therefore $q_{\text{accept}} \leq 1$ is a valid probability of every value of $i$. Therefore, the probability of sample and accepting $i$ is proportional to

$$\bar{p}_i \cdot \exp \left( \frac{f_{(i)}(x)}{\epsilon'} - \frac{f_{(i)}(\bar{x})}{\epsilon'} \right) \propto \exp \left( \frac{f_{(i)}(x)}{\epsilon'} \right) \propto p_i(x),$$

---

**Algorithm 9:** SOFTMAXGRADEST($\{f_{(i)}\}, \{\bar{p}_i\}, \bar{x}, x$)

---

**Input:** Functions $f_{(i)}$, pre-computed $f_{(i)}(\bar{x})$ and $\bar{p}_i = p_i(\bar{x})$ for $i \in [N]$, query point
$\quad\quad x \in \mathbb{B}_{r_\epsilon}(\bar{x})$.
**Output:** An unbiased estimator for $\nabla f_{\mathrm{smax}}$ with norm at most $G$.

**1 Loop**
**2** $\quad$ Sample $i$ from $\bar{p}$
**3** $\quad$ Let $q_{\mathrm{accept}} = \exp(f_{(i)}(x)/\epsilon' - f_{(i)}(\bar{x})/\epsilon' - 1)$
**4** $\quad$ Draw $A \sim$ Bernoulli($q_{\mathrm{accept}}$)
**5** $\quad$ **if** $A = 1$ **then return** $\nabla f_{(i)}(x)$

---

which proves the correctness of the sampling distribution for the output $i$ and, via eq. (17), the unbiasedness of the gradient estimator. The norm bound on the output of the procedure is immediate from Lipschitzness of $f_{(i)}$.

Next, we prove the complexity bound. Clearly, the algorithm only queries a single subgradient at termination. To bound the number of function value queries, note that Lipschitz continuity and the ball radius imply $f_{(i)}(x)\epsilon' - f_{(i)}(\bar{x})/\epsilon' \geq -1$, and therefore the probability of acceptance is at least $e^{-2}$. Consequently, the expected number of iterations before accepting a sample is at most $e^2 = O(1)$. $\quad\square$

### D.2.2 Estimating the proximal mapping and Moreau envelope gradient

Using gradient estimator for $\hat{\nabla} f_{\mathrm{smax}}$ developed above, we can implement lines 4 and 5 in Algorithm 4, provided that the true proximal bound $\hat{x} = \mathsf{P}_{\lambda, f_{\mathrm{smax}}}(y)$ satisfies $\|\hat{x} - y\| \leq r = r_\epsilon$. We begin the implementation of the approximate proximal step in line 4, which we obtain by directly applying EPOCHSGD. The following is an immediate consequence of Lemma 8 and Proposition 3.

**Lemma 9.** *Let $f_{(i)}$ be convex and $G$-Lipschitz for all $i \in [N]$, let $\epsilon, \varphi > 0$ and $r_\epsilon = \epsilon/(2\log GN)$. For any $\bar{x} \in \mathbb{R}^d$ and $\lambda > 0$, if $\mathsf{P}_{f_{\mathrm{smax}}, \lambda}(\bar{x}) \in \mathbb{B}_{r_\epsilon}(\bar{x})$ then EPOCHSGD($\hat{\nabla} f_{\mathrm{smax}}, \frac{\lambda}{2}\|\cdot - \bar{x}\|, \lambda, \mathcal{X} \cap \mathbb{B}_{r_\epsilon}(\bar{x}), \lceil 16G^2/(\lambda\varphi) \rceil$) (with $\hat{\nabla} f_{\mathrm{smax}}$ implemented with Algorithm 9) outputs a valid point $\widetilde{\mathsf{P}}^\varphi_{f_{\mathrm{smax}}, \lambda}(\bar{x})$, and has complexity*

$$\mathbb{E}\mathcal{N}_{f_{(i)}} = O\left(N + \frac{G^2}{\lambda\varphi}\right) \quad \text{and} \quad \mathcal{N}_{\partial f_{(i)}} = O\left(\frac{G^2}{\lambda\varphi}\right).$$

Similarly combining Lemma 1 with Corollary 2, one can also obtain the following expected oracle complexity guarantee for estimating the Moreau envelope gradient .

**Lemma 10.** *Let $f_{(i)}$ be convex and $G$-Lipschitz for all $i \in [N]$, let $\sigma, \epsilon, \delta > 0$ and $r_\epsilon = \epsilon/(2\log N \cdot G)$. For any $\bar{x} \in \mathbb{R}^d$ and $\lambda > 0$, if $\mathsf{P}_{f_{\mathrm{smax}}, \lambda}(\bar{x}) \in \mathbb{B}_{r_\epsilon}(\bar{x})$ then $\hat{g} = $ MORGRADEST($\hat{\nabla} f_{\mathrm{smax}}, \lambda, \bar{x}, \delta, \sigma^2, \mathcal{X} \cap \mathbb{B}_{r_\epsilon}(\bar{x})$) (with $\hat{\nabla} f_{\mathrm{smax}}$ implemented with Algorithm 9) is an estimator of the Moreau envelope gradient $\nabla f_{\mathrm{smax}, \lambda}(\bar{x})$ with bias at most $\delta$ and expected square error at most $\sigma^2$. Its complexity is*

$$\mathbb{E}\mathcal{N}_{f_{(i)}} = O\left(N + \frac{G^2}{\sigma^2}\log^2\left(\frac{G}{\min\{\delta, \sigma\}}\right) + \log\left(\frac{G}{\min\{\delta, \sigma\}}\right)\right)$$

$$\mathbb{E}\mathcal{N}_{\partial f_{(i)}} = O\left(\frac{G^2}{\sigma^2}\log^2\left(\frac{G}{\min\{\delta, \sigma\}}\right) + \log\left(\frac{G}{\min\{\delta, \sigma\}}\right)\right).$$

### D.2.3 Implementing NEXTLAMBDA via bisection

The third and final component in our algorithm is an implementation of the subroutine NEXTLAMBDA in line 2 of Algorithm 4 that guarantees the following things on $\lambda_{k+1}$ and $\hat{x}_{k+1} = \mathsf{P}_{f_{\mathrm{smax}}, \lambda}(y_k)$: $(i)$ that $\|\hat{x}_{k+1} - y_k\| \leq r$ and $(ii)$ either $\|\hat{x}_{k+1} - y_k\| \geq 3r/4$ or $\lambda < 2\lambda_{\min}$; we later set $r = r_\epsilon$ and $\lambda_{\min} = \widetilde{O}(\epsilon/(r_\epsilon^{4/3} R^{2/3}))$ but for the development of the bisection procedure we keep them general.

Our implementation of NEXTLAMBDA is identical to the one in [13], and we reproduce it here for completeness.

We start by introducing the notion of a *Ball Regularization Optimization Oracle* (BROO).

**Definition 4** ([13, Definition 1]). *We say that a mapping $\mathcal{O}_{\lambda,\rho}(\cdot) : \mathcal{X} \to \mathcal{X}$ is a Ball Regularized Optimization Oracle of radius $r$ ($r$-BROO) for $f$, if for every query point $\bar{x}$, regularization parameter $\lambda$ and desired accuracy $\rho$, it return $\tilde{x} = \mathcal{O}_{\lambda,\rho}(\bar{x})$ satisfying*

$$f(\tilde{x}) + \frac{\lambda}{2}\|\tilde{x} - \bar{x}\|^2 \leq \min_{x \in \mathbb{B}_r(\bar{x}) \cap \mathcal{X}} \left\{ f(x) + \frac{\lambda}{2}\|x - \bar{x}\|^2 \right\} + \frac{\lambda}{2}\rho^2. \tag{19}$$

While a BROO is quite similar to the approximate proximal mapping $\widetilde{\mathsf{P}}_\lambda^\varphi$, there are two important differences. First, in the BROO definition we constrain the minimization to $\mathbb{B}_r(\bar{x})$ where the approximate proximal mapping is defined for the all domain—this allows us to efficiently compute a BROO via stochastic methods even for values of $\lambda$ where the true (unconstrained) proximal point is far from $\bar{x}$. Second, we require the sub-optimality guarantee to hold deterministically (a requirement that we will satisfy with high probability), as opposed the requirement (4) of an expected suboptimality bound. In addition, note that the accuracy parameter $\varphi$ and $\rho$ are related via $\varphi = \lambda\rho^2/2$ and that $\rho$ has units of distance. Strong convexity of the BROO optimization objective then implies that $\|\mathcal{O}_{\lambda,\rho}(x) - \mathsf{P}_{f,\lambda}(x)\| \leq \rho$ whenever $\mathsf{P}_{f,\lambda}(x) \in \mathbb{B}_r(x)$.

We have the following high-probability complexity guarantee for implementing a BROO.

**Lemma 11** ([13, Corollary 1]). *Let $f_{(i)}$ be convex and $G$-Lipschitz for all $i \in [N]$, let $p_f \in (0,1)$, $\epsilon, \rho > 0$ and $r_\epsilon = \epsilon/(2\log N \cdot G)$. For any $\bar{x} \in \mathbb{R}^d$ and $\lambda \leq O(G/r_\epsilon)$, with probability at least $1 - p_f$, EPOCH-SGD-PROJ [13, Algorithm 2] that outputs a valid $r_\epsilon$-BROO response for $f_{\mathrm{smax}}$ to query $\bar{x}$ with regularization $\lambda$ and accuracy $\rho$, and has complexity*

$$\mathcal{N}_{f_{(i)}} = O\left(N + \frac{G^2}{\lambda^2\rho^2}\log\left(\frac{\log(G/(\lambda\rho))}{p_f}\right)\right) \text{ and } \mathcal{N}_{\partial f_{(i)}} = O\left(\frac{G^2}{\lambda^2\rho^2}\log\left(\frac{\log(G/(\lambda\rho))}{p_f}\right)\right) \tag{20}$$

Given a BROO implementation Algorithm 10 outputs values of $\lambda$ meeting the requirements of Proposition 2. The algorithm and the formal guarantee below are reproduced from [13] for completeness, and we refer the reader to Appendix B.3 of that paper for additional description and discussion.

---

**Algorithm 10: $\lambda$-BISECTION$(x, v, A)$**

**Input:** Points $x, v \in \mathcal{X}$, scalar $A \geq 0$.
**Parameters :** BROO $\mathcal{O}_{\lambda,\delta}(\cdot)$ (see Definition 4), bisection bounds $\lambda_{\min}, \lambda_{\max}$, Lipschitz bound $G$, distance bounds $R$ and $r$.

1 For all $\lambda'$, let $y_{\lambda'} := \alpha_{2A\lambda'} \cdot x + (1 - \alpha_{2A\lambda'}) \cdot v$, where $\alpha_\tau := \frac{\tau}{1+\tau+\sqrt{1+2\tau}}$
2 Define $\Delta(\lambda) := \|\mathcal{O}_{\lambda,\frac{r}{17}}(y_\lambda) - y_\lambda\|$      ▷ approximation of ball optimizer to $y_\lambda$
3 Let $\lambda = \lambda_{\max}$
4 **while** $\lambda \geq \lambda_{\min}$ **and** $\Delta(\lambda) \leq \frac{13r}{16}$ **do** $\lambda \leftarrow \lambda/2$   ▷ terminates in $O(\log\frac{\lambda_{\max}}{\lambda_{\min}})$ steps
5 **if** $\lambda \leq \lambda_{\min}$ **then return** $2\lambda$    ▷ happens only if ball optimizer is $O(\epsilon)$-optimal
6 Let $\lambda_u = 2\lambda$, $\lambda_\ell = \lambda$ and $\lambda_m = \sqrt{\lambda_u\lambda_\ell}$
7 **if** $\Delta(\lambda_\ell) \leq \frac{15r}{16}$ **then return** $\lambda_\ell$     ▷ happens only if $\Delta(\lambda_\ell) \in [\frac{13r}{16}, \frac{15r}{16}]$
8 **while** $\Delta(\lambda_m) \notin [\frac{13r}{16}, \frac{15r}{16}]$ **and** $\log_2 \frac{\lambda_u}{\lambda_\ell} \geq \frac{r}{8(R+G/\lambda_\ell)}$ **do**
9   **if** $\Delta(\lambda_m) < \frac{13r}{16}$ **then** $\lambda_u = \lambda_m$ **else** $\lambda_\ell = \lambda_m$
10   $\lambda_m = \sqrt{\lambda_u\lambda_\ell}$
11 **return** $\lambda_m$      ▷ the while loop terminates in $O\left(\log\left(\frac{R}{r} + \frac{G}{\lambda_{\min}r}\right)\right)$ steps

---

**Proposition 4** ([13, Proposition 2]). *Let $f : \mathbb{R}^d \to \mathbb{R}$ be $G$-Lipschitz and convex, and let $x, v \in \mathbb{R}^d$, $\epsilon, r, R \in \mathbb{R}_{>0}$ satisfy $\epsilon \leq GR$, $r \leq R$ and $\|x - v\| \leq 2R$. Given $\lambda_{\max} \geq \frac{2G}{r}$ and $\lambda_{\min} \in (0, \lambda_{\max})$, $\lambda$-BISECTION$(x, v, A)$ outputs $\lambda \in [\lambda_{\min}, \lambda_{\max}]$ such that*

$$\|\mathsf{P}_{f,\lambda}(y_\lambda) - y_\lambda\| \leq r.$$

*The subroutine uses $O(\log(\frac{\lambda_{\max}}{\lambda_{\min}}) + \log(\frac{R+G/\lambda_{\min}}{r}))$ calls to $\mathcal{O}_{\lambda', \frac{r}{17}}(\cdot)$ with $\lambda' \in [\frac{1}{2}\lambda, \lambda_{\max}]$. Moreover, for $\alpha_{2\lambda A} = \frac{2\lambda A}{1+2\lambda A+\sqrt{1+4\lambda A}}$ and $y_\lambda := \alpha_{2\lambda A}x + (1-\alpha_{2\lambda A})v$ one of the following outcomes must occur:*

*1. $\lambda \in [2\lambda_{\min}, \lambda_{\max}]$ and $\|\mathsf{P}_{f,\lambda}(y_\lambda) - y_\lambda\| > \frac{3r}{4}$, or*

*2. $\lambda < 2\lambda_{\min}$.*

*When taking $\lambda_{\max} = \frac{2G}{\epsilon}$ and $\lambda_{\min} = \Omega(\frac{\epsilon}{rR})$, the number of calls to $\mathcal{O}_{\lambda', \frac{r}{17}}(\cdot)$ is $O(\log \frac{GR^2}{r\epsilon})$.*

### D.2.4   Proof of Theorem 4

Finally, we combine the guarantees collected above to prove our near-optimal rate for minimizing the maximum-loss.

**Theorem 4.** *Let $f_{(1)}, \ldots, f_{(N)} : \mathcal{X} \to \mathbb{R}$ be convex and G-Lipschitz and let $\mathcal{X} \subseteq \mathbb{B}_R(x_0)$. For any $\epsilon < \frac{1}{2}GR/\log N$, Algorithm 4 (with $\widetilde{\mathsf{P}}^\varphi_{f_{\mathrm{smax}},\lambda}$ implemented in Algorithm 8, $\hat{\nabla} f_{\mathrm{smax}}$ given by Algorithm 9 and NextLambda given by Algorithm 10 with $\lambda_{\min} = \widetilde{O}(\epsilon/(r_\epsilon^{4/3}R^{2/3}))$) outputs $x \in \mathcal{X}$ that with probability at least $\frac{1}{2}$ is $\epsilon$-suboptimal for $f_{\max}(x) = \max_{i\in[N]} f_{(i)}(x)$ and has complexity*

$$\mathbb{E}\mathcal{N}_{f_{(i)}} = O\left(\left[N\left(\frac{GR\log N}{\epsilon}\right)^{2/3} + \left(\frac{GR}{\epsilon}\right)^2\right]\log^2\frac{GR}{\epsilon}\right) \text{ and } E\mathcal{N}_{\partial f_{(i)}} = O\left(\left(\frac{GR}{\epsilon}\right)^2\log^2\frac{GR}{\epsilon}\right).$$

*Proof.* We first prove correctness. Since $\epsilon' = \frac{\epsilon}{2\log N}$, we have $0 \le f_{\mathrm{smax}}(x) - f_{\max}(x) \le \epsilon/2$ [see, e.g., 12, Lemma 45]. Therefore, it suffices to find an $\epsilon/2$-approximate solution of $f_{\mathrm{smax}}$ over the domain $\mathcal{X} \subseteq \mathbb{B}_R(x_0)$. Let $p_{\mathrm{BROO}}$ be the probability that all the BROO calls within Algorithm 10 (implemented as described in Lemma 11) result in a valid output. Then, noting that $y_\lambda$ defined in Proposition 4 is precisely $y_k$ defined in Algorithm 4, the guarantees of Proposition 4 imply that, for $\hat{x}_{k+1} = \mathsf{P}_{f_{\mathrm{smax}},\lambda}(y_k)$, we have $\|\hat{x}_{k+1} - y_k\| \le r$ and either $\|\hat{x}_{k+1} - y_k\| \ge 3r/4$ or $\lambda < 2\lambda_{\min}$ with probability at least $p_{\mathrm{BROO}}$. Consequently, Proposition 2 (with $\epsilon \to \epsilon/2$ and $K_{\max}$ as required by the proposition), the output $x$ of Algorithm 4 satisfies $f_{\mathrm{smax}}(x) - f_{\mathrm{smax}}(x^\star) \le \epsilon/2$ with probability at least $1 - (1-\frac{2}{3}) - (1 - p_{\mathrm{BROO}}) = p_{\mathrm{BROO}} - \frac{1}{3}$.

To finish the proof of correctness, it remains to verify that $p_{\mathrm{BROO}} \ge 5/6$. To that end, let $K_{\max}^{\mathrm{bisect}} = O(\log \frac{GR^2}{r_\epsilon\epsilon})$ to be the total number of BROO calls in a single execution of $\lambda$-Bisection, as per Proposition 4. Then, if the probability of failure of a single BROO implementation is $p_{\mathrm{f}}$ and we perform at most $K_{\max}$ calls to $\lambda$-Bisection, we have $p_{\mathrm{BROO}} \ge 1 - K_{\max}K_{\max}^{\mathrm{bisect}}p_{\mathrm{f}}$. Therefore, taking

$$p_{\mathrm{f}} \le \frac{1}{6K_{\max}K_{\max}^{\mathrm{bisect}}}$$

guarantees correctness.

We now proceed to bound the algorithm's complexity. To that end, we set

$$\lambda_{\min} = \frac{\epsilon}{r_\epsilon^{4/3}R^{2/3}}\log^2\left(\frac{GR}{\epsilon}\right).$$

Recalling that $r_\epsilon = \frac{\epsilon}{2G\log N}$, the total number of iterations in Algorithm 4 is at most

$$K_{\max} = O\left(\left(\frac{R}{r_\epsilon}\right)^{2/3}\log\frac{GR}{\epsilon} + \sqrt{\frac{\lambda_{\min}R^2}{\epsilon}}\right) = O\left(\left(\frac{GR\log N}{\epsilon}\right)^{2/3}\log\frac{GR}{\epsilon}\right). \qquad (21)$$

Setting the approximation parameters to be $\varphi_k = O(\frac{\epsilon}{\lambda_k a_k})$, $\delta_k = O(\frac{\epsilon}{R})$ and $\sigma_k^2 = O(\frac{\epsilon}{a_k})$ as required in Proposition 2, the complexity of lines 4 and 5 in the $k$th iteration of Algorithm 4 is bounded by

Lemma 9 and Lemma 10 as

$$\mathbb{E}\mathcal{N}_{f_{(i)}}^{(k),1} = O\left(N + \frac{G^2}{\lambda_k \varphi_k} + \frac{G^2}{\sigma_k^2}\log^2\left(\frac{G}{\min\{\delta_k, \sigma_k\}}\right) + \log\left(\frac{G}{\min\{\delta_k, \sigma_k\}}\right)\right)$$

$$= O\left(N + \frac{G^2 a_k}{\epsilon}\log^2\left(\frac{GR}{\epsilon}\right) + \log\left(\frac{GR}{\epsilon}\right)\right)$$

$$\mathbb{E}\mathcal{N}_{\partial f_{(i)}}^{(k),1} = O\left(\frac{G^2}{\lambda_k \varphi_k} + \frac{G^2}{\sigma_k^2}\log^2\left(\frac{G}{\min\{\delta_k, \sigma_k\}}\right) + \log\left(\frac{G}{\min\{\delta_k, \sigma_k\}}\right)\right)$$

$$= O\left(\frac{G^2 a_k}{\epsilon}\log^2\left(\frac{GR}{\epsilon}\right) + \log\left(\frac{GR}{\epsilon}\right)\right).$$

To bound the complexity of the bisection procedure at the $k$th iteration of Algorithm 4, note that it makes a total of $K_{\max}^{\text{bisect}} = O(\log\frac{GR^2}{r_\epsilon\epsilon}) = O(\log\frac{GR\log N}{\epsilon}) = O(\log\frac{GR}{\epsilon})$ BROO calls, $r_\epsilon = \frac{\epsilon}{2G\log N}$ and $\log N \leq \frac{GR}{2\epsilon}$. Applying Lemma 11 with $p_{\text{f}}$ and $\lambda_{\min}$ as determined above, the complexity is bounded by

$$\mathcal{N}_{f_{(i)}}^{(k),2} = O\left(\left(N + \frac{G^2}{\lambda_{\min}^2 r_\epsilon^2}\log\left(\frac{\log(G/(\lambda_{\min}r_\epsilon))}{p_{\text{f}}}\right)\right)K_{\max}^{\text{bisect}}\right)$$

$$= O\left(\left(N + \frac{G^2 r_\epsilon^{2/3}R^{4/3}}{\epsilon^2\log^4\left(\frac{GR}{\epsilon}\right)}\log\left(\frac{GR}{\epsilon}\right)\right)\log\left(\frac{GR}{\epsilon}\right)\right),$$

$$\mathcal{N}_{\partial f_{(i)}}^{(k),2} = O\left(\frac{G^2}{\lambda_{\min}^2 r_\epsilon^2}\log\left(\frac{\log(G/(\lambda_{\min}r_\epsilon))}{p_{\text{f}}}\right)K_{\max}^{\text{bisect}}\right) = O\left(\frac{G^2 r_\epsilon^{2/3}R^{4/3}}{\epsilon^2\log^2\left(\frac{GR}{\epsilon}\right)}\right).$$

Summing the bounds above over iterations 1 to $K \leq K_{\max}$ and noting that $\sum_{k \leq K}a_k = A_K \leq 2A_{\max} = O(R^2/\epsilon)$ (see proof of Proposition 2) we obtain the total complexity bounds

$$\mathbb{E}\mathcal{N}_{f_{(i)}} = \sum_{k \leq K}\left(\mathbb{E}\mathcal{N}_{f_{(i)}}^{(k),1} + \mathbb{E}\mathcal{N}_{f_{(i)}}^{(k),2}\right)$$

$$= O\left(K_{\max}N\log\frac{GR}{\epsilon} + \frac{G^2 A_K}{\epsilon}\log^2\frac{GR}{\epsilon} + K_{\max}\frac{G^2 r_\epsilon^{2/3}R^{4/3}}{\epsilon^2\log^4\left(\frac{GR}{\epsilon}\right)}\cdot\log^2\left(\frac{GR}{\epsilon}\right)\right)$$

$$= O\left(\left(\frac{GR\log N}{\epsilon}\right)^{2/3}N\cdot\log^2\frac{GR}{\epsilon} + \frac{G^2 R^2}{\epsilon^2}\log^2\frac{GR}{\epsilon}\right),$$

and

$$\mathbb{E}\mathcal{N}_{\partial f_{(i)}} = \sum_{k \leq K}\left(\mathbb{E}\mathcal{N}_{\partial f_{(i)}}^{(k),1} + \mathbb{E}\mathcal{N}_{\partial f_{(i)}}^{(k),2}\right) = O\left(\frac{G^2 R^2}{\epsilon^2}\log^2\frac{GR}{\epsilon}\right),$$

where we have used formula (21) for $K_{\max}$. This concludes the proof. $\qquad\square$

## E   Proofs from Section 5

In the section we prove Theorem 5, the convergence guarantee for Algorithm 5, our gradient-efficient composite optimization method. We first provide a lemma (Lemma 12) that helps us analyze the behavior of the $\beta_k$ and $\gamma_k$ sequences in the algorithm. Then we combine it with the approximation guarantees of our estimator to show the convergence rate of Algorithm 5 in Proposition 5. Finally we apply this proposition and bound the expected number of gradient queries complete the proof of Theorem 5.

The following helper lemma is also used in Lan [33, 34]; we provide it here for completeness of analysis.

**Lemma 12** (Convergence of geometric sequence, cf. Lemma 2 of Lan [34]). *Given $\gamma_k \in (0, 1)$, for all $k \in \mathbb{N}$, and $\Gamma_1 > 0$, define the sequence*

$$\Gamma_k := (1 - \gamma_k)\Gamma_{k-1}, \quad \forall k \geq 2.$$

*If a sequence $E_k$ satisfies $E_k \le (1 - \gamma_k)E_{k-1} + B_k$, for all $k \ge 1$, then we have for any $k \ge 1$,*

$$E_k \le \Gamma_k \left[ \frac{1 - \gamma_1}{\Gamma_1} E_0 + \sum_{i \in [k]} \frac{B_i}{\Gamma_i} \right].$$

Using the helper lemma, we can show the following convergence rate for Algorithm 5.

**Proposition 5** (Convergence rate). *Given problem (5) with optimizer $x^\star$ and initial point $\|x_0 - x^\star\| \le R$, let $\sigma_k^2 = \frac{R^2}{4N}$, $\delta_k = \frac{R}{16N}$, $\epsilon_k = \frac{LR^2}{2kN}$, and let parameters $\beta_k = \frac{2L}{k}$, $\gamma_k = \frac{2}{k+1}$. Then, the iterates of Algorithm 5 satisfy*

$$\Psi(x_N) - \Psi(x_\star) \le O\left( \frac{LR^2}{N^2} \right).$$

*Proof.* We first observe that

$$
\begin{aligned}
\Lambda(x_k) &\overset{(i)}{\le} \Lambda(y_k) + \langle \nabla\Lambda(y_k), x_k - y_k \rangle + \frac{L}{2}\|x_k - y_k\|^2 \\
&\overset{(ii)}{=} (1 - \gamma_k)\left[\Lambda(y_k) + \langle \nabla\Lambda(y_k), x_{k-1} - y_k \rangle\right] \\
&\quad + \gamma_k\left[\Lambda(y_k) + \langle \nabla\Lambda(y_k), \bar{v}_k - y_k \rangle\right] + \frac{L\gamma_k^2}{2}\|\bar{v}_k - \mathsf{Proj}_{\mathcal{X}}(v_{k-1})\|^2 \\
&\overset{(iii)}{\le} (1 - \gamma_k)\Lambda(x_{k-1}) + \gamma_k\left[\Lambda(y_k) + \langle \nabla\Lambda(y_k), \bar{v}_k - y_k \rangle + \frac{\beta_k}{2}\|\mathsf{Proj}_{\mathcal{X}}(v_{k-1}) - \bar{v}_k\|^2\right] \\
&\quad - \frac{\gamma_k\beta_k - L\gamma_k^2}{2}\|\mathsf{Proj}_{\mathcal{X}}(v_{k-1}) - \bar{v}_k\|^2 \\
&\overset{(iv)}{\le} (1 - \gamma_k)\Lambda(x_{k-1}) + \gamma_k\left[\Lambda(y_k) + \langle \nabla\Lambda(y_k), \bar{v}_k - y_k \rangle + \frac{\beta_k}{2}\|\mathsf{Proj}_{\mathcal{X}}(v_{k-1}) - \bar{v}_k\|^2\right],
\end{aligned}
$$

where we use $(i)$ $L$ smoothness of function $\Lambda$, $(ii)$ expanding $x_k = (1 - \gamma_k)x_{k-1} + \gamma_k\bar{v}_k$ and replacing $y_k - x_k = \gamma_k(\mathsf{Proj}_{\mathcal{X}}(v_{k-1}) - \bar{v}_k)$, $(iii)$ convexity of $\Lambda$, and $(iv)$ that $\beta_k \ge L\gamma_k$.

Similarly using convexity of the non-smooth component $f$ and the definition of $x_k$ and $\bar{v}_k$, we obtain

$$f(x_k) \le (1 - \gamma_k)f(x_{k-1}) + \gamma_k f(\bar{v}_k).$$

Thus, summing the two inequalities and recalling the definition $\bar{\Lambda}_k(v) = \Lambda(y_k) + \langle \nabla\Lambda(y_k), v - y_k \rangle$, this is equivalent to

$$\Lambda(x_k) + f(x_k) \le (1 - \gamma_k)\left(\Lambda(x_{k-1}) + f(x_{k-1})\right) + \gamma_k\left[\bar{\Lambda}_k(\bar{v}_k) + f(\bar{v}_k) + \frac{\beta_k}{2}\|\mathsf{Proj}_{\mathcal{X}}(v_{k-1}) - \bar{v}_k\|^2\right].$$

Now we recall the definition of composite objectives $\Psi(x) = \Lambda(x) + f(x)$ and define

$$\Phi_k(x) = \bar{\Lambda}_k(x) + f(x) + \frac{\beta_k}{2}\|x - \mathsf{Proj}_{\mathcal{X}}(v_{k-1})\|^2.$$

By convexity of $\Psi$ one has the recursion

$$\Psi(x_k) - \Psi(u) \le (1 - \gamma_k)\left(\Psi(x_{k-1}) - \Psi(u)\right) + \gamma_k\left(\Phi_k(\bar{v}_k) - \Phi_k(u) + \frac{\beta_k}{2}\|\mathsf{Proj}_{\mathcal{X}}(v_{k-1}) - u\|^2\right).$$

Let $v_k^\star$ be the exact minimizer of $\Phi_k$ restricted to $\bar{\mathcal{X}} := \mathbb{B}_R(v_0) \cap \mathcal{X}$. We have, for any $u \in \bar{\mathcal{X}}$, that $\Phi_k(u) \ge \Phi_k(v_k^\star) + \frac{\beta_k}{2}\|v_k^\star - u\|^2$, and consequently

$$
\begin{aligned}
\Psi(x_k) - \Psi(u) \le &(1 - \gamma_k)\left(\Psi(x_{k-1}) - \Psi(u)\right) \\
&+ \gamma_k\left(\Phi_k(\bar{v}_k) - \Phi_k(v_k^\star) + \frac{\beta_k}{2}\left(\|\mathsf{Proj}_{\mathcal{X}}(v_{k-1}) - u\|^2 - \|v_k^\star - u\|^2\right)\right).
\end{aligned}
$$

Conditioning on past events and taking expectation over randomness of $v_k$ and $\bar{v}_k$, this gives for any $u \in \mathcal{X}$,

$$
\begin{aligned}
\mathbb{E}\Psi(x_k) - \Psi(u) \leq{} & (1 - \gamma_k)\left(\Psi(x_{k-1}) - \Psi(u)\right) + \gamma_k\left(\mathbb{E}\Phi_k(\bar{v}_k) - \Phi_k(v_k^\star)\right) \\
& + \frac{\gamma_k\beta_k}{2}\left(\|\mathsf{Proj}_{\mathcal{X}}(v_{k-1}) - u\|^2 - \mathbb{E}\|v_k - u\|^2 + \mathbb{E}\|v_k - v_k^\star\|^2 + \mathbb{E}2\langle v_k^\star - u, v_k - v_k^\star\rangle\right) \\
\overset{(i)}{\leq}{} & (1 - \gamma_k)\left(\Psi(x_{k-1}) - \Psi(u)\right) + \gamma_k\left(\mathbb{E}\Phi_k(\bar{v}_k) - \Psi(v_k^\star)\right) \\
& + \frac{\gamma_k\beta_k}{2}\left(\|\mathsf{Proj}_{\mathcal{X}}(v_{k-1}) - u\|^2 - \mathbb{E}\|v_k - u\|^2 + \mathbb{E}\|v_k - v_k^\star\|^2 + 4R\|\mathbb{E}v_k - v_k^\star\|\right) \\
\overset{(ii)}{\leq}{} & (1 - \gamma_k)\left(\Psi(x_{k-1}) - \Psi(u)\right) + \gamma_k\left(\mathbb{E}\Phi_k(\bar{v}_k) - \Phi_k(v_k^\star)\right) \\
& + \frac{\gamma_k\beta_k}{2}\left(\|\mathsf{Proj}_{\mathcal{X}}(v_{k-1}) - u\|^2 - \mathbb{E}\|\mathsf{Proj}_{\mathcal{X}}(v_k) - u\|^2 + \mathbb{E}\|v_k - v_k^\star\|^2 + 4R\|\mathbb{E}v_k - v_k^\star\|\right)
\end{aligned}
$$

where we use $(i)$ the triangle inequality and $v_k^\star \in \bar{\mathcal{X}}$ to conclude $\|v_k^\star - u\| \leq \|v_k^\star - x_0\| + \|x_0 - u\| \leq 2R$, and $(ii)$ the projection property that $\|\mathsf{Proj}_{\mathcal{X}}(v_k) - u\|^2 \leq \|v_k - u\|^2$ for any $u \in \mathcal{X}$.

Note that $\mathbb{E}\Phi_k(\bar{v}_k) - \Phi_k(v_k^\star) \leq \epsilon_k$ by the definition of $\bar{v}_k = \widetilde{\mathsf{P}}^{\epsilon_k}_{\bar{\Lambda}_k + f, \beta_k}(v_{k-1})$. Moreover, Theorem 1 guarantees that $\mathbb{E}\|v_k - v_k^\star\| \leq \delta_k$ and that $\mathbb{E}\|v_k - v_k^\star\|^2 \leq \sigma_k^2$. Therefore, writing

$$
E_k = \mathbb{E}\Psi(x_k) - \Psi(u)
$$

and

$$
B_k = \frac{\gamma_k\beta_k}{2}\left(\mathbb{E}\|\mathsf{Proj}_{\mathcal{X}}(v_{k-1}) - u\|^2 - \mathbb{E}\|\mathsf{Proj}_{\mathcal{X}}(v_k) - u\|^2\right) + \gamma_k\beta_k\left(\frac{\epsilon_k}{\beta_k} + \frac{\sigma_k^2}{2} + 2R\delta_k\right),
$$

we conclude that $E_k \leq (1 - \gamma_k)E_{k-1} + B_k$. Applying Lemma 12, we obtain

$$
\begin{aligned}
\mathbb{E}\Psi(x_N) - \Psi(u) \leq{} & \Gamma_N \frac{1 - \gamma_1}{\Gamma_1}\left[\Psi(x_0) - \Psi(u)\right] \\
& + \Gamma_N \sum_{k=1}^{N} \frac{\beta_k\gamma_k}{2\Gamma_k}\left(\mathbb{E}\|\mathsf{Proj}_{\mathcal{X}}(v_{k-1}) - u\|^2 - \mathbb{E}\|\mathsf{Proj}_{\mathcal{X}}(v_k) - u\|^2\right) \\
& + \Gamma_N \sum_{k\in[N]} \frac{\beta_k\gamma_k}{\Gamma_k}\left(\frac{\epsilon_k}{\beta_k} + \frac{\sigma_k^2}{2} + 2R\delta_k\right) \\
\overset{(i)}{\leq}{} & \Gamma_N L\|v_0 - u\|^2 + \Gamma_N \sum_{k\in[N]} \frac{\beta_k\gamma_k}{\Gamma_k}\left(\frac{\epsilon_k}{\beta_k} + \frac{\sigma_k^2}{2} + 2R\delta_k\right) \overset{(ii)}{\leq} \frac{4LR^2}{N(N+1)},
\end{aligned}
$$

where $(i)$ follows from telescoping and $\gamma_1 = 1$, and $(ii)$ is due to $\Gamma_k = \prod_{k\geq 2}(1 - \gamma_k) = \frac{2}{k(k+1)}$, so that $\frac{\beta_k\gamma_k}{\Gamma_k} = 2L$, and $\frac{\epsilon_k}{\beta_k} + \frac{\sigma_k^2}{2} + 2R\delta_k \leq \frac{R^2}{2N}$ by the choices $\sigma_k^2 = \frac{R^2}{4N}$, $\delta_k = \frac{R}{16N}$ and $\epsilon_k = \frac{LR^2}{2kN}$. $\square$

We are now ready to prove the main theorem of the section.

**Theorem 5.** *Given problem* (5) *with solution* $x_\star$, *a point* $x_0$ *such that* $\|x_0 - x_\star\| \leq R$ *and target accuracy* $\epsilon > 0$, *Algorithm 5 with* $\epsilon_k = LR/2kN$, $\delta_k = R/16N$, $\sigma_k^2 = R^2/4N$, *and* $N = \Theta(\sqrt{LR^2/\epsilon})$ *finds an approximate solution* $x$ *satisfying* $\mathbb{E}\Psi(x) \leq \Psi(x_\star) + \epsilon$ *and has complexity* $\mathcal{N}_{\nabla\Lambda} = O\left(\sqrt{\frac{LR^2}{\epsilon}}\right)$ *and* $\mathbb{E}\mathcal{N}_{\hat{\nabla}f} = O\left(\left(\frac{GR}{\epsilon}\right)^2 \log^2 \frac{GR}{\epsilon} + \sqrt{\frac{LR^2}{\epsilon}} \log\left(\frac{GR}{\epsilon}\right)\right)$.

*Proof.* By Proposition 5, it suffices to run Algorithm 5 for $N = O(\sqrt{LR^2/\epsilon})$ iterations, which immediately implies the stated bound on $\mathcal{N}_{\nabla\Lambda}$.

Now we consider the cost of attaining the requiring accuracy $\epsilon_k$ when computing $\bar{v}_k$. Using the EPOCHSGD and Proposition 3 we can do so with

$$
N_k^{(1)} = O\left(\frac{G^2}{\beta_k\epsilon_k}\right) = O\left(\frac{G^2 k^2 N}{L^2 R^2}\right)
$$

queries to $\hat{\nabla} f$.

Applying Theorem 1, the expected cost of attaining bias $\delta_k = \frac{R}{16N}$ and variance $\sigma_k^2 = \frac{R^2}{4N}$ is

$$\mathbb{E} N_k^{(2)} = O\left(\log\left(\frac{GNk}{LR}\right) + \frac{NG^2}{\beta_k^2 R^2}\log^2\left(\frac{GNk}{LR}\right)\right)$$

$$= O\left(\log\left(\frac{GNk}{LR}\right) + \frac{G^2 k^2 N}{L^2 R^2}\log^2\left(\frac{GNk}{LR}\right)\right)$$

queries to $\hat{\nabla} f$.

Summing these over all $k \leq N = O(\sqrt{LR^2/\epsilon})$, we obtain the the required complexity bound

$$\mathbb{E}\mathcal{N}_{\hat{\nabla} f} = \sum_k \left(N_k^{(1)} + \mathbb{E} N_k^{(2)} + 1\right) = O\left(N\log\left(\frac{GN^2}{LR}\right) + \frac{G^2 N^4}{L^2 R^2}\log^2\left(\frac{GN^2}{LR}\right)\right)$$

$$= O\left(\sqrt{\frac{LR^2}{\epsilon}}\log\left(\frac{GR}{\epsilon}\right) + \frac{G^2 R^2}{\epsilon^2}\log^2\left(\frac{GR}{\epsilon}\right)\right).$$

$\square$

## F    Proofs and additional remarks from Section 6

In this section we prove Theorem 6 which gives an optimal complexity and generalization bound for differentially private stochastic convex optimization, conditional on the existence of an improved optimum estimator (Definition 3). We begin by stating a standard privacy guarantee for the Gaussian mechanism applied on mappings with bounded $\ell_2$ sensitivity, and a lemma that helps us bound the sensitivity of the conjunctured bounded estimator. With these results in hand, we prove Theorem 6. Finally, we discuss some challenges and prospects for constructing bounded estimators that satisfy Definition 3.

### F.1    Helper lemmas

**Privacy of the Gaussian mechanism.** In this section, we present the privacy guarantees of the Gaussian mechanism which will be useful for the proof of Theorem 6. First, for an estimator (or a function) $h : \mathbb{S}^n \to \mathbb{R}^d$, the $\ell_2$-sensitivity of the estimator is upper bounded by $\Delta$ if $\sup_{\mathcal{S},\mathcal{S}' \in \mathbb{S}^n : d_{\mathsf{ham}}(\mathcal{S},\mathcal{S}') \leq 1} \|h(\mathcal{S}) - h(\mathcal{S}')\| \leq \Delta$, where $d_{\mathsf{ham}}$ is the hamming distance between the two samples (i.e., $\mathcal{S}, \mathcal{S}'$ with hamming distance $d_{\mathsf{ham}}(\mathcal{S},\mathcal{S}') \leq 1$ have at most a single different element). We can now state the privacy guarantees of the Gaussian mechanism.

**Lemma 13** (Gaussian mechanism [21, Theorem A.1]). *Let $h : \mathbb{S}^n \to \mathbb{R}^d$ have $\ell_2$-sensitivity $\Delta$. Then the Gaussian mechanism $\mathcal{A}(\mathcal{S}) = h(\mathcal{S}) + \mathsf{N}(0, \sigma^2 I_d)$ with $\sigma = 2\Delta\log(2/\beta)/\alpha$ is $(\alpha, \beta)$-DP.*

**Bounding the number of estimator copies that use a particular sample.** To prove Theorem 6, we begin with a lemma which bounds the number of optimum estimator copies that each sample can participate in. To this end, let $S_{i,t}$ denote the set of samples used in iteration $i$ of Algorithm 6 during the computation of the $t$'th optimum estimator copy. For a sample $s_\ell$, we let $K_{i,\ell}$ denote the number of sets $S_{i,t}$ such that $z_\ell \in S_{i,t}$. Recalling that the number of iterations $k = \lceil\log n\rceil$ and that $\bar{n} = n/k$, we have the following lemma.

**Lemma 14.** *Let $\mu_i = \frac{1}{\eta_i \bar{n}}$. Assume we use an optimum oracle $\mathcal{O}$ satisfying Definition 3 with constant $C_2$ and $\delta_i^2 = \frac{G^2}{\mu_i^2 \bar{n}}$. Then, for any $\beta \leq 1/n$,*

$$\mathbb{P}\left(\max_{1 \leq i \leq k, 1 \leq \ell \leq n} K_{i,\ell} \geq 20\log(1/\beta) + 6C_2\log^2 n\right) \leq \beta/2.$$

*Proof.* We first prove the claim for a fixed $i$ and $\ell$ and then we apply a union bound. Fix $1 \leq i \leq k$ and $1 \leq \ell \leq n$ and define $Y_t = \mathbb{1}_{\{z_\ell \in S_{i,t}\}}$. Now we upper bound $p = \mathbb{P}(Y_t = 1)$. Let the random variable $N_t$ denote the number of subgradients the $t$'th query to $\mathcal{O}_\delta$ at iteration $i$ uses. First, note that whenever $N_t = j$, we have

$$\mathbb{P}(Y_t = 1 \mid N_t = j) \leq j/\bar{n},$$

by the union bound. Thus, Definition 3 now implies

$$\mathbb{P}(Y_t = 1) = \sum_{k=1}^{\infty} \mathbb{P}(Y_t = 1 \mid N_t = j)\mathbb{P}(N_t = j)$$

$$\leq \frac{1}{\bar{n}} \sum_{k=1}^{\infty} \mathbb{P}(N_t = j)j = \frac{\mathbb{E}[N]}{\bar{n}} = \frac{C_2}{\bar{n}} \log \frac{G}{\mu_i \delta_i}.$$

We can now use a Chernoff bound to prove the claim. Indeed, as $K_{i,\ell} = \sum_{t=1}^{n} Y_t$ and $Y_t \sim$ Bernoulli$(p)$ are i.i.d., Lemma 15 below implies that for $c \geq 6$,

$$\mathbb{P}(K_{i,\ell} \geq c\mathbb{E}[K_{i,\ell}]) = \mathbb{P}\left(\sum_{t=1}^{n} Y_t \geq cnp\right) \leq 2^{-cnp}.$$

As $p \leq C_2 \log(n) \log(G/\mu_i \delta_i)/n$, we take $c \geq 6$ such that $cnp \geq 20 \log(1/\beta)$, hence we have

$$\mathbb{P}(K_{i,\ell} \geq 20 \log(1/\beta) + 6C_2 \log(n) \log(G/\mu_i \delta_i))) \leq \beta^4.$$

Applying a union bound over all $n$ samples and all $k = \lceil \log n \rceil$ iterations, we have that

$$\mathbb{P}\left(\max_{1 \leq i \leq k, 1 \leq \ell \leq n} K_{i,\ell} \geq 20 \log(1/\beta) + 6C_2 \log(n) \log(G/\mu_i \delta_i))\right) \leq \beta/2.$$

The claim now follows by noting that $\frac{G}{\mu_i \delta_i} \leq \sqrt{n}$ using our choice of $\delta_i$ in Algorithm 6.

$\square$

**Lemma 15** ([37], Ch. 4.2.1). *Let* $X = \sum_{i=1}^{n} X_i$ *for* $X_i \overset{\text{iid}}{\sim}$ Bernoulli$(p)$. *Then for* $c \geq 6$,

$$P(X \geq cnp) \leq 2^{-cnp}.$$

### F.2   Proof of Theorem 6

**Theorem 6** (conditional). *Given an efficient bounded low-bias estimator* $\mathcal{O}_\delta$ *satisfying Definition 3 for any* $\delta > 0$, *then for* $\alpha \leq \log(1/\beta)$, $\mathcal{X} \in \mathbb{B}_R(x_0)$, *convex and* $G$-*Lipschitz* $\hat{f}(x; s)$, *Algorithm 6 is* $(\alpha, \beta)$-*DP, queries* $\widetilde{O}(n)$ *subgradients and has (hiding logarithmic factors in* $n$) $\mathbb{E}[f(x_k) - \min_{x \in \mathcal{X}} f(x)] \leq GR \cdot \widetilde{O}\left(\frac{1}{\sqrt{n}} + \frac{\sqrt{d \log^3(1/\beta)}}{n\alpha}\right)$.

*Proof.* We begin by proving the privacy claim. We show that each iterate is $(\alpha, \beta)$-DP which completes the proof by post-processing as each sample is used in exactly one iterate. To this end, first we show that, with high probability, each sample $z_\ell$ is used in at most $B = 20(\log(\frac{1}{\beta}) + C_2 \log^2 n)$ different optimum-estimator queries; we let $\mathfrak{E}$ denote this event. More precisely, let $S_{i,t}$ denote the set of samples used in iteration $i$ during the application of the $t$'th oracle. Then for every $i$ and sample $z_\ell$, letting $K_{i,\ell}$ be the number of sets $S_{i,t}$ such that $z_\ell \in S_{i,t}$. Using this notation, the event $\mathfrak{E}$ is equivalent to $\max_{1 \leq i \leq k, 1 \leq \ell \leq n} K_{i,\ell} \leq B$. Lemma 14 implies that $P[\mathfrak{E}] \geq 1 - \beta/2$, therefore we only have to prove $(\alpha, \beta^2/2)$-differential privacy assuming event $\mathfrak{E}$ happens as we have using $e^\alpha \leq 1/\beta$ that

$$P[\mathcal{A}(\mathcal{S}) \in \mathcal{O}] \leq P[\mathcal{A}(\mathcal{S}') \in \mathcal{O} \mid \mathfrak{E}]P[\mathfrak{E}] + (1 - P[\mathfrak{E}])$$
$$\leq e^\alpha P[\mathcal{A}(\mathcal{S}') \in \mathcal{O} \mid \mathfrak{E}]P[\mathfrak{E}] + \beta/2$$
$$\leq e^\alpha P[\mathcal{A}(\mathcal{S}') \in \mathcal{O}] + \beta.$$

We therefore assume $\mathfrak{E}$ holds and proceed to bound the $\ell_2$-sensitivity of $\tilde{x}_i$. To this end, let $\mu_i = 1/(\eta_i \bar{n})$ and $\hat{x}_i = \arg\min_{x \in \mathcal{X}} F_i(x)$. First, note that each optimum estimation oracle output satisfies

$$\|\mathcal{O}_{\delta_i}(F_i) - x_{i-1}\| \leq \|\mathcal{O}_{\delta_i}(F_i) - \hat{x}_i\| + \|\hat{x}_i - x_{i-1}\|$$
$$\overset{(\star)}{\leq} \sqrt{C_1}G\sqrt{\log n}/\mu_i + G/\mu_i$$
$$= (\sqrt{C_1 \log n} + 1)G/\mu_i,$$

where the first term in inequality $(\star)$ above holds since the estimator $\mathcal{O}_{\delta_i}$ satisfies Definition 3 and $F_i = f_i + \psi_i$ where $f_i$ is $G$-Lipschitz and $\psi_i$ is $\mu_i$-strongly convex with $G/(\mu_i\delta_i) \leq \sqrt{n}$. The second term of the inequality holds since $\psi_i(x) = \mu_i\|x - x_{i-1}\|^2$, thus as $f_i$ is $G$-Lipschitz we have

$$\mu_i\|\hat{x}_i - x_{i-1}\|^2 \leq f_i(x_{i-1}) - f_i(\hat{x}_i) \leq G\|\hat{x}_i - x_{i-1}\|.$$

As event $\mathfrak{E}$ holds, each sample participates in at most $B$ of the optimum estimator computations queries, hence we have that the $\ell_2$-sensitivity of $\tilde{x}_i$ is at most $2\frac{B}{\bar{n}}(\sqrt{C_1 \log n} + 2)G/\mu_i$. Privacy properties of the Gaussian mechanism (Lemma 13) and our choice of $\sigma_i$ now imply that each iterate is $(\alpha, \beta^2/2)$-DP whenever event $\mathfrak{E}$ holds, which proves the claim about privacy.

Let us now prove utility following steps similar to the proof of Theorem 4.4 in [23]. We define the non-private minimizers, $\hat{x}_i = \mathrm{argmin}_{x\in\mathcal{X}} F_i(x)$ and $\hat{x}_0 = x_\star$. We have

$$f(x_k) - f(x_\star) = \sum_{i=1}^{k}[f(\hat{x}_i) - f(\hat{x}_{i-1})] + f(x_k) - f(\hat{x}_k). \tag{22}$$

Using the definitions of $\sigma_i$ and $\eta_i$ in Algorithm 6, we also have that for every $i \geq 1$

$$\mathbb{E}[\|\hat{x}_i - x_i\|^2] \leq 2\mathbb{E}[\|\hat{x}_i - \tilde{x}_i\|^2] + 2\mathbb{E}[\|\tilde{x}_i - x_i\|^2]$$
$$\leq 2\mathbb{E}[\|\hat{x}_i - \tilde{x}_i\|^2] + O\left(\frac{G^2 B^2 \eta_i^2 d \log(n) \log(1/\beta)}{\alpha^2}\right)$$
$$\leq 2\mathbb{E}[\|\hat{x}_i - \tilde{x}_i\|^2] + O\left(\frac{G^2 B^2 \eta^2 d \log(n) \log(1/\beta)}{\alpha^2 2^{8i}}\right).$$

Moreover, using properties of the bounded-optimum estimator from Definition 3, that is, $\|\mathcal{O}_{\delta_i}(F_i, x_{i-1}) - \hat{x}_i\|^2 \leq C_1 G^2 \log(n)/\mu_i^2$ and $\|\mathbb{E}[\mathcal{O}_{\delta_i}(F_i, x_{i-1}) - \hat{x}_i]\|^2 \leq \delta_i^2$, we have by choosing $\delta_i^2 = G^2/\mu_i^2\bar{n} = G^2\eta_i^2\bar{n}$,

$$\mathbb{E}\|\tilde{x}_i - \hat{x}_i\|^2 = \mathbb{E}\left\|\frac{1}{\bar{n}}\sum_{j=1}^{\bar{n}}\mathcal{O}_{\delta_i}(F_i, x_{i-1}) - \hat{x}_i\right\|^2$$
$$\leq \frac{C_1 G^2 \log(n)}{\mu_i^2\bar{n}} + \rho^2 \leq (C_1 + 1)G^2\eta_i^2\bar{n}\log(n).$$

We can now bound the terms in (22). For the second term, the choice of $\eta$ gives

$$\mathbb{E}[f(x_k) - f(\hat{x}_k)] \leq G\mathbb{E}[\|x_k - \hat{x}_k\|]$$
$$\leq G \cdot O\left(G\eta_k\sqrt{\bar{n}\log(n)} + \frac{RB}{2^{6k}}\right)$$
$$\leq G \cdot O\left(\frac{2G\eta\sqrt{\bar{n}\log(n)}}{2^{4k}} + \frac{RB}{2^{6k}}\right)$$
$$\leq O\left(\frac{RG}{n}\right).$$

For the first term in (22), as $F_i$ is $G$-Lipschitz over $\mathcal{X}_i = \{x \in \mathcal{X} : \|x - x_{i-1}\| \leq 2G\eta_i\bar{n}\}$, Theorems 6 and 7 in [46] imply that for all $y \in \mathcal{X}_i$

$$\mathbb{E}[f(\hat{x}_i) - f(y)] \leq \frac{\mathbb{E}[\|y - x_{i-1}\|^2]}{\eta_i\bar{n}} + 2G^2\eta_i,$$

hence we now have

$$\sum_{i=1}^{k} \mathbb{E}[f(\hat{x}_i) - f(\hat{x}_{i-1})] \le \sum_{i=1}^{k} \mu_{i-1} \mathbb{E}[\|\hat{x}_{i-1} - x_{i-1}\|^2] + 2G^2 \eta_i$$

$$\le O\left( \frac{R^2}{\eta \bar{n}} + \sum_{i=2}^{k} \mu_i \left( \frac{G^2 \log(n)}{\mu_i^2 \bar{n}} + \frac{G^2 B^2 \eta_i^2 d \log(n) \log(1/\beta)}{\alpha_i^2} \right) + G^2 \eta_i \right)$$

$$\le O\left( \frac{R^2}{\eta \bar{n}} + \sum_{i=2}^{k} G^2 \eta_i \log(n) + \frac{G^2 B^2 \eta_i d \log(n) \log(1/\beta)}{\alpha_i^2 \bar{n}} \right)$$

$$\le O\left( \frac{R^2}{\eta \bar{n}} + G^2 \eta \log(n) + \sum_{i=2}^{k} 2^{-i} \frac{G^2 B^2 \eta d \log(n) \log(1/\beta)}{\alpha^2 \bar{n}} \right)$$

$$\le GR \cdot O\left( \frac{\log n}{\sqrt{n}} + \frac{B \log(n) \sqrt{d \log(1/\beta)}}{n\alpha} \right),$$

where the last inequality follows since $\bar{n} = n/\lceil \log(n) \rceil$, and $\eta = \frac{R}{G} \min(1/\sqrt{n}, \alpha/B \log(n) \sqrt{d \log(1/\beta)})$. $\qquad\square$

### F.3 The challenges of obtaining a bounded optimum estimator

To highlight the challenge of finding bounded estimators that satisfy Definition 3, let us explain why our MLMC optimum estimator (1) fails to do so. For this estimator, we have (when $2^J \le T_{\max}$)

$$\|\hat{x}_\star - x_\star\| \le \|x_\star - x_0\| + 2^J \|x_J - x_{J-1}\|,$$

where $x_j$ is the output of an ODC algorithm with query budget $2^j$. The ODC property and the triangle inequality then roughly imply that $\|x_j - x_{j-1}\| = O(2^{-j/2}G/\mu)$ and consequently (since $\|x_\star - x_0\| = O(G/\mu)$) we have $\|\hat{x}_\star - x_\star\| = O(2^{J/2}G/\mu) = O(\sqrt{T_{\max}}G/\mu)$ which clearly is not enough to guarantee an $\widetilde{O}(G/\mu)$ bound on $\|\hat{x}_\star - x_\star\|$. Indeed, to guarantee such bound with a similar analysis we would have needed $\|x_j - x_{j-1}\| = O(2^{-j}G/\mu)$. However, this would imply that, by the triangle inequality,

$$\|x_j - x_\star\| = \|x_j - x_\infty\| \le \sum_{k=j+1}^{\infty} \|x_k - x_{k-1}\| = \sum_{k=j+1} O(2^{-k}G/\mu) = O(2^{-j}G/\mu),$$

which contradicts the lower bound on the optimal distance convergence rate in Appendix A.4.

Having explained why the analysis strategy underlying our estimator (1) cannot directly yield a bounded optimum estimator, we discuss two approaches with a potential to solve the problem. The first approach is to apply ODC algorithms on a smooth surrogate of the true objective $F$, for which the faster convergence to the optimum is possible, e.g., using randomized smoothing [20, 32].

The second approach is try to directly bound the $\ell_2$ sensitivity of our MLMC-based approach. In particular, it might be possible to leverage the structure of our estimator (or an improved version thereof) in order to control the $\ell_2$ sensitivity without relying on the boundedness of the estimator as we currently do in the proof of Theorem 6.