# OpenReview forum: "Stochastic Bias-Reduced Gradient Methods"
_NeurIPS.cc/2021/Conference — NeurIPS 2021 Poster_

### Official Review · Reviewer_zGn6 · 2021-07-05

**Rating:** 6
**Confidence:** 3

**Summary:**

The paper uses a multilevel Monte Carlo technique to produce a minimizer with a reduced cost and bias of a Lipschitz strongly convex function $f$ . The paper then applies this technique to obtain a cheap and nearly unbiased estimator for gradient of the Moreau envelope of Lipschitz convex functions, and presents different applications of this technique.

**Ethical Concerns:**

I am not aware of ethical issues regarding this paper

**Limitations And Societal Impact:**

I am not aware of limitations of the paper, except that it is not clear whether the algorithms in the paper are efficient in practice.

**Main Review:**

The current paper uses the Multilevel Monte Carlo method to find an estimator $\hat x_*$ of the minimizer $x_*$ of a function $F$ that satisfies certain assumptions (see Definition 2.1). The bias of $\hat x_*$ is $O(\delta)$, the variance is $O(\log(1/\delta))$ and the expected number of gradient evaluations required by the estimator is  $O(\log(1/\delta))$ (see Proposition 1). This result is interesting and is new to me. Averaging $m$ independent draws of this estimator yields an estimator with bias $O(\delta)$ and variance $O(\log(1/\delta)/m)$, which implies Theorem 1 for a suitable choice of $m$.

Using the fact that the gradient of the Moreau envelope of a Lipschitz convex function can be calculated easily from the proximal operator, the paper shows that their technique can be applied to obtain a cheap and nearly unbiased estimator for the gradient of the Moreau envelope (see Corollary 2.1).

The paper uses this gradient estimator and an accelerated gradient descent algorithm to design a projection-efficient convex optimization algorithm (Theorem 2). The authors note that their algorithm matches, up to polylogarithmic factors, the performance of a previous algorithm by Thekumparampil et al [39].

The authors also use their technique to minimize the maximum of $N$ Lipshitz convex functions with $\tilde O(N\epsilon^{-2/3}+\epsilon^{-2})$ complexity, improving upon a previous result by Carmon et al [14] (Theorem 3).

Finally, the authors use their technique to provide a gradient-efficient composite acceleration algorithm that matches a previous algorithm by Lan [29] up to logarithmic factors (Theorem 4).

The paper also discusses potential applications to private convex optimization.

The results in this paper, especially those that improve upon previous results, seem interesting to me. On the negative side, the paper contains no numerical experiments, and I find it hard to read. Some suggestions:

--It is probably better to present the applications in Section 1.2 in the same order as in later sections.

--It is probably better to clarify the assumptions made on $\psi$, rather than putting them in a footnote (page 4).

--Line 138: the reference [26] shows how to optimize the function $f$, rather than the function $F$. This point needs to be clarified.

--In (1), is it assumed that $J\ge1$?

--It may be better to put Algorithm 9 in the main paper, rather than the appendix.

------after feedback----- The authors address my comments well, so I increased my score.

**Time Spent Reviewing:**

8

---

> ### Author Response · Authors · 2021-08-10
> **Response to reviewer zGn6**
>
> Thanks for the review and for finding our paper novel and interesting. The negative points raised in the review are a lack of numerical experiments, and that the paper was hard to read. Regarding experiments, please see our general response (paragraphs 4 and 5). Regarding readability, we will work to improve it and incorporate the specific issues mentioned in the review (which we address below).
>
> Since experiments are not necessary for supporting our paper’s claims on complexity bounds, and the readability issues appear simple to fix, we would like to ask the reviewer to reconsider if they justify rejecting our paper.
>
> > It is probably better to present the applications in Section 1.2 in the same order as in later sections
>
> We ordered the introduction by the significance of the result, and ordered the sections in a way we thought makes them easier to understand. We will clarify this in the revision.
>
> > It is probably better to clarify the assumptions made on $\psi$ , rather than putting them in a footnote (page 4)
>
> Agreed - we will revise the section accordingly. Thank you for the suggestion.
>
> > --Line 138: the reference [26] shows how to optimize the function $f$, rather than the function $F$. This point needs to be clarified.
>
> Indeed, extending the result of [26] from optimizing a bounded gradient, strongly-convex $f$ to optimizing a strongly-convex $F = f + \psi$ where $f$ has a bounded gradient is a contribution of our paper. We will clarify that in the revision.
>
> > In (1), is it assumed that $J≥1$?
>
> Yes. We will clarify this in the revision. Thank you for noticing the ambiguity.
>
> > It may be better to put Algorithm 9 in the main paper, rather than the appendix.
>
> Given an additional page of content, we would like to add both Algorithm 8 and Algorithm 9 to the main paper, as we believe that they are fairly novel and having them close to the main exposition could improve readability.

---

### Official Review · Reviewer_p9S5 · 2021-07-13

**Rating:** 4
**Confidence:** 4

**Summary:**

The paper develops a new primitive for stochastic optimization – a low-biased and low-cost estimator the minimizer x* for any Lipschitz strongly-convex function. Specially, the author(s) use a multi-level Monte-Carlo method to turn any optimal stochastic gradient method into an estimator x*. The paper also shows the potential of the estimator through four applications.

**Ethics Review Area:**

["I don’t know"]

**Main Review:**

The results could be interesting. However, the authors should explain why the contributions of the paper are important. I am still questionable about the contributions.

1) Can you please explain what is “nearly unbiased gradient estimators” and “nearly linear-time”? Through the paper, I am still confused what these terms are.

2) It is not clear what set X could be. Does it require to be bounded, closed, or convex? Or it could be for any set?

3) For the strongly convex function in R^d, there is a unique optimal solution x*. In order to determine the set X containing the optimal solution x*, the computational cost may be high. If x* is not in X, then the problem of finding argmin_{x in X} F ( x ) may not be interesting since the solution could be on the boundary of X.

4) Assumption 1 should be correct if set X is bounded. However, as discussed in [Nguyen et al, “SGD and Hogwild! Convergence Without the Bounded Gradients Assumption”, ICML 2018], if X = R^d, then the \mu-strongly convex function f cannot be bounded gradients E || \hat{\nabla} F ( x ) ||^2 <= G^2 for all x in R^d.

5) There are four applications listed in the paper, some numerical experiments to verify theory could be useful. I wonder if the author(s) could provide some interesting experiments to support the results.

6) Proposition 2 and Theorem 3 require \epsilon to be smaller than some fixed number. Therefore, the results in Proposition 2 and Theorem 3 cannot be considered as complexity result since it should hold for any \espilon > 0.

7) It seems that this paper can apply only for convex case. I am not sure how to generalize it to the non-convex case since it would have more applications.

I am happy to raise the scores if my concerns have been addressed properly.

---------------------

After rebuttal:

Thank the authors for the response. I am still not convinced with the bounded assumption in the strongly convex case. Moreover, the lack of experiments is the true weakness of this paper. I have decide to keep my score unchanged.

**Time Spent Reviewing:**

5

---

> ### Author Response · Authors · 2021-08-10
> **Response to reviewer p9S5**
>
> Thank you for the feedback. Regarding the comment on the importance of the contribution, please see our general response (specifically paragraphs 3 and 4); we will add more clarifications in the revision. Below we answer your remaining questions in detail.
>
> > 1. Can you please explain what is “nearly unbiased gradient estimators” and “nearly linear-time”?
>
> “Nearly linear time” is a standard term in theoretical computer science that means that the runtime is bounded by $O( n \log^{O(1)}(n) )$ for problem size $n$ (see reference [1] below). By a similar token, by “nearly unbiased” we mean that the dependence on the complexity on the bias is polylogarithmic. We will include a clear definition in the revision.
>
> > 2. It is not clear what set X could be.
>
> $\mathcal{X}$ needs to be convex and closed (standard assumptions in convex optimization) - we will clarify this in the revision. Most of our applications also require $\mathcal{X}$ to be bounded and our theorems make this assumption explicit by requiring $\mathcal{X}$ to be contained in a Euclidean ball.
>
> > 3. The problem of finding argmin_{x in X} F ( x ) may not be interesting since the solution could be on the boundary of X.
>
> Assuming that the domain is bounded is standard. In problems that are originally unconstrained we can often use a doubling scheme to find a domain size where the minimizer is in its interior without increasing the computational cost by more than a constant factor. In other applications, the domain encodes important constraints on the problem and solutions on its boundary are interesting.
>
> > 4. [A] strongly convex function f cannot be bounded gradients
>
> This is precisely why we only assume that $\mathbb{E} \| \hat{\nabla} f ( x ) \|^2$ is bounded while the sum $f + \psi$ is strongly convex (and inevitably has unbounded gradients on unbounded sets). In particular, we do not assume that $f$ is strongly convex.
>
> > some numerical experiments to verify theory could be useful
>
> Please see our general response (paragraphs 4 and 5)
>
> > the results in Proposition 2 and Theorem 3 cannot be considered as complexity result since it should hold for any \espilon > 0
>
> Assuming that $\epsilon$ is smaller than a trivial threshold is standard in optimization complexity results. In Proposition 2, every point has error at most $GR$ and thus for larger epsilon any point can be output. For Theorem 3, when $\epsilon$ is larger that the threshold, subgradient descent solves the problem deterministically with at most $O(N \log^2 N)$ queries.
>
> > It seems that this paper can apply only for convex case
>
> While the non-convex case is outside the scope of the paper, we believe our results are quite likely to be relevant there as well. In particular, for smooth non-convex functions (as well as the more general class of weakly-convex functions [2]) the problem of computing proximal points with sufficiently high regularization is strongly convex and our estimator applies. Such non-convex proximal points play an important role in non-convex optimization [2] with applications in deep learning [e.g., 3]. This is a good point, and we will discuss it in the revision.
>
> > I am happy to raise the scores if my concerns have been addressed properly.
>
> We believe that our answers properly address the reviewer’s concerns and would also be happy if the score is raised :).
>
> _References_
>
> [1] Yuri Gurevich, Saharon Shelah, Nearly linear time, International Symposium on Logical Foundations of Computer Science, 1989
>
> [2] Damek Davis, Dmitriy Drusvyatskiy, Stochastic model-based minimization of weakly convex functions, SIAM Journal on Optimization 2019
>
> [3] Aman Sinha, Hongseok Namkoong, Riccardo Volpi, John Duchi,  ​​Certifying Some Distributional Robustness with Principled Adversarial Training, ICLR 2018

---

### Official Review · Reviewer_DN81 · 2021-07-16

**Rating:** 7
**Confidence:** 4

**Summary:**

The paper proposes a constructive way to obtain an estimator of the minimizer of a Lipschitz strongly convex function $f$. Assuming that we have an access to a stochastic subgradient of $f$, we can run a variant of SGD that guarantees good complexity. Furthermore, the authors demonstrate on several examples how this estimator leads to new algorithms with a better complexity.


I think it is a good paper.


**Ethical Concerns:**

0

**Ethics Review Area:**

["I don’t know"]

**Limitations And Societal Impact:**

0

**Main Review:**


1. Although for optimization tasks (projection-efficient or gradient-efficient methods) different algorithms with the same complexity were known, I would say the proposed idea is more natural and easier to remember. The chapter with a non-smooth private convex optimization was not that interesting, especially taking into account the need to consider conjecture as an assumption.

2. Comparing the proposed approach with [32] or [46], the authors state that the former is a "simple alternative". I understand it as previous approaches are more complicated. However, from the algorithmic point of view I don't see it, perhaps even the opposite.

3. Corollary 2.1: I am not sure I understand the final complexity. In order to apply previous result, we have to use $\sigma:=\frac{\sigma}{\lambda}$ and $\delta := \frac{\delta}{\lambda}$, but this will give us a slightly different fractions in $O(\cdot)$.

4. I only checked mathematics for main results relating to the stochastic estimator, and they seem correct. The rest is a rather tedious calculation of various complexity bounds. Although it is possible that the authors have made a mistake, there is nothing contradictory in their statements and I am inclined to trust their results.



__Minor__:
1. line 74: two instead.
2. line 84: subproblems.
3. line 124: what is $X$?
4. line 173: Why do we ignore $\sigma$ as a constant in $\tilde O(1)$?
5. line 263: Why is $O(L^2R^2/\epsilon^2)$ optimal? Subgradient method needs $O(LR/\epsilon^2)$.
6. The authors call their stochastic estimator as low-bias and low-cost. To me it sounds a big vague. First, these two notions are dependent: the smaller the bias, the larger the cost. Second, in general why $O(1/\sigma^2)$ is considered to be as a low cost?



**Time Spent Reviewing:**

0

---

> ### Author Response · Authors · 2021-08-10
> **Response to reviewer DN81**
>
> Thank you for a careful and appreciative review. We address each comment below.
>
> > 1. The chapter with a non-smooth private convex optimization was not that interesting, especially taking into account the need to consider conjecture as an assumption
>
> Determining the complexity of non-smooth private optimization is an important open problem in private learning. Our technique provides a new and natural approach to this problem, and Section 6 pinpoints a precise difficulty in improving upon our result. While a proof of the conjecture would have been much better, we believe that even in its current form Section 6 will be of interest to researchers working in differential privacy.
>
> > 2. the authors state that [the proposed] is a "simple alternative" … However, from the algorithmic point of view I don't see it, perhaps even the opposite.
>
> While our algorithm may be of similar complexity as a whole to the ones given in [23,46], we remark that our paper is primarily theoretical in nature. Thus, our notion of “simpler” is first and foremost “easier to analyze,” and also “natural and easier to remember” (as noted by the reviewer). Our approach is significantly more modular than that of [32,46] and arises as a direct combination of our MLMC-based bias reduction and standard accelerated methods.  We agree that “simple” could mean other things as well, and will clarify it in our paper.
>
> > 3. Corollary 2.1
>
> You are correct: the correct complexity term is $O( \log \frac{G}{\min\{\delta, \sigma\}} + \frac{G^2}{\sigma^2}   \log^2 \frac{G}{\min\{\delta, \sigma\}} )$. This error in logarithmic terms does not propagate into subsequent results. Thank you for pointing this out!
>
> > (Typos)
>
> Thank you, we will fix these typos in the revision.
>
> > line 124: what is X?
>
> X is the function’s domain (convex, closed and, for most applications, bounded). We will include it in the notation section.
>
> > line 173: Why do we ignore σ as a constant in $\widetilde{O}(1)$?
>
> Note that the variance of the basic stochastic gradient estimator is $O(G^2)$. Therefore, to get an estimate of the Moreau envelope gradient with comparable variance we may set sigma^2 = G^2, resulting in $\widetilde{O}(1)$ work. We will clarify this point (which we make in the introduction) in Section 1 as well.
>
> > line 263: Why is $O(L^2 R^2/\epsilon^2)$ optimal? Subgradient method needs $O(LR/\epsilon^2)$.
>
> If $f$ is a convex function with Lipschitz constant $L$ and we have a point $x_0$ which is distance $R$ from the minimizer of $f$, the subgradient method requires $O(L^2 R^2 /\epsilon^2)$ iterations to compute an $\epsilon$-additive error minimizer - see, e.g., Table 1.1 in “Convex Optimization: Algorithms and Complexity” by Sébastien Bubeck.
>
> > low-bias and low-cost ... sounds a big vague… in general why $O(1/\sigma^2)$ is considered to be as a low cost?
>
> We call the estimator “low-bias” because we can make the bias as low as machine precision, while retaining the variance of the naive estimator and increasing the expected computational cost by a logarithmic factor.  In some cases we can even obtain truly unbiased estimators by introducing a weak (logarithmic) dependence on problem dimension (see Appendix A). Regarding “low cost”, the benchmark is the variance of and number of gradients computed by a typical gradient estimator - see response to question about line 173 above. We will be more explicit on both these points in the revision.

---

> > ### Comment · Reviewer_DN81 · 2021-08-30
> > **Thanks authors for their answers**
> >
> > I understand it is a minor thing. I pointed out that the paper has inaccuracy about the complexity of the subgradient method. How can it be $L^2 R^2$? I do not need to check the Table 1.1, to me it is obvious that power $L^2$ cannot be right even without knowing what is the optimal complexity: otherwise, by scaling $f\mapsto cf$ we can get arbitrary good complexity.

---

> > > ### Author Response · Authors · 2021-08-30
> > > **Reply**
> > >
> > > Note that if you scale $f \to c f$ then we have $L \to c L$ and the suboptimality $f(x) - f(x^\star)$ is scaled by $c$ as well, i.e., $\epsilon \to c \epsilon$. Therefore, the quantity $L^2 R^2 / \epsilon^2$ remains invariant under such scaling while $LR^2 / \epsilon^2$ will be scaled by $1/c$. This shows that complexity proportional to $1/\epsilon^2$ must be proportional to $L^2$ as well. We would be happy to provide additional clarification if necessary - please let us know.

---

> > > > ### Comment · Reviewer_DN81 · 2021-09-01
> > > > **Sorry for the confusion**
> > > >
> > > > Yes, I was wrong. I got $f(x_k) - f_* \leq \frac{LR}{\sqrt k}$, but forgot to express complexity in terms of iterations $k$.
> > > >
> > > > Thanks for clarification.

---

### Official Review · Reviewer_x1ay · 2021-07-18

**Rating:** 7
**Confidence:** 4

**Summary:**

This paper proposes a procedure that converts any optimal stochastic gradient method into an estimator of the optimum $x^*$ using the multilevel Monte-Carlo approach. The estimator has a smaller expected cost of stochastic gradient computation compared with the original stochastic gradient method. The authors demonstrate that the proposed estimator can be combined with existing methods to improve the computation complexity and provide alternative algorithms to other methods in different applications.


**Limitations And Societal Impact:**

The major limitation of this paper is the lack of empirical comparison with existing methods, while its main contribution is to develop efficient and nearly unbiased gradient estimators, which are claimed to achieve optimal bias to the optimum point with much fewer stochastic gradient evaluations.

I do not see any potential negative societal impact directly related to this work


**Main Review:**

This paper has a strong theoretical impact on stochastic optimization due to the introduction of the simple optimum estimator, which has potentially lots of applications in the field. However, some necessary empirical evidence is needed to demonstrate the efficiency and the advantage of the proposed estimator versus existing alternatives.

Can you provide an example where you have to call ODC multiple times to get the estimator in (1)? In addition, there are many theoretical results in the main paper that are not followed by any discussion on the optimality and the comparison with existing methods. For example, for Theorem 4, it is as least interesting to the readers in what scenarios the proposed Algorithm 5 will have advantage over the method in [32]. Even if your goal is not to beat their algorithm, it is still helpful to convey the proper message to readers how the results stated here are positioned in the literature.

It is a little bit concerning to claim Theorem 5 as a theorem since it is not completely proved. The conclusion in Theorem 5 depends on Conjecture 1, which is not proved but shown to be hard to prove at the end of this paper.

It is nice to see the theoretical improvement of the proposed estimator. However, since its biggest advantage is to reduce the sampling cost of stochastic gradient estimators in computation, there should be at least some simulation study on the applications discussed in this paper to compare the empirical performance of the estimator with other alternatives.

Typos:
L10: “up logarithmic factors”
L74: “we instead instead apply ...”

---post rebuttal---
I have read the authors’ responses. I will keep my score as “accept” as long as the authors made the promised revision about discussions of theoretical results and conjecture.


**Time Spent Reviewing:**

6

---

> ### Author Response · Authors · 2021-08-10
> **Response to reviewer x1ay**
>
> Thank you for the positive review and particularly for the assessment that our paper has “a strong theoretical impact and has potentially lots of applications.” We share the opinion that it is likely to have a substantial impact. Below, we address your comments in detail.
>
> > Can you provide an example where you have to call ODC multiple times to get the estimator in (1)?
>
> If the question is referring to the comment in lines 147-149, note that naive implementation of the estimator (1) needs to calculate $x_{0}$, $x_{J-1}$ and $x_{J}$ which would require three calls to ODC, if we think about it as a generic black box procedure satisfying Definition 2.1. For example, extracting $x_{J-1}$ from the ODC algorithm of Rakhlin, Shamir and Sridharan would require some extra work. Please let us know if additional clarification is required.
>
> > There are many theoretical results in the main paper that are not followed by any discussion on the optimality and the comparison with existing methods.
>
> Algorithms 5 and 3 do not improve on previous theoretical guarantees in the literature (they match up to logarithmic factors), but we believe that they are easier to analyze and exemplify the use of our proposed optimum estimator. Our main theoretical improvement over past work is with Algorithm 4. We explain this in the introductions section but will add additional discussion within each section of the main paper to clarify this.
>
> > It is a little bit concerning to claim Theorem 5 as a theorem since it is not completely proved
>
> In the revision we will change Conjecture 1 to a definition and rename “Theorem 5” to “Theorem 5 (conditional)”.
>
> > there should be at least some simulation study on the applications discussed in this paper to compare the empirical performance of the estimator with other alternatives.
>
> Please see our general response regarding the lack of experiments in this paper.
>
> > (Typos)
>
> Thank you for catching these typos - we will correct them in the revision.

---

### Author Response · Authors · 2021-08-10
**General response**

We thank the reviewers for their time and are happy that they were unanimous in finding our paper interesting, with reviewer x1ay predicting a “strong theoretical impact on stochastic optimization.” We agree with this assessment and indeed were ourselves surprised and excited by the discovery of low-bias optimum estimators. Consequently, we were surprised by the somewhat low scores our paper received.

Other than typos and technical questions, the reviews primarily raise a single point of critique: the paper does not have experiments that test the practical performance of the algorithms we analyze. However, we believe that this critique misses what the paper is about. To explain why this is so, let us briefly restate our main contribution.

The main contributions of our paper are (a) to provide a new broadly applicable tool for reasoning about the complexity of fundamental, large-scale optimization problems, and (b) apply this tool to the problem of  minimizing the maximum of $N$ convex functions. To make point (a) that our new tool (nearly-unbiased optimum estimation) is broadly applicable, we also show how it easily recovers recent nontrivial complexity bounds and discuss its potential impact on non-smooth differentially private optimization. Pursuing point (b), we settle (up to logarithmic factors) the complexity of minimizing the maximum of $N$ convex functions, arriving at the following surprising result: when $N$ is not very large (less that $\epsilon^{-4/3}$), maximum risk minimization has similar complexity to empirical (i.e., average) risk minimization.

We believe that experiments are not essential to our paper because proposing practical optimization techniques is not part of our main contributions. Following a long tradition in theoretical computer science, our paper uses algorithms as a means to derive upper bounds on computational complexity. In particular, we *do not* claim that the algorithms we analyze perform well in practice. Consequently, experiments meant to test whether they do would not add or subtract from our main claims and are therefore, we believe, best left to follow-up work. Indeed, we hope that, in addition to helping theoreticians prove new complexity bounds, our paper will inspire practitioners to experiment with our proposed optimum estimator: we submitted the paper to NeurIPS with hopes of reaching both audiences.

Finally, we would like to point out that there is an extensive precedent for NeurIPS optimization papers that, like our own, do not have experiments. For example, looking at the three optimization-themed orals and spotlight tracks from NeurIPS 2020 (tracks 21, 30 and 32), we found 7 (out of 43) papers accepted for spotlight and oral presentation that did not include experiments.

---

### Decision · Program_Chairs · 2021-09-27

**Decision:**

Accept (Poster)

**Comment:**

Although there was no consensus here, I think the paper should be accepted based on the the new theoretical results and the novelty of the algorithm.

One issue that was discussed quite a bit was the lack of empirical evaluation, which is strange for a NeurIPS paper proposing a new algorithm (as opposed to COLT, for example). Most NeurIPS readers will ask "does this algorithm actually work in practice", and without an answer to this the paper seems incomplete. Indeed, without this information it is likely that several poor graduate students will spend time implementing and trying out the method. The authors could save these people by doing it once, particularly if the algorithm does not work well in practice. Further, there might be some hidden assumption/disadvantage that makes the algorithm work much-worse in practice than the theory indicates (a classic example is the finite-termination of conjugate gradient which never actually happens empirically). Because of this, I am recommending acceptance but do not think the paper should be considered for a spotlight or oral or awards.

Finally, the camera-ready version should *at least* include a qualitative discussion of how the algorithm works empirically. Some examples:
- "Despite its theoretical properties, we found that the algorithm did not perform well empirically. More research is needed on this topic."
- "Although theoretically-faster in the worst case, the algorithm performed similarly to competing methods when we implemented it."
- "The algorithm outperformed method YYY in a preliminary implementation on the standard problem of ZZZ, although we did not explore extensive numerical experiments."
- "The algorithm converged very-quickly in practice, although more work needs to be done on decreasing the iteration cost."
- "The algorithm does not appear practical at all, but we hope our work will inspire others to develop practical variants with similar complexity bounds."